



# Recovery of the first ever multi-year lidar dataset of the stratospheric aerosol layer, from Lexington, MA, and Fairbanks, AK, January 1964 to July 1965.

Juan-Carlos Antuña-Marrero[1], Graham W. Mann[2,3], John Barnes[4], Albeth Rodríguez-Vega[5], Sarah Shallcross[2], Sandip Dhomse[2,3], Giorgio Fiocco[†] and Gerald W. Grams[†]

[1]Departamento de Física Teórica, Atómica y Óptica, Universidad de Valladolid, 47002, Spain

[2]School of Earth and Environment, University of Leeds, Leeds, LS2 9JT, UK.

[3]National Centre for Atmospheric Science (NCAS-Climate), University of Leeds, Leeds, UK

[4]NOAA ESRL Global Monitoring Division, Colorado, US

[5]Grupo de Óptica Atmosférica de Camagüey, Centro Meteorológico de Camagüey, INSMET, Cuba

[†]Deceased

*Correspondence to*: Juan-Carlos Antuña-Marrero (antuna@goa.uva.es)

**Abstract.** We report the recovery and processing methodology of the first ever multi-year lidar dataset of the stratospheric aerosol layer. A Q-switched Ruby lidar measured 66 vertical profiles of 694nm attenuated backscatter at Lexington, Massachusetts between January 1964 and August 1965, with an additional 9 profile measurements conducted from College, Alaska during July and August 1964. We describe the processing of the recovered lidar backscattering ratio profiles to produce mid-visible (532nm) stratospheric aerosol extinction profiles ($sAEP_{532}$) and stratospheric aerosol optical depth ($sAOD_{532}$) measurements, utilizing a number of contemporary measurements of several different atmospheric variables. Stratospheric soundings of temperature, and pressure generate an accurate local molecular backscattering profile, with nearby ozone soundings determining the ozone absorption, those profiles then used to correct for two-way ozone transmittance. Two-way aerosol transmittance corrections were also applied based on nearby observations of total aerosol optical depth (across the troposphere and stratosphere) from sun photometer measurements. We show the two-way transmittance correction has substantial effects on the retrieved $sAEP_{532}$ and $sAOD_{532}$, calculated without the corrections resulting in substantially lower values of both variables, as it was not applied in the original processing producing the lidar scattering ratio profiles we rescued. The combined transmittance corrections causes the aerosol extinction to increase by 67 % for Lexington and 27 % for Fairbanks, for $sAOD_{532}$ the increases 66 % and 26 % respectively. Comparing the magnitudes of the aerosol extinction and sAOD with the few contemporary available measurements reported show a better agreement in the case of the two way transmittance corrected values.

The sAEP and sAOD timeseries at Lexington show a surprisingly large degree of variability, three periods where the stratospheric aerosol layer had suddenly elevated optical thickness, the highest $sAOD_{532}$ of 0.07 measured at the end of March 1965. The two other periods of enhanced $sAOD_{532}$ are both two-month periods where the lidars show more than 1 night where retrieved $sAOD_{532}$ exceeded 0.05: in January and February 1964 and November and December 1964. Interactive stratospheric aerosol model simulations of the 1963 Agung cloud illustrate that although substantial variation in mid-latitude $sAOD_{532}$ is expected from the seasonal cycle in the Brewer-Dobson circulation, the Agung cloud dispersion will have caused much slower increase than the more episodic variations observed, with also different timing, elevated optical thickness from Agung occurring in winter and spring. The abruptness and timing of the steadily increasing sAOD from January to July 1965 suggests this variation was from a different source than Agung, possibly from one or both of the two VEI3 eruptions that occurred in 1964/65: Trident, Alaska and Vestmannaeyjar, Heimey, south of Iceland. A detailed error analysis of the uncertainties in each of the variables involved in the processing chain was conducted, relative errors of 54 % for Fairbanks and 44 % Lexington for the uncorrected $sAEP_{532}$, corrected $sAEP_{532}$ of 61 % and 64 % respectively.



The analysis of the uncertainties identified variables that, with additional data recovery and reprocessing could reduce these relative error levels. Data described in this work are available at *https://doi.pangaea.de/10.1594/PANGAEA.922105 (Dataset in Review)* (Antuña-Marrero et al., 2020a).

45

## 1. Introduction:

The abrupt enhancements to the stratospheric aerosol layer from historical large magnitude volcanic eruptions (e.g. Deshler, 2008) cause substantial radiative forcings to the Earth's climate system, and reducing their uncertainty remains an important priority for international scientific research, volcanic forcings being the strongest driver of natural climate variability (e.g. Hansen, 1978; Robock, 2000). One of the co-ordinated multi-model experiments within the current international ISA-MIP activity (Interactive Stratospheric Aerosol Model Intercomparison Project, Timmreck et al., 2018), involves simulations of the volcanic aerosol clouds from the largest volcanic eruptions in the last century, Mt. Agung in 1963, El Chichón in 1982 and Mt. Pinatubo in 1991. The main motivation for this HErSEA multi-model experiment (Historical Eruption $SO_2$ Emission Assessment) is to gather stratospheric aerosol observations to evaluate the model simulations, and understand the current diversity in the sulphur emission amount and altitude distribution interactive stratospheric aerosol models use when simulating the Pinatubo aerosol cloud (see section 3.3.2 of Timmreck et al., 2008). The first of the ISA-MIP modelling groups to present results from all three of the HErSEA eruption cloud experiments was recently published (Dhomse et al., 2020), with another recent study focusing on assessing the variability and global distribution of the Agung aerosol cloud (Niemeier et al., 2019).

Whereas the models participating in ISA-MIP simulate volcanic aerosol clouds interactively, the historical climate model simulations that provide the main basis for attributing past climate variability (Hegerl and Schwierz, 2011; Gillett et al., 2016) use prescribed volcanic forcing datasets (e.g. Sato et al., 1993; Ammann et al., 2003; Luo, 2016; Thomason et al., 2018), reference aerosol optical properties used to enact volcanic surface cooling. The observational data constraining the Agung aerosol cloud in both the interactive models and for the volcanic forcing datasets has hitherto tended to be based on column optical properties measured at the surface, primarily the extensive synthesis of surface radiation observations summarized by Dyer and Hicks (1968), with additional turbidity anomaly data from astronomical measurements of the atmospheric attenuation of starlight (Stothers, 2001). Although the literature includes several papers reporting profile measurements of the Agung aerosol cloud from balloon measurements (Rosen, 1964; 1968), lidars (Fiocco and Grams, 1964; Clemesha et al., 1966) and searchlights (Elterman and Campbell, 1964), no profile dataset of Agung backscatter ratio or aerosol extinction has yet been available to the scientific community. Whereas the Jamaica lidar (Clemesha et al., 1966) also measured the Agung cloud, the first multi-year dataset of lidar measurements of the volcanic aerosol from that eruption was conducted from Lexington, Massachusetts from January 1964 to August 1965 (Grams and Fiocco, 1967, hereinafter **GF-67**). No digital record of these lidar measurements existed until now, the data apparently only presented in Figures of the lidar backscattering ratio profiles within published scientific papers, providing only quantitative information about the altitude of the Agung aerosol cloud. Although the descent in the peak of the backscatter ratio profile from Lexington is analysed within GF-67, only limited estimates of the cloud's aerosol extinction exist (2 x $10^{-3}$ km$^{-1}$ at 16 km and the aerosol optical depth of 0.015 (Deirmejian, 1971) were been produced.

We discovered however, after initial search of digital archives at several institutions, that the original lidar backscattering ratio profile measurements from the Lexington and Alaska 1964/5 soundings are fully tabulated in Gerald W. Grams PhD thesis conducted under the supervision of Prof. Giogio Fiocco (Grams, 1966) hereinafter identified as **G-66**. Fortunately, at those times it was quite common for some observational datasets to be tabulated within PhD theses or grant reports etc., a practice that after several decades is becoming required again, with many journals now mandating authors to make available the data they use via a recognised open-access data archive.





Dhomse et al. (2020) used preliminarily processed lidar data from Lexington, MA, one of the two sites reported in (**GF-67**) to compare model aerosol extinction at 16 km with lidar observations, finding good agreement. They also noted the large differences between the CMIP5 and CMIP6 volcanic aerosol datasets for the Agung and El Chichón periods, pointing out the importance of reducing this uncertainty by reconciling the datasets with additional stratospheric aerosol observations. There only an initial (preliminary) single-level version the Lexington aerosol extinction dataset was used, our analysis here

completing the processing of the full vertical profile (between 12 and 24km), with two-way transmittance corrections applied to the aerosol backscatter ratio, aerosol extinction and optical depth datasets, also with a detailed and transparent assessment of the relative error in each metric.

 This work is a contribution to the Data Rescue activity of the Stratospheric Sulfur and its Role in Climate (SSiRC) a SPARC initiative (SSiRC, 2020), following a recent dataset recovery of two ship-borne lidar datasets that measured the

95 progression of the highly uncertain "tropical core" of the Pinatubo aerosol cloud in July 17th to September 13th 1991, 4-12 weeks after the 15th June 1991 Pinatubo eruption (Antuña-Marrero et al., 2020b). Those datasets were identified priority for the data recovery, being in is the period when the Stratospheric Aerosols and Gas Experiment II (SAGE II) satellite could only observe the stratospheric aerosol above around 22 km due to the extreme opacity of the aerosols (McCormick and Veiga, 1992).

## 2. Materials and Methods.

### 2.1 Lidar instrumentation:

The first successful laser radar ranging experiment was conducted at the Research Laboratory of Electronics, Massachusetts Institute of Technology, at Lexington, Massachusetts, and consisted of analysing the return signal from a very high frequency (nano-second) laser and for the 60-140km altitude range (Smullin and Fiocco, 1962). The research team, led by Prof. Giorgio Fiocco, continued developing applications of the lidar for atmospheric research. Scattering layers were detected in the upper atmosphere between 110 and 140 km (Fiocco and Smullin, 1963) and were interpreted to originate

from meteoric fragments entering the outer atmosphere (Fiocco and Colombo, 1964). After some changes and improvements, stratospheric aerosols were detected and the first lidar measurements of the stratospheric aerosol layer began (Fiocco and Grams, 1964).

 The schematic diagram and a photo of the instrument are in figures 3 and 4 of **G-66** respectively. There are listed also the main features of the lidar instruments used for the measurements at Lexington and College, Alaska, reported in its table 1,

reproduced below. Both instruments used a Q-switched ruby laser, at the 694 nm wavelength.

| Observation period | January-May | July-August | October 1964 |
|---|---|---|---|
| Observation site | Lexington | College | Lexington |
| Transmitted wavelength | 0.694 μm | 0.694 μm | 0.694 μm |
| Pulse length | < 1 μs | < 1 μs | < 1 μs |
| Pulse energy | ~ 0.5 Joule | ~ 0.5 Joule | ~ 2 Joule |
| Pulse repetition rate | ~ 0.1 s$^{-1}$ | ~ 0.1 s$^{-1}$ | ~ 0.5 s$^{-1}$ |
| Transmitted beam width | < 1 mrad | < 1 mrad | < 1 mrad |
| Transmitter efficiency (estimated) | ~ 75% | ~ 75% | ~ 75% |
| Aperture of receiving telescope | 40 cm | 30 cm | 40 cm |
| Receiver efficiency (estimated) | ~ 30% | ~ 30% | ~ 30% |
| Quantum efficiency of photodetector | ~ 5% | ~ 5% | ~ 5% |
| Bandwidth of receiver filter | 20 Å | 3 Å | 6 Å |



**Table 1: Technical features of the lidar instruments operated at Lexington and College, Fairbanks.**
An additional set of relevant features of the instrument follows. The fluorescent emission after the laser pulse has been emitted, was prevented incorporating a small rotating shutter into the transmitting unit, synchronized with the Q-switching device. The sensing unit for the backscattered signal consisted in an astronomical telescope, with an interference filter and a photomultiplier synchronized to another rotating shutter to avoid its exposition to the intense returns from short distances. The photomultiplier was cooled with methanol and dry ice, to reduce the levels of its dark current (**G-66**).

*2.2 Lidar measurements:*

Lidar observations were conducted at Lexington, Massachusetts (42° 25'N, 71° 15'W) and also at College, (64° 53'N, 148°3'W) located in the city of Fairbanks, Alaska, hereinafter identified as Fairbanks. The measurements were supported by the NASA Grant NGR-22-009-131. One of the semi-annual reports mention more than 100 measurements conducted (Fiocco, 1966a). However the amount of total profiles appearing in Grams PhD dissertation was 75. Nine days of measurements from July 26 to August 21, 1964, were conducted in Fairbanks. At Lexington, Massachusetts, 23 days of measurements from January 14 to May 20, 1964, and 43 days from October 11, 1964 to July 21, 1965 were made. At both sites measurements were conducted at night to avoid the contribution of daylight to the signals registered by the photomultipliers.

The lidar signal returns at both sites were registered photographically from oscilloscopes covering up to 40 km and then digitized. Then the digitized lidar return signals from a set of daily laser shots were averaged in 1 km bins (**G-66**; **GF-67**).

*2.3 Backscattering ratios in the original lidar dataset:*

It is well known that solving the equation for the single wavelength elastic lidar is an ill-posed problem. The single returned signal, is the result of two main species, molecules and particles, making it necessary to the use of additional information to estimate the solution (eg, Kovalev, 2015). That is the reason why still today processing single wavelength lidar profiles remains a challenge. Considering this fact we may understand the magnitude of the challenge confronted by Prof. Girogio Fiocco and then BSc Gerald W. Grams, Prof. Fiocco's PhD student, when they conducted the processing of the first ever set of lidar returned signals from stratospheric aerosols.

We now describe the procedure applied in **G-66** to derive SRo(z). The average photoelectron flux registered by the photomultiplier was represented by the expression:

$$\frac{dn(z)}{dt} = K \frac{N_A(z)}{z^2} \qquad (1)$$

Where z is the altitude, $n_z$ is the number of photons at the altitude z, $N_A$ is the molecular number density at altitude z, obtained from the US 1962 Standard Atmosphere. K is a constant resulting from all the terms not depending on the altitude in the optical radar equation, including $T_{2w}^2$, the two-way atmospheric transmittance (see **G-66** for more details). The assumption of a constant value for $T_{2w}^2$ was based on the atmospheric attenuation model proposed by Elterman (1964). The model provided magnitudes of the molecular and aerosol scattering and the ozone absorption, showing that almost all attenuation of the laser bean occurs in the troposphere. The model allowed an estimate at 700 nm of the variability of the term $T_{2w}^2$ between 10 and 30 km which was below 3%. The correction of the returned signal, associated with the two way transmittance of the laser beam throughout the atmosphere, was then neglected and it was assumed that the atmospheric attenuation term was constant.

The returned signal from a set of laser shots was averaged in time and in altitude to a resolution of 1 km between 12 and 30 km. Next, the ratios between the averaged signal at each level and the values at the same level of the right side of the equation (1) were calculated for each profile between 12 and 30 km. A final step consisted in normalizing the ratios

calculated in each profile between 12 and 24 km. To that end, for each profile the average value between 25 and 30 km of the ratios calculated in the former step were determined. Then for each profile the ratios in the altitude range 12 and 24 km were divided by the average value of the ratios between 25 and 30 km from the same profile. The resulting values were considered to be the backscattering ratio ($SR_o(\lambda, z)$): the ratio between the total (aerosols + molecules) backscattering divided by the molecular backscattering. The normalization procedure assigned the backscattering ratio to be equal to one above 25 km, after assuming the contribution from aerosols was negligible compared to the molecular at those levels. This assumption would lead to an under-estimate of stratospheric aerosol since there would have been aerosol at these altitudes (Russell, et al., 1979).

The $SR_o(\lambda, z)$ from the lidar measurements conducted at Lexington and Fairbanks were reported in tabular format in the Gerald W. Grams PhD Thesis (**G-66**), and cited in the acknowledgements section of **GF-67**. It was the unique reference of its existence, the clue that guided us in our search for the lidar measurements.

*2.4 Algorithm and complementary datasets used in the processing:*

Far beyond the mere rescue and deposit on public repositories of datasets, mainly from stratospheric aerosols from past volcanic eruptions from the 60's to the present, the SSiRC Data Rescue Activity is committed, whenever it will possible, to re-calibrate each dataset and determine its levels of uncertainties (SSiRC, 2020). Because some stratospheric aerosol lidar datasets have already been identified and located, we consider that its recalibration or reprocessing should be conducted using a standardized algorithm, to guarantee the best possible consistence among the different lidar datasets. To contribute to this task we describe below the processing algorithm we used as a first step in that direction.

The lidar backscattering ratio ($SR(\lambda, z)$) is defined as the ratio between the total backscatter ($\beta_T(\lambda, z)$) and the molecular backscatter $\beta_m(\lambda, z)$, at the altitude z and wavelength $\lambda$. $\beta_T(\lambda, z)$ is the sum of $\beta_m(\lambda, z)$ and the aerosol attenuated backscatter ($\beta_a^A(\lambda, z)$). That definition is associated to the fact that in the retrieval of $SR_o(z)$ the two-way total transmittance($T_T^2$) correction was neglected (Hosteler et al., 2006):

$$SR(\lambda, z) = \left\{ \frac{\beta_m(\lambda, z) + \beta_a^A(\lambda, z)}{\beta_m(\lambda, z)} \right\} \qquad (2)$$

$\beta_m(\lambda, z)$ is derived using the equation:

$$\beta_m(\lambda, z) = \frac{\sigma_m(\lambda, z)}{S_m} = \frac{3 \, \sigma_m(\lambda, z)}{8 \pi} \qquad (3)$$

where $S_m = (8\pi/3)k_{bw}$ is the molecular extinction to backscatter ratio for the molecular scattering, commonly approximated by $8\pi/3$ (Collins and Russell, 1976) after neglecting the dispersion of the refractive index and the King factor of the air represented by $k_{bw}$. The volume molecular scattering coefficient, $\sigma_m(\lambda, z)$ is determined by the equation:

$$\sigma_m(\lambda, z) = \frac{N_A \, Pr(z)}{R_a \, Temp(z)} \, Q_s(\lambda) \qquad (4)$$

Where $N_A = 6.02214 \times 10^{23}$ (1/mol) is Avogadro's number; Ra = 8.314472 (J/K/mol) is the gas constant and $Q_s(\lambda)$ the total molecular scattering cross section per molecule for the standard air. The derived equation for $Q_s(\lambda)$ for standard air is (Hostetler et al., 2006):

$$Q_s(\lambda) = 4.5102 \times 10^{-27} \left[ \frac{\lambda(nm)}{550} \right]^{-4.025 - 0.05627 \times \left[ \frac{\lambda(nm)}{550} \right]^{-1.647}} \qquad (5)$$

Then from equation 2 we retrieved $\beta_a^A(\lambda, z)$, using the expression:

$$\beta_a^A(\lambda, z) = (SR_o(z) - 1)\beta_m(\lambda, z) \qquad (6)$$

To derive the aerosol backscatter profiles at 532 nm ($\beta_a(532, z)$) we used the wavelength exponents (kb(z, t)) for aerosol backscatter in the range of wavelengths between 532 and 694 nm derived for the stratospheric aerosols produced by the 1991 Mt Pinatubo eruption (Jäger and Deshler, 2002) according to the expression:

$$\beta_a(532,z) = \left[\frac{532}{694}\right]^{kb(z,t)} \frac{\beta_a^A(694,z)}{T_m^2(694,z)\,T_{O3}^2(694,z)} \quad (7)$$

Applying in addition the corrections for, $T_m^2(694,z)$ and $T_{O3}^2(694,z)$ the two-way molecular and ozone transmittances at $\lambda$ = 694 nm to the $\beta_a^A(694,z)$. The generic definition of the two-way tansmittance is:

$$T_j^2(\lambda,z) = e^{-2\int_{sup}^Z \alpha_j(\lambda,z)dz} \quad (8)$$

With the sub index j representing in $\alpha_j(\lambda,z)$ the vertical profiles of the scattering by the molecules $\alpha_m(\lambda,z)$), ozone ($\alpha_{O3}(\lambda,z)$) and aerosols ($\alpha_a(\lambda,z)$).

Then the aerosol extinction ($\alpha_a(532,z)$) is calculated by the expression:

$$\alpha_a(532,z) = EB_c(z,t)\,\beta_a(532,z) \quad (9)$$

where $BE_c(z,t)$ are the backscattering to extinction conversion coefficients from $\lambda$ = 694 nm to $\lambda$ = 532 nm also derived for the Mt Pinatubo (Jäger and Deshler, 2003).

Finally we derive the aerosols scattering corrected by the total two-way transmittance ($\alpha_a^{Ta}(532,z)$) applying the correction

by the two-way aerosol transmittance $T_a^2(z)$

$$\alpha_a^{Ta}(532,z) = \frac{\alpha_a(532,z)}{T_a^2(532,z)} \quad (10)$$

Because the information available to calculate the $T_a^2(532,z)$ should be determined from total aerosol optical depth (TAOD) measurements from sun photometers we applied a two-step procedure. The first step consists of using the TAOD to calculate a first guess $T_a^2(532,z)$ followed by the calculation of a first guess $\alpha_a^{Ta}(532,z)_*$ profile. Then the stratospheric

AOD (sAOD) is calculated integrating $\alpha_a^{Ta}(532,z)_*$ between 12 and 24 km. The second step calculates:

$$tAOD = TAOD - sAOD \quad (11)$$

and the calculation of $T_a^2(532,z)$ is repeated but using tAOD instead of TAOD in equation (11) getting the definitive values of $\alpha_a^{Ta}(532,z)$.

The algorithms for the solution of the single wavelength lidar equations apply the two-way transmittance correction to the

raw lidar returned signal, together with squared distance correction, well before the backscattering ratio is calculated. In our case the available information we have are the backscattering ratios which have been derived without conducting the two way transmittance correction (**G-66**). That is the reason that correction was included in the retrieval of $\beta_a(694,z)$ in equation (7). However only the molecular and ozone two way transmittance corrections ( $T_m^2(694,z)$, $T_{O3}^2(694,z)$) were included in this step.

The aerosol two-way transmittance correction, $T_a^2(532,z)$, was delayed until the final step to derive $\alpha_a^{Ta}(532,z)$. The reason was that the available information on the AOD was at $\lambda$ = 500 nm and it was the total AOD, including the sAOD that we are trying to retrieve. No Ångstrom exponent contemporary information for the Agung eruption in the eastern US in the range 500 to 694 nm was found. Using the Ångstrom exponent climatological values covering the cited wavelength range from 1995 to 2019 from the nearest Aerosol Robotic Network (AERONET, 2020) stations we derived, we converted

the total AOD at 500 nm to 532 nm only, because it was a very near $\lambda$. It was the solution to lack of trusted information with the aim to minimize the error that could be introduced for converting the AOD to 694 nm. There have been abundant accounts about the changes of the physical-chemical properties of aerosols in the eastern US from the sixties until the present (Went, 1960; Husar et al., 1991).

*2.5 Complementary datasets used:*

The correction for the attenuation of the lidar signal by the two-way transmission is often considered negligible and ignored, based on signal to noise ratio considerations or for simplicity as it was the case in the original processing of these set of measurements (**G-66**; **GF-67**). We were motivated to make that correction by the fact that, during a little more than





half a century, the accuracies of the different instruments available for measurements of the stratospheric aerosols from the 1963 Mt. Agung eruption have been under, still unsettled, scrutiny and discussions (ex., Deirmendjian, 1965; Dyer, 1971a; Clemesha, 1971; Dyer, 1971b; Deirmendjian, 1971; Stothers, 2001; Timreck et al., 2018). Our goal was to produce the most accurate aerosol extinction and optical depth from the rescued measurements, based on the contemporary state of the art measurements in the sixties of the XX century.

Table 2 summarize the locations of the sites where radiosonde, ozone soundings and atmospheric turbidity measurements were conducted. Also the distances from each individual site to the corresponding lidar site are provided. Following each individual dataset is described.

**Table 2: Locations of the observation sites where thermodynamic variables and ozone vertical soundings were meassured nearby College, AK and Lexington, MA. Also the site of the atmospheric turbidity meassurements is**
**listed. The last column list the distances to Lexington (\*) and to Fairbanks(\*\*).**

| Station | Variable | Latitude | Longitude | Elevation | Dist. |
|---|---|---|---|---|---|
| **Nantucket, MA\*** | Temp, Pr | 41.2°N | 70.0°W | 14 m | 162.1 km |
| **Bedford, MA\*** | $O_3$ | 42.5°N | 71.3°W | 251 m | 10.7 km |
| **Blue Hill Obs., MA\*** | TAOD 500 nm | 42.2°N | 71.1°W | 192 m | 24.1 km |
| **CARTEL, Canada\*** | TAOD 500 & 675 nm | 45.38°N | 71.93°W | 251m | 334.4 km |
| **Fairbanks, AK \*\*** | $O_3$, Temp, Pr, | 64.8°N | 147.9°W | 353 m | 11.7 km |
| **Fairbanks, AK\*\*** | TAOD 500 nm | 64.86°N | 147.85°W | 133 m | 9.8 km |
| **Bonanza Creek, AK\*\*** | TAOD 500 & 675 nm | 64.74°N | 148.32°W | 353m | 23.6 km |

*2.5.1 Datasets used to estimate the thermodynamic local variables:*

To derive $\beta_m(\lambda, z)$, $\alpha_m(\lambda, z)$, $\alpha_a(\lambda, z)$ and $T_m^2(z)$ the required variables are the temperature (Temp(z)), pressure (Pr(z)) and
molecular number density ($N_d(z)$). We used the vertical profiles of Temp(z), Pr(z) and $N_d(z)$ from the 1962 US Standard Atmosphere (U. S. Standard Atmosphere, 1962).

We also made use of the Temp(z) and Pr(z) profiles (deriving $N_d(z)$) from the most complete and nearest sounding station. In the case of the soundings we took into account the fact that lidar observations were performed at night, typically near 21:00 EST (**G-66**; **GF-67**), then we selected Temp(z) and Pr(z) profiles from nearby soundings stations conducted at 00
GMT from the Integrated Global Radiosonde Archive (IGRA) Version 2 database (Dumre et al., 2016). **G-66** and **GF-67** mention the contemporary Bedford, MA, soundings near Lexington but although a total of 731 temperature profiles from this site are available in digital format (IGRA – 2, 2020) they only cover 1943 to 1945. The information about the temperature profiles from the ozone soundings from 1963 to 1964 exists, but it is plotted on the ozonogram reports (Hering and Borden, 1965).

For Lexington, Massachusetts (42° 25'N, 71° 15'W) we used the soundings from the station at Nantucket (41° 15' N, 70° 4' W, 14 m asl), code USM00072506, located at 163 km and around 1° of latitude south. Because the altitude of interest is between 12 km and 24 km, the upper troposphere and lower stratosphere in this region, they could be considered more representative than the US 1962 Standard Atmosphere. For Fairbanks, Alaska (64° 53'N, 148°3'W) we used Fairbanks (64° 49'N, 147° 53'W, 134 m asl), code USM00070261, at a distance of 11 km. They are the nearest sounding stations in
IGRA-2 with Pr(z) and Temp(z) profiles during 1964 and 1965. We used the soundings conducted the next day at 00 GMT because the local time at Nantucket, MA and Fairbanks, AK are -4 and -8 hours respect to GMT. At Nantucket, no sounding was available the next day at 00 GMT in 2 of the 66 measurements days. In those 2 cases, the same day 12 GMT sounding were used. The few gaps in Temp(z) and Pr(z) in both sets of data below 25 km were filled with the mean Pr(z) and Temp(z) profiles derived from all 1964 and 1965 Nantucket and Fairbanks daily Temp(z) and Pr(z) profiles at 00Z.




2.5.2 *Datasets used for the estimation of the ozone 2-way transmittance:*

We used the $N_{O_3}(z)$ from the 1966 US Standard Atmosphere Supplement (COESA, 1967). In addition, we used the seasonal means of $N_{O_3}(z)$ between 1963 and 1967 from ozone soundings conducted at L. G. Hanscom Fla., Bedford, MA and Fairbanks, AK for Lexington and Fairbanks respectively (Hering and Borden, 1967).

The profile of the ozone absorption coefficient at a given wavelength ($k_{O_3}(\lambda, z)$) is calculated using the profile of ozone cross sections ($\sigma_{O_3}(\lambda, \text{Temp}(z))$):

$$k_{O_3}(\lambda, z) = \sigma_{O_3}(\lambda, \text{Temp}(z)) \times N_{O_3}(z) \qquad (12)$$

at the temperature ($\text{Temp}(z)$), where $N_{O_3}(z)$ is the number density of ozone. The $\sigma_{O_3}(\lambda, \text{Temp}(z))$ at $\lambda$ = 694 nm is provided by Serdyuchenko et al., (2014) in the temperature range 193 to 293 ºK. We used the average of $\sigma_{O_3}(\lambda, \text{Temp}(z))$, 9.88e-22 $cm^2$ molecules$^{-1}$ , considering that the standard deviation of this averaging profile represents 2.4 % variability of the average value. This set of absorption coefficients have been recommended by the recent status report from the International Ozone Commission from WMO (Orphal et al., 2016).

2.5.3 *Datasets used for the estimation of the tropospheric aerosol 2-way transmittance:*

For Lexington we found contemporary statistics of turbidity measurements (**B**) at $\lambda$ = 500 nm. Those measurements were part of the turbidity network established in 1961 by the U. S. Weather Bureau Research Station (later National Center for Air Pollution Control), Cincinnati, Ohio. For Lexington, the measurements were conducted at Blue Hill Observatory, Boston, Ma, 24 km from the lidar location. The Blue Hill Observatory frame plot with the monthly means for the period 1961 to 1966 are in figure 3 in Flowers et al., (1969). The curve of the monthly mean **B** belonging to Blue Hill Observatory in the frame plot on the figure were digitized (WebPlotDigitize, 2020). Then TAOD at $\lambda$ = 500 nm was calculated using the equation (Volz, 1969), resulting from converting the decadic logarithm used to define **B** to the neperian logarithm used for AOD:

**B** = 0.434 x TAOD        (13)

For Fairbanks we did not find contemporary measurements, but there were manually conducted measurements in several places in the Arctic and Antarctic, including at Fairbanks (64.86°N, 147.85 °W, 133 m), with sun photometers at several wavelengths (Shaw, 1982). Those measurements are reported to be corrected by the molecular scattering and gas absorption (Shaw et al., 1973). The instruments were calibrated at Mauna Loa Observatory using Langley method with root mean square errors (RMSE) of sun photometer voltage output readings (V) of $\frac{\delta v}{v} \approx 10^{-3}$ having a systematic RMSE for AOD = +/- 0.002 and the total error estimated as +/- 0.004 (Shaw, 1982). At Fairbanks the annual mean AOD = 0.110 from 105 observations at $\lambda$ = 500 nm is reported on table 1 of the cited reference, but also the AOD annual cycle appears in the lower panel of figure 2, showing the high AOD values in late winter and spring, peaking up to 0.135. We then digitized the mean AOD and its variation range values for July (no data for August appears on the figure), resulting AOD = 0.082 ± 0.022. Although Shaw (1982) does not provide the information of the year the measurements were conducted this data is cited and cited to have been conducted in 1978 by Freund (1983).

We also used AOD data at 500 nm from the 2 nearest AERONET stations to each site having long-term records. Bonanza Creek, AK, is less than 30 km from the location the lidar measurements were reported to be conducted at Fairbanks. TAOD measurements at this site have been conducted from 1997 to the present. The other site belongs to the Centre d'Applications et de Recherches en Teledetection (CARTEL), at the Universite de Sherbrooke, Canada, 334 km from Lexington.

From both sites we also used the climatological monthly means of TAOD at 500 and 675 nm from 1997 to 2019 for Bonanza Creek, and 1995 to 2019 for CARTEL (AERONET, 2020). We then derived the TAOD at 532 nm using the



Ångstrom exponents calculated from the TAOD climatological monthly means for the interval 440 to 870 nm ,that we used

for Lexington. In the case of Bonanza Creek, we had one "contemporary" value from July 1978 (Shaw, 1982), and we selected the July climatological monthly mean, as the "current" value. After the conversion to 532 nm they were respectively 0.087 and 0.242, and we used the same value for both July and August 1964 lidar measurements at Fairbanks.

In the case of Lexington, for comparison purposes, we digitized the average monthly mean TAOD for 26 Eastern US stations from 1972 to 1975 from Husar et al., (1981). The series of monthly mean TAOD values were converted from 500

nm to 532 nm, with the procedure described above, using the Ångstrom exponents for the interval 440 to 870 nm. The figure 1 shows the resulting values. The high TAOD values for the two series before the 1960s and 1970s are representative of what have been reported for the Eastern US (Husar et al., 1991). The natural conditions contribute to an elevated background AOD attributed to the combination of absolute and relative humidity, and the vegetation density, both of which could be responsible for increased natural aerosols either from hygroscopic marine aerosols or from secondary

aerosols originating from vegetation (e.g. Went, 1960). Additionally an important contribution to the TAOD came from anthropogenic aerosols originating from the extensive use of fossil fuels in the region. In fact, as their figure 1 shows, the pollution increased with respect to the 1960s (Husar et al., 1991). Recent research reports aerosol simulations for the historical period from 1850 to 2014 using the GISS ModelE2.1 with two different aerosol schemes to contribute to the Coupled Model Intercomparison Project Phase 6 (CMIP6). The simulations result in the AOD showing an increasing trend

from well before 1900 until the early 1970s in the Eastern US, supporting the AOD data features we collected from Blue Hills Observatory (Bauer et al., 2020).

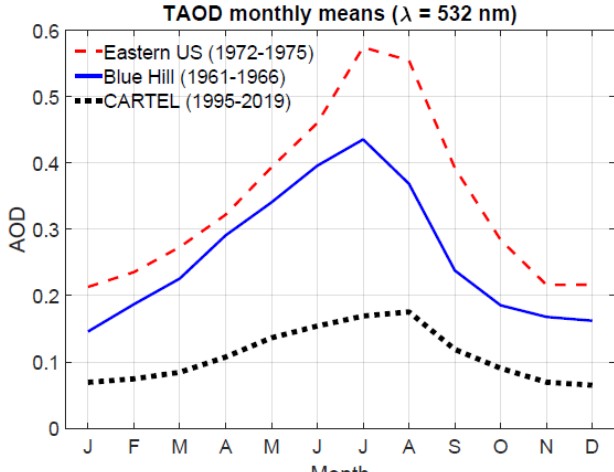

**Figure 1: Contemporary and current TAOD monthly means at 532 nm from Blue Hill Observatory, MA, from 1961 to 1966, the average of 26 stations in Eastern US between 1972 and 1975 and CARTEL, Canada, from 1995 to 2019.**


*2.6 Numerical and statistical methods:*

For each of the two datasets we calculated the statistical results of the differences ($\Delta\alpha_{*US}$) between $\alpha_a(532, z)_{US}$ calculated using the same $\beta_m(594, z)$ profile from the 1962 US Standard Atmosphere for all the days and the $\alpha_a(532, z)_*$ calculated

using the $\beta_m(594, z)$ profiles derived from the daily soundings:

$$\Delta\alpha_{a*} = \alpha_a(532, z)_{US} - \alpha_a(532, z)_* \qquad (14)$$

and the percent differences $\Delta\alpha\%_{a*}$ by the expression:

$$\Delta\alpha_{a*}\% = \left[\frac{\alpha_a(532,z)_{US} - \alpha_a(532,z)_*}{\alpha_a(532,z)_{US}}\right] \times 100 \qquad (15)$$



Similarly we defined the differences $\Delta\alpha_{at2w}$ and the percent differences $\Delta\alpha_{at2w}\%$ between the $\alpha_a(532,z)_*$ calculated using
the $\beta_m(594,z)$ profiles derived from the daily soundings, and its corrected values $\alpha_a(532,z)_{t2w}$ resulting for accounting
for the two-way atmospheric transmittance.

Also we defined, for cumulative aerosol optical depth in the layer 12 to 24 km, $\tau_a(532,)_*$ and $\tau_a(532,z)_{US}$ calculated from
the $\alpha_a(532,z)_*$ and $\alpha_a(532,z)_{US}$ respectively:

$$\Delta\tau_{a*} = \tau_a(532,z)_{US} - \tau_a(532,z)_* \qquad (16)$$

and the percent differences $\Delta\tau\%_{a*}$ by the expression:

$$\Delta\tau_{a*}\% = \left[\frac{\tau_a(532,z)_{US} - \tau_a(532,z)_*}{\tau_a(532,z)_{US}}\right] \times 100 \qquad (17)$$

*2.7 Relative Error estimates:*

The present evaluation of the relative errors in the different processing steps of the single wavelength elastic lidar followed
the algorithms developed by Russell (1979). Whenever it was possible we calculated the different terms of the equation
based in the available dataset error. In several cases we combined information from the rescued metadata associated with
the measurements and from available additional information in literature.

*2.7.1 Backscattering ratio relative error:*

As it was explained above, the data we rescue are a reasonable approximation of what we today know as the backscattering
ratio described in equation (2). Then we use the equation (19) from Russell (1979) quantifying the contributions from the
different sources to the relative error in backscattering ratio $\frac{\delta SR}{SR}$:

$$\left(\frac{\delta SR}{SR}\right)^2 = \left(\frac{\delta N_s}{N_s}\right)^2 + \left(\frac{\delta T_{2w}}{T_{2w}}\right)^2 + \left(\frac{\delta\beta_m}{\beta_m}\right)^2 + \left(\frac{\delta\beta_{m*}}{\beta_{m*}}\right) - \left(\frac{C_{FF*}^2}{\beta_m\beta_{m*}}\right) + \left(\frac{\delta SR_{min}}{SR_{min}}\right)^2 \qquad (19)$$

Where $SR(\lambda,z)$ is the total backscattering ratio; $N_s$ is the signal measured; $T_{2w}$ the two-way transmittance from aerosols;
molecules and ozone; $\beta_m$ the molecular backscattering; $\beta_{m*}$ molecular backscatter at the normalization level; $SR(\lambda,z)$ total
backscattering ratio at the normalization level and $C_{FF*}^2$ the covariance between measured $\beta_m$ and $\beta_{m*}$.

For estimating the magnitude of the signal measurement error we rely on the information provided by **G-66**. He estimated
statistical fluctuation of the signal, the shot noise of the photodetector and other sources on the order of 0.2 to 3%. Then
both for Lexington and Fairbanks: $\left(\frac{\delta N_s}{N_s}\right) = 3\%$

As cited above, according to **G-66** if no $T_{2w}$ correction was conducted then the term $\left(\frac{\delta T_{2w}}{T_{2w}}\right)^2 = 0$.

Because in the calculation of $SR(\lambda,z)$ values of $N_d(z)$ from the 1962 US Standard Atmosphere were used (**G-66**) it was
assumed $\frac{\delta\beta_m(\lambda,Z)}{\beta_m(\lambda,z)} = 3\%$ for both sites (Russell et al, 1979). In addition we assumed $\left(\frac{\delta\beta_m}{\beta_m}\right) = \left(\frac{\delta\beta_{m*}}{\beta_{m*}}\right)$, and $C_{FF*}^2 = 0$ after
assuming measurement errors are uncorrelated. It is a plausible assumption because the profile $\beta_m$ used the US 1962
Standard Atmosphere for the vertical resolution of the lidar.

The term $\delta SR_{min}$ was evaluated according to table (1b) in Russell (1979) for the $SR_{min} = 1.01$ and the respective latitudes
of both sites. Then for both sites, according to Russell et al. (1979).

$$\delta SR_{min} = 0.07(SR_{max} - 1) \qquad (20)$$


*2.7.2 Aerosol backscattering relative errors:*

The equation (18) in Russell (1979) to estimate the relative error in $\beta_a(694,z)$ can be approximated in our case by





$$\left(\frac{\delta\beta_a(694,z)}{\beta_a(694,z)}\right)^2 = \left(\frac{\beta_m}{\beta_a}\right)^2 \left\{(SR)^2 \left[\left(\frac{\delta SR}{SR}\right)^2 + \left(\frac{\delta T_{2w}}{T_{2w}}\right)^2\right] + \left(\frac{\delta\beta_m}{\beta_m}\right)^2\right\} \qquad (21)$$

The estimated error for the 2-way transmission corrections in Russell et al. (1979) provides the expression:

$$\left(\frac{\delta T_{2w}}{T_{2w}}\right)^2 = 4\{[\delta\tau_a(\lambda,z)]^2 + [\delta\tau_m(\lambda,z)]^2 + [\delta\tau_{O3}(\lambda,z)]^2\} \qquad (22)$$

and considering the standard error of determinations of $\tau_a$, $\tau_{O3}$, and $\tau_m$ are respectively 50, 20 and 10% the following estimates are produced. That is: $\delta\tau_a = 0.5\,\tau_a$, $\delta\tau_{O3} = 0.2\,\tau_{O3}$ and $\delta\tau_m = 0.1\,\tau_m$. However, in our calculus chain of $\beta_a$ only the ozone and molecular two-way transmittances were used.

For this section of the procedure we considered $\left(\frac{\delta\beta_m}{\beta_m}\right) = 10\%$ because we used radiosonde soundings at both sites (Russell et al, 1979). We neglected the error in computing Qs using equation (5) because its maximum relative error is 0.2 % for a spectral region of 350-1600 nm (Hosteler et al., 2006), well below the error in $\left(\frac{\delta\beta_m}{\beta_m}\right)$.

Next we determined the relative error in $\beta_a(532,z)$ associated with the conversion from $\beta_a(694,z)$ in equation (7), using the wavelength exponents (kb(z, t)) for aerosol backscatter in the range of wavelengths between 694 and 532 nm (Jäger and

Deshler, 2003). The errors were estimated from their figure 1 with $\left(\frac{\delta kb}{kb}\right)^2 = 10$ %:

$$\left(\frac{\delta\beta_a(532,z)}{\beta_a(532,z)}\right)^2 = \left(\frac{\delta\beta_a(694,z)}{\beta_a(694,z)}\right)^2 + \left(\frac{\delta kb}{kb}\right)^2 \qquad (23)$$

*2.7.3 Aerosol extinction relative errors:*

In the case of the $\alpha_a$, its relative errors are:

$$\left(\frac{\delta\alpha_a}{\alpha_a}\right)^2 = \left(\frac{\delta\beta_a}{\beta_a}\right)^2 + \left(\frac{\delta EB_c}{EB_c}\right)^2 \qquad (24)$$

The last term in the right side represents the error in the $EB_C$ for $\lambda = 532$ nm. In the case of the ones we used (Jäger and Deshler, 2002; 2003) the error has been estimated in ± 40 % according to Deshler et al., (2003). For $\alpha_a^{Ta}$, the aerosols extinction corrected by the aerosols two-way aerosols transmittance, using the estimates of its relative error described

above:

$$\left(\frac{\delta\alpha_a^{Ta}}{\alpha_a^{Ta}}\right)^2 = \left(\frac{\delta\alpha_a}{\alpha_a}\right)^2 + \left(\frac{\delta T_{2wa}}{T_{2wa}}\right)^2 \qquad (25)$$

Using the cited set of equations and the assumptions described above we evaluated the error for each level in each measurement.

**3.0 Results:**

*3.1 Aerosols extinction cross sections and optical depth:*

Figure 2 shows the $\alpha_a(532,z)$ cross-sections for Lexington. Panel a) $\alpha_a(532,z)_{US}$ is calculated using the same $\beta_m(594,z)$

profile from the 1962 US Standard Atmosphere for all the days; b) $\alpha_a(532,z)_*$ was calculated using the daily $\beta_m(594,z)$ profiles derived from the sounding at Nantucket, MA. On top of the figures we plotted the dates the measurements were conducted (red starts at 24.5 km level). In the case of Lexington the two data gaps higher than 1 month, March and July to September both in 1964 have been left blank in the cross-sections plots. The temporal/vertical cross-section of the aerosols extinctions were generated using a linear time interpolation.

In general, the cross-sections show a high level of variability of the aerosol extinction for Lexington both in time and altitude associated with the complex thermodynamic processes in the upper troposphere-lower stratosphere of the eastern US. Three main maximums are identified across the entire period. The first between 16 and 18 km at the beginning of the

record in middle January 1964. The second between 14 and 16 km by November 1964 and the third at the same altitude but

in the transition between March and April 1965. Evident is the decaying altitude of the maximums in time typical of the

volcanic aerosols clouds in the lower stratosphere. However the occurrence of the absolute maximum at this time cannot be

attributed to the volcanic aerosols from Mt Agung, it will be discussed below. No long term analysis of this type could be

conducted on figure 3 for Fairbanks because of the very short period of time it covers. However, the cross-section of

$\alpha_a(532, z)_*$ for Fairbanks reveals the maximum values between 14 and 16 km with the absolute maximum around mid-

August, centred at 15 km.

Regarding the magnitudes of $\alpha_a(532, z)_{US}$ in figure 2, they are slightly higher than the ones from $\alpha_a(532, z)_*$. That is also

the case in figure 3 showing the cross-sections for Fairbanks, with panels similar to figure 2. This is quantified in table 3.

At both sites the mean and maximum values for $\Delta\tau_{a*}$ and $\Delta\alpha_{a*}$ are positive showing that the magnitudes of $\alpha_{a_{US}}$ and $\tau_{a_{US}}$

are in general higher than $\alpha_{a_*}$ and $\tau_{a*}$. Also in the table we appreciate that the magnitudes of the mean percent difference

increase of both variables is around 1%.

The fact described above disagrees with the possibility **G-66** mentions about lower aerosol backscatter from the retrieval

they conducted, using the 1962 US Standard Atmosphere, and the more realistic ones using soundings. He arrived to that

conclusion from "a cursory examination" of the local variations of molecular number density ($N_d(z)$) estimated with the

Temp(z) profiles from ozone soundings at Bedford, MA (Hering and Borden, 1965). He reported $N_d(z)$ variability rarely

exceeded 5% of the $N_{dUS}(z)$ values at altitudes between 10 and 30 km.

**Table 3: Relative differences between the $\alpha_{a_{US}}$ and $\alpha_{a_*}$ as well as $\tau_{a_{US}}$ and $\tau_{a*}$**

|  | Lexington | | | | Fairbanks | | | |
|---|---|---|---|---|---|---|---|---|
|  | $\Delta\alpha_{a*}$ | $\Delta\alpha\%_{a*}$ | $\Delta\tau_{a*}$ | $\Delta\tau_{a*}\%$ | $\Delta\alpha_{a*}$ | $\Delta\alpha\%_{a*}$ | $\Delta\tau_{a*}$ | $\Delta\tau_{a*}\%$ |
| **Mean** | 1.89E-05 | 1.4 | 2.46E-04 | 1.2 | 1.42E-05 | 0.2 | 1.84E-04 | 1.6 |
| **|Mean|** | 5.92E-05 | 3.2 | 7.42E-04 | 3.3 | 1.85E-05 | 2.1 | 1.90E-04 | 1.7 |
| **Max** | 4.22E-04 | 42.2 | 2.71E-03 | 13.6 | 1.13E-04 | 6.4 | 4.30E-04 | 3.1 |

To estimate the effects of the differences between the magnitudes of $N_{dUS}(z)$ and $N_d(z)$ in the backscattering ratios we

calculate the differences between the ratios defined by:

$$\Delta N_d(z) = \frac{N_{dUS}(z)}{M_{dUS}} - \frac{N_d(z)}{M_d} \qquad (18)$$

The values in the denominators $M_{dUS}$ and $M_d$ are the mean values of $N_{dUS}(z)$ and $N_d(z)$ between 25 and 30 km

respectively, replicating the procedure used by **G-66**. In figure 4 the differences $\Delta N_d(z)$ for all the 66 soundings at

Nantucket used to calculate $N_d(z)$ and the 9 for Fairbanks are plotted. For Lexington, on panel a), $N_{dUS}(z)$ values are both

negative and positive, but higher values of $N_{dUS}(z)$ dominate. It is confirmed that the relative means and the maximum

values of $\Delta\alpha_{a*}$ between the $\alpha_a(532, z)_{US}$ and $\alpha_a(532, z)_*$ for Lexington in table 3 are the same order of magnitude, $10^5$

km[-1] for the relative and absolute means and $10^{-4}$ km-1 for the maximum, larger than for Fairbanks. The values of the

relative means $\Delta\alpha_{a*}\%$ confirm the higher values when the 1992 US Standard Atmosphere is used, in contradiction with **G-66** estimation.

It has been stablished that the errors in lidar retrievals of $\alpha_a(532, z)_*$ attributed to the use of Temp(z) and Pr(z) from a

model atmosphere to retrieve $N_d(z)$ are of the order of 3% and decrease to 1% when the source of Temp(z) and Pr(z) are

soundings (Russell et al., 1979). Again in table 3, the magnitudes of the absolute differences between the US 1962 Standard

Atmosphere and the soundings at Lexington and Fairbanks for $\alpha_a(532, z)$ are in the order of 3%. That magnitude matches

the error attributed if case models are used instead of soundings to derive $\beta_m(\lambda, z)$.

The figure 5 shows $\tau_{a*}$ both for Lexington (blue stars) and for Fairbanks (red diamonds). Also figure 5 shows the monthly

mean $\tau_a$ for the northen hemisphere (Sato et al., 1993). The means for the entire period of measurements available at each



site are 0.0215 and 0.0099 respectively. The magnitude of the mean $\tau_{a*}$ at Fairbanks are half that of Lexington, providing evidence of the decreasing aerosol amount with increasing latitude. At the same time, some of the daily $\tau_{a*}$ values at Lexington are around the magnitude of the mean $\tau_{a*}$ at Fairbanks, because of the variability of $\alpha_a(532, z)_*$. Few $\tau_{a*}$ values from Lexignton have magnitudes near the values of Sato $\tau_a$, the current reference for this period. However, as we will see in the next section a better agreement is found when the measurements are corrected by two way transmittance attenuation.

Taking into account the little difference between the results using the US 1962 Standard Atmosphere or the soundings to derive $\beta_m(\lambda, z)$, the first simpler option could reliably be used. However we decided to use the soundings to minimize the errors and to capture the more realistic features of the aerosol cloud.

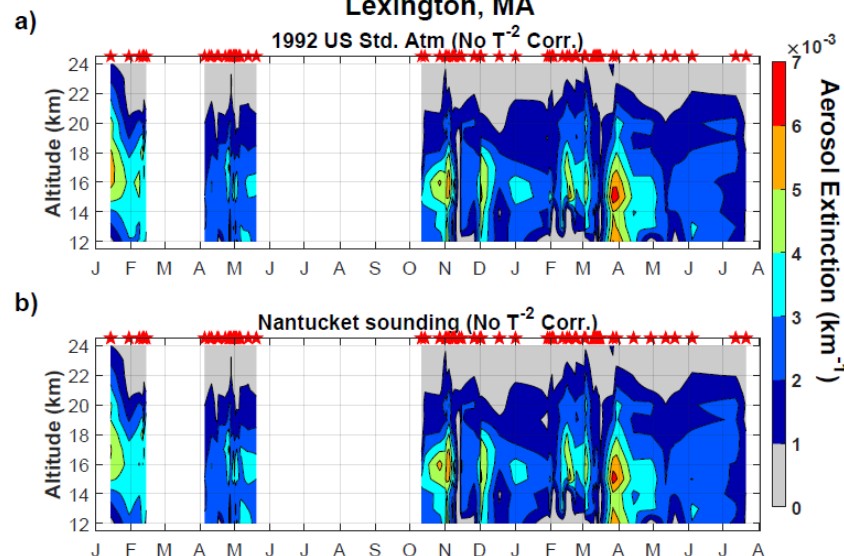

**Figure 2: Panel a) $\alpha_a(532, z)$ calculated using the same $\beta_m(594, z)$ profile from the 1962 US Standard Atmosphere**
**for all the days; b) $\alpha_a(532, z)$ was calculated using the daily $\beta_m(594, z)$ profiles from the sounding at Nantucket, MA.**

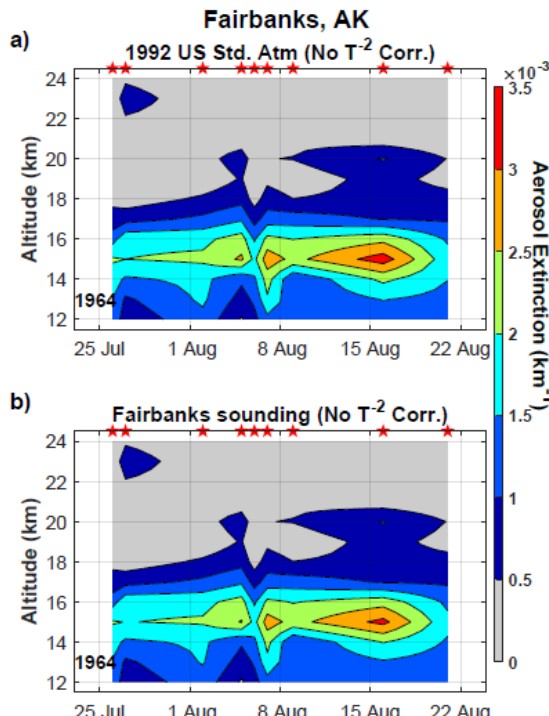

**Figure 3: Idem figure 2, but for Fairbanks, AK.**

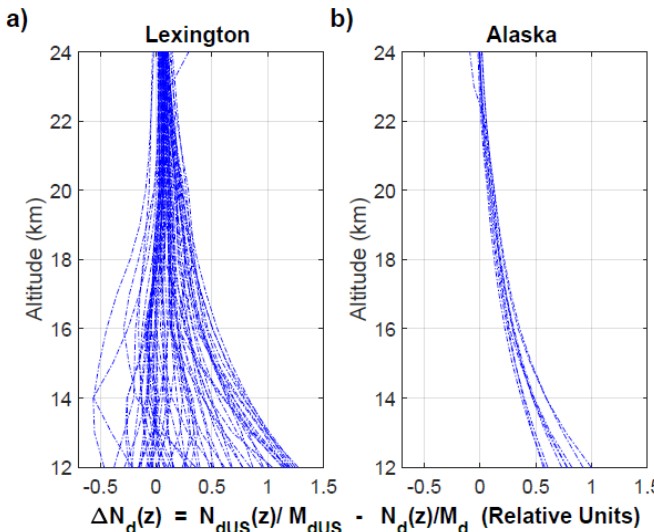


**Figure 4: Differences between the number molecular density (Nd (z)) from soundings and from the 1962 US Standard Atmosphere in the region from 12 to 24 km. Panel a) Represents $N_d$ (z) from Nantucket soundings used for Lexington and b) $N_d$ (z) from Fairbanks.**

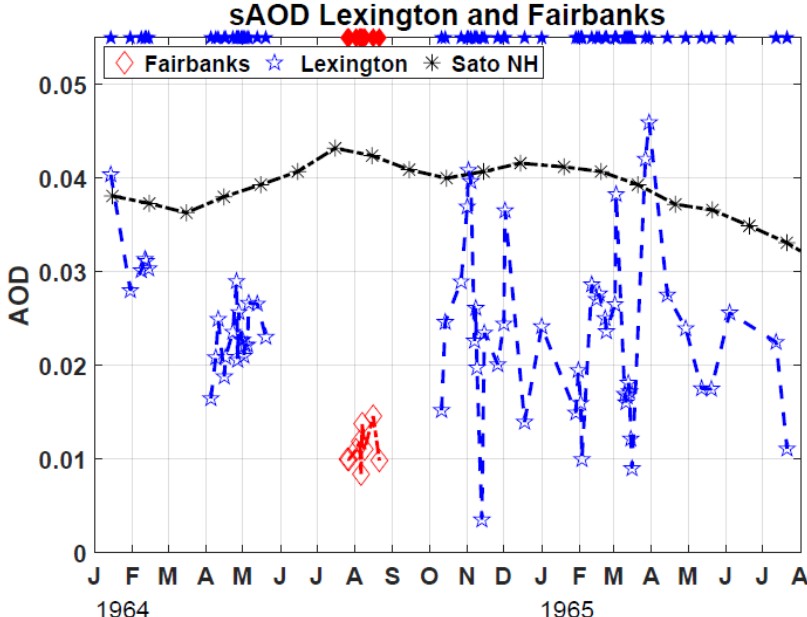


**Figure 5: Daily aerosol optical depth ($\tau_{a*}$) for Lexington (blue stars), Fairbanks (red diamonds) and for the northern hemisphere (black asterisks) for the period the measurements were conducted. $\tau_{a*}$ was calculated from the $\alpha_a(532, z)_*$ derived using local soundings. Blue stars and red diamonds on the top axes of the figure are the dates the measurements were conducted.**


*3.2 Aerosols extinction cross-sections and optical depth corrected by aerosol two-way transmittance attenuation:*

Figure 6 shows the cross-sections of $\alpha_a(532, z)_*$ for uncorrected and corrected two-way transmittance ($\alpha_a^{Ta}(532, z)$) for
Lexington. The initial values of TAOD were used to obtain a first estimate of $\alpha_a(532, z)_{*t2w}$. This $\alpha_a^{Ta}(532, z)_*$ is only used to calcuate sAOD for each day and is subtracted from TAOD to produce the tropospheric corrected value (tAOD) and the calculation is repeated to determine new profiles of the two-way aerosol transmittance and correct $\alpha_a(532, z)_*$ generating the $\alpha_a^{Ta}(532, z)$. Panel a) shows the cross-section of uncorrected values of $\alpha_a(532, z)_*$, in panel b) the cross-sections of $\alpha_a^{Ta}(532, z)$. The magnitudes of $\alpha_a^{Ta}(532, z)$ are higher than $\alpha_a(532, z)_*$. The two-way transmittance correction
is dominated by the aerosols, in particular the tropospheric aerosols. The maximum extinction is at the third maximum, 1.071 x 10$^{-2}$ km$^{-1}$ located at 15 km, on March 27th 1965. Similarly in figure 7, the Fairbanks cross-sections for $\alpha_a(532, z)_*$ and $\alpha_a^{Ta}(532, z)$ show a notable difference in magnitude. The absolute maximum extinction occurred on August 16th 1964 at 15 km, with a magnitude of 3.8 x 10$^{-3}$ km$^{-1}$.

Table 4 contains the relative and absolute means and maximums for $\Delta\alpha_a^{Ta}$, $\Delta\alpha_a^{Ta}\%$, $\Delta\tau_a^{Ta}$ and $\Delta\tau_a^{Ta}\%$ calculated using
equations (14) to (17) respectively but for $\alpha_a(532, z)_*$ vs $\alpha_a^{Ta}(532, z)$ and $\tau_{a*}$ vs $\tau_a^{Ta}$. The magnitude of $\Delta\alpha_a^{Ta}$ produced by the two-way transmittance correction is in the order of 10$^{-3}$ km$^{-1}$ for Lexington and 10$^{-4}$ km$^{-1}$ for Fairbanks. They represent an increase of 72 % in the first case and 26 % in the second. These increases are due mainly to the two-way aerosol transmittances, dominated by the tropospheric AOD with magnitudes more than twice as high at Lexington than at Fairbanks. The increase in magnitude reveals more details of the vertical distribution of the $\alpha_a^{Ta}(532, z)$ and in the case of



Lexington the presence of a 4[th] maximum during May 1964, who's vertical location matches the decreasing trend at the core of the stratospheric aerosols cloud.

In figures 8 and 9 the increases of $\tau_a^{Ta}$ with respect to $\tau_{a*}$ for Fairbanks and for Lexington respectively are shown: At Lexington the $\tau_a^{Ta}$ magnitudes are around the values of $\tau_a$ from Sato et al., (1993) for the northern hemisphere represented by the dot-dash black line, a confirmation that the results of the present study are in the accepted range of magnitudes for

$\tau_a$ from Agung at the northern hemisphere. Again in table 3 the magnitudes of the increase of $\tau_a^{Ta}$ are in the order of $10^{-2}$ for Lexington and $10^{-3}$ for Fairbanks, representing a 66 % and 26 % increases respectively.

At Lexington the absolute maximum value of $\tau_a^{Ta}$, 0.076 occurs on March 30[th] 1965, 3 days after the absolute maximum extinction was registered at 15 km. At Fairbanks the absolute maximum value of $\tau_a^{Ta}$, 0.018, was registered on August 16[th] 1964, the same day the absolute maximum extinction was registered at 15 km at this site.

**Table 4: Idem than table 3, but for the comparison of $\alpha_a(532,z)_*$ vs. $\alpha_a^{Ta}(532,z)$ and $\tau_{a*}$ vs $\tau_a^{Ta}$ See text for details.**

|  | Lexington | | | | Fairbanks | | | |
|---|---|---|---|---|---|---|---|---|
|  | $\Delta\alpha_a^{Ta}$ | $\Delta\alpha_a^{Ta}\%$ | $\Delta\tau_{at2w}$ | $\Delta\tau_{at2w}\%$ | $\Delta\alpha_a^{Ta}$ | $\Delta\alpha_a^{Ta}\%$ | $\Delta\tau_{at2w}$ | $\Delta\tau_{at2w}\%$ |
| **Mean** | 1.17E-03 | 67.2 | 1.52E-02 | 66.2 | 2.22E-04 | 26.5 | 2.89E-03 | 25.9 |
| **\|Mean\|** | 1.17E-03 | 67.2 | 1.52E-02 | 66.2 | 2.22E-04 | 26.5 | 2.89E-03 | 25.9 |
| **Max** | 3.60E-03 | 152.6 | 3.09E-02 | 148.8 | 8.35E-04 | 29.1 | 3.89E-03 | 26.7 |

During the course of more than two decades after the pioneering stratospheric aerosols measurements with lidar work by Fiocco and Grams (1964) multiple researchers contributed to the development of the processing algorithms to retrieve aerosols optical properties and its errors (Russell et al, 1979, Klett, 1981; Klett, 1985, Kovalev, 2015). Those facts explain

the limitations that do not allow the retrieval of the full set of optical variables characterizing the stratospheric aerosols from the Fiocco and Grams dataset. However using a Junge size-distribution model, and assuming Mie scattering with refractive index 1.5, they produced estimates of the aerosol content of the stratosphere at 16 km: number concentration, surface area and the aerosol density per unit volume of air. They also use the mean profile they derived to estimate the total of particles/cm$^3$, total surface area and a total mass, integrating the concentrations obtained between 12 and 24 km (**GF-67**).

The only available optical property estimates, based in some of the cited particle concentration estimates at 16 km and in the column are the aerosol extinction at 16 km, 2 x$10^{-3}$ km$^{-1}$ and the aerosol optical depth of 0.015, both at 694 nm (Deirmejian, 1971).

For comparing with the values reported above, we made estimates at of $\alpha_a(694,z)$ from $\alpha_a(532,z)$ as well as $\tau_a(694,z)$ from $\tau_a(532,z)$ using the wavelength exponents for aerosols from Mt Pinatubo in the range of wavelengths 532 to 694 nm

(Jäger and Deshler, 2002). We made the estimates both for Lexington and Fairbanks because no clear assignation of the values cited above to one of the two sites is made in **G-66** and **GF-67**. At the 16 km level the mean value of $\alpha_a(694,z)$ was $10^{-3}$ km$^{-1}$ for Fairbanks and 2 x $10^{-3}$ km$^{-1}$ for Lexington matching for both sites the order of magnitude estimated by Deirmejian, (1971).

An additional validation of those results, in particular for $\tau_a^{Ta}(532,z)$ at Lexington appears in figure 9, where the

stratospheric $\tau_a(532,z)$ for the northern hemisphere from January 1964 to July 1965 has been plotted (Sato et al., 1993). The magnitude of $\tau_a^{Ta}(532,z)$ is the same at Lexington (and also at Fairbanks, figure 8) as the $\tau_a(532,z)$ from Sato et al., (1993).

From 1963 to mid-1965, in addition to the 1963 Mt Agung, two other volcanoes were reported to have erupted in the northern hemisphere with Volcanic Explosivity Index (VEI) 3. They were the Trident volcano in Alaska at 58°N and

155°W and the Vestmannaeyjar volcano (also known as *Surtsey*) south of Iceland at around 63°N and 20°W and. The first was reported to have erupted in April 1963 and its plume reaching 15 km (Decker, 1967). The second remained in in eruption between November 1963 and February 1964, with its plume reaching more than once in November 1963 an altitude around 4.5 km above the tropopause, located approximately at 10.5 km (Thorairinsson, 1965). They were attributed


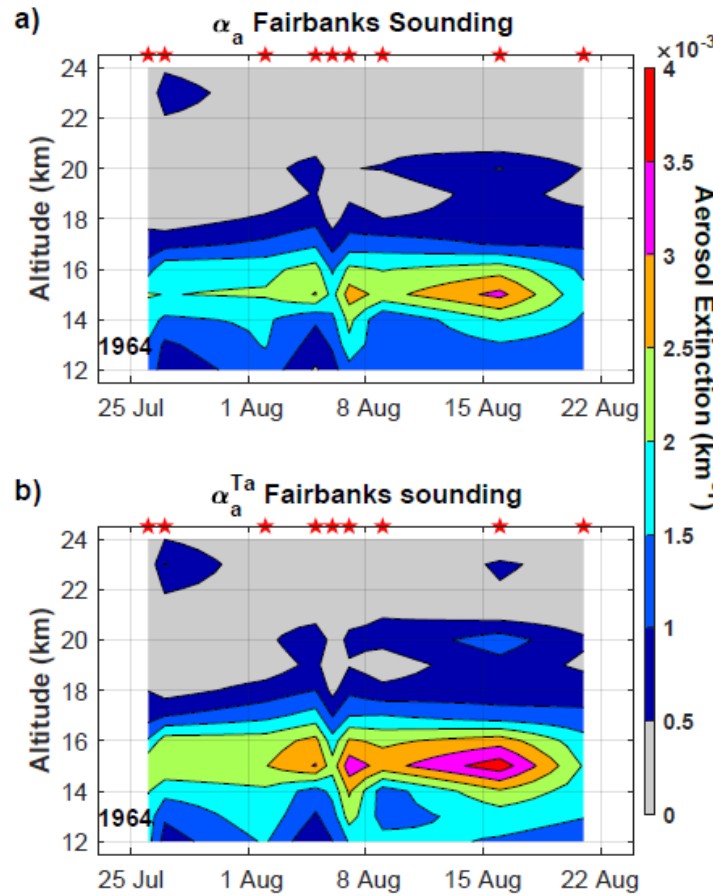

**Figure 7:** Cross sections of $\alpha_a(532, z)_*$ for uncorrected and corrected two-way transmittance ($\alpha_a^{Ta}(532, z)$) for Fairbanks.



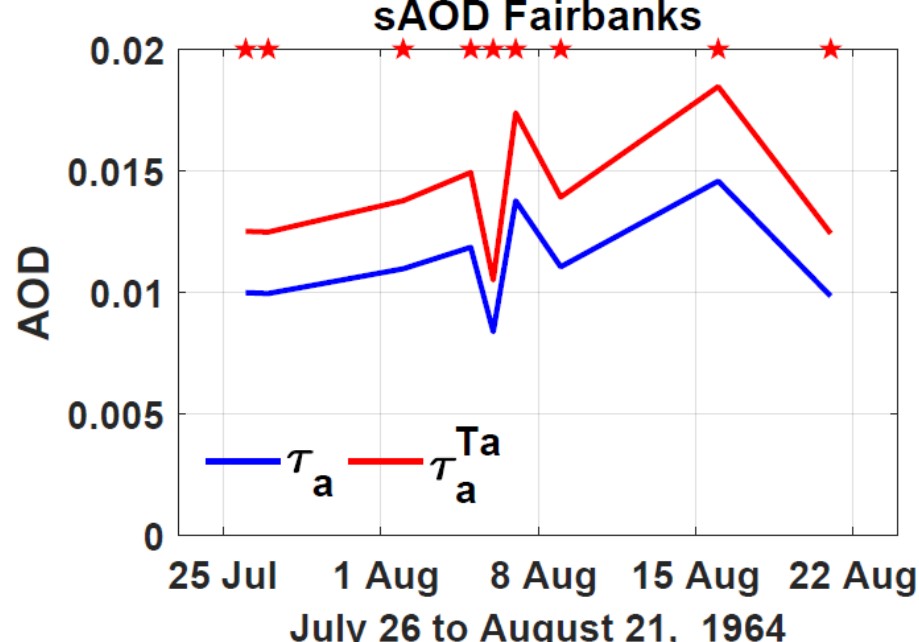

**Figure 8: Stratospheric AOD (sAOD) for Fairbank for** $\tau_a(532, z)$ **and** $\tau_a^{Ta}(532, z)$.

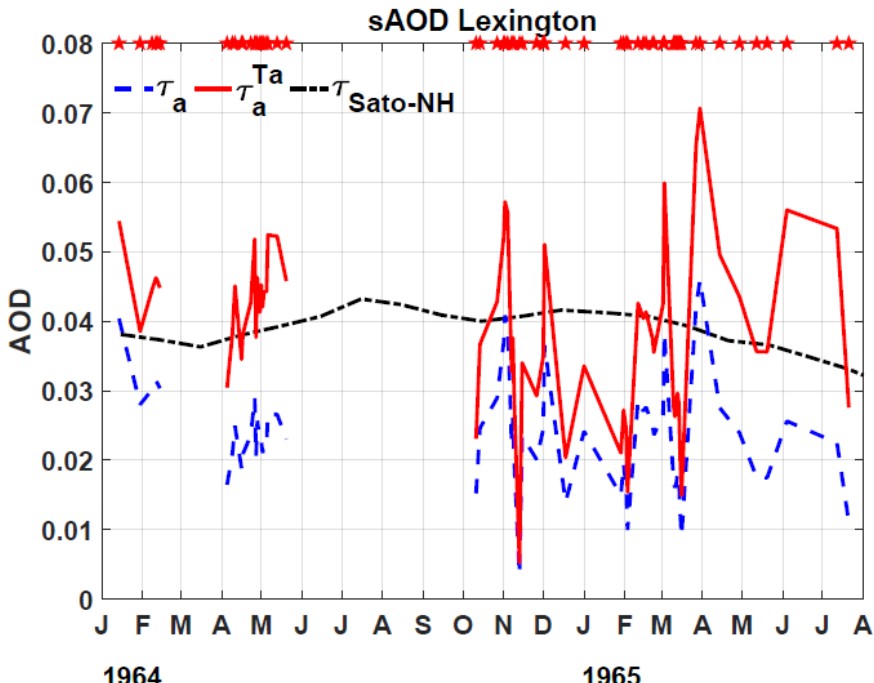

**Figure 9: Stratospheric AOD (sAOD) for Lexington for $\tau_a(532, z)$ and $\tau_a^{Ta}(532, z)$.**

*3.3 Relative Errors:*

Table 5 reports the results for the estimated relative errors in the aerosol extinction with and without the aerosol two-way transmittance correction for both sites. In addition, the relative errors of the backscattering ratio and aerosol backscatter at 694 nm and the aerosol backscatter at 532 nm are reported. The relative errors for $\alpha_a^{Ta} \leq 5 \times 10^{-4}$ km$^{-1}$ were excluded in

the statistics.

Note the increases in the mean relative errors from $\left(\frac{\delta SR}{SR}\right)$ to $\left(\frac{\delta \beta_a}{\beta_a}\right)$, 12 % to 48 % at Fairbanks and 13 % to 36 % at

Lexington, the higher increases occur during the full processing. It is explained by the factor $\left(\frac{\beta_m}{\beta_a}\right)^2$ in equation (21).

Because the processing algorithm relies on equation (6) to derive $\beta_a$ from $\beta_m$ the squared ratio will be lower than 1 if $SR_o$

< 2, increasing as $SR_o$ decreases and reaching the value $\left(\frac{\beta_m}{\beta_a}\right)^2 = 10^4$ for $SR_o = 1.01$. Only with $SR_o => 2$ is the ratio is

lower than 1, which in the case of Fairbanks happens at one level on one day. In the case of Lexington, 10% of the $SR_o$ are

higher than 2.

In table 5, the second high increase in the mean relative error happened in the calculation of $\left(\frac{\delta \alpha_a^{Ta}}{\alpha_a^{Ta}}\right)$ from $\left(\frac{\delta \alpha_a}{\alpha_a}\right)$. At

Fairbanks the increase is 7%, from 54% to 61%. At Lexington the increase is 20% from 44% to 64%. The error is

associated with the magnitudes of the relative errors from $\left(\frac{\delta T_a}{T_a}\right)$, conducted at this step by the reasons explained above. At

Fairbanks the mean value of $\left(\frac{\delta T_a}{T_a}\right)$ is 8% while 44% at Lexington, associated to the expression $\delta \tau_a = 0.5 \tau_a$. It should be

taken into account that the total AOD at both sites are dominated by the magnitude of the tropospheric AOD, which is

higher at Lexington.

**Table 5: Relative error estimates of the backscattering ratio, aerosol backscatter at 694 nm and 532 nm, aerosol extinction with and without correction for aerosol two-way transmittance at 532 nm for Lexington and Fairbanks.**

**Errors for $\alpha_a^{Ta} \leq 5 \times 10^{-4}\,km^{-1}$ were not included in the statistics.**

| FAIRBANKS | | | | | | LEXINGTON | | | | | |
| | 694 nm | 532 nm | | | | 694 nm | 532 nm | | | | |
| | $\left(\dfrac{\delta SR}{SR}\right)$ | $\left(\dfrac{\delta\beta_a}{\beta_a}\right)$ | $\left(\dfrac{\delta\beta_a}{\beta_a}\right)$ | $\left(\dfrac{\delta\alpha_a}{\alpha_a}\right)$ | $\left(\dfrac{\delta T_a}{T_a}\right)$ | $\left(\dfrac{\delta\alpha_a^{Ta}}{\alpha_a^{Ta}}\right)$ | $\left(\dfrac{\delta SR}{SR}\right)$ | $\left(\dfrac{\delta\beta_a}{\beta_a}\right)$ | $\left(\dfrac{\delta\beta_a}{\beta_a}\right)$ | $\left(\dfrac{\delta\alpha_a}{\alpha_a}\right)$ | $\left(\dfrac{\delta T_a}{T_a}\right)$ | $\left(\dfrac{\delta\alpha_a^{Ta}}{\alpha_a^{Ta}}\right)$ |
| **Mean** | 12% | 48% | 49% | 54% | 8% | 61% | 13% | 36% | 38% | 44% | 21% | 64% |
| **Maximum** | 13% | 120 | 121 | 122 | 8% | 125% | 16% | 151 | 151 | 152 | 42% | 162% |
| **Minimum** | 11% | 24% | 26% | 31% | 7% | 42% | 11% | 18% | 20% | 27% | 9% | 43% |

The vertical distributions of the $\left(\dfrac{\delta\alpha_a^{Ta}}{\alpha_a^{Ta}}\right)$ realtive errors on consecutive measurements are shown in figures 10 and 11 for Lexington and Fairbanks respectively. Panels a) in both figures are the cross-sections of the $\left(\dfrac{\delta\alpha_a^{Ta}}{\alpha_a^{Ta}}\right)$ relative errors and panels b) are the cross-sections of the $\alpha_a^{Ta}(z, n)$, where n is the consecutive number order of the measurements in each one

of the datasets. We selected this variable to provide a compact view of the magnitudes of the $\left(\dfrac{\delta\alpha_a^{Ta}}{\alpha_a^{Ta}}\right)$ relative errors and

$\alpha_a^{Ta}$ . As expected at both sites the regions with maximum magnitudes of $\alpha_a^{Ta}$ are associated with the lower relative errors. In figure 10 note that at Lexington, for $\alpha_a^{Ta} > 8 \times 10^{-3}\,km^{-1}$ the relative errors has a magnitude equal or lower than 30%. It is also evident that relative errors equal or lower than 50% dominate both in time and altitude. In the case of Fairbanks, figure 11, for $\alpha_a^{Ta} > 2 \times 10^{-3}\,km^{-1}$ the relative error has a magnitude equal or lower than 40%.

Considering the magnitudes of the relative errors for $\alpha_a^{Ta}$ in table 5 it is evident that the $\tau_a^{Ta}$ relative errors are above 100%. Those estimated values of the relative errors for $\tau_a^{Ta}$ together with the ones in table 5 show high magnitudes compared with other sets of volcanically perturbed stratospheric aerosols lidar measurements.

As explained above, the highest error introduced in the $\left(\dfrac{\delta\beta_a}{\beta_a}\right)$ at 694 nm estimation could be reduced if the **SR$_o$** have higher values. In several of the 75 **SR$_o$** profiles renormalization processing could increase its magnitude. That is possible,

because the normalization procedure applied, considered that above 24 km no aerosols were present. Inspection of the plots of **SR$_o$** vs altitude in figures 14, 15 and 16 in **G-66** shows the presence of aerosols between 25 and 30 km and in some of the cases at all of those levels **SR$_o$** magnitude is above 1, the value representing no aerosols. In addition, what will definitely increase the magnitude of **SR$_o$** , will be the introcution of the two-way transmittance correction in the processing generationg **SR$_o$** from the raw returned lidar signal.

In the search for the raw lidar data several options are available. Searching for the filmed images of the oscilloscopes used as registers and/or the original punched cards (probably transferred to tapes) both reported in **G-66**. The last resort would be the digitalization of the **SR$_o$** from the figures cited above. Then the original signal profiles could be reconstructed inverting the normalization procedure applied to produce the **SR$_o$** profiles.

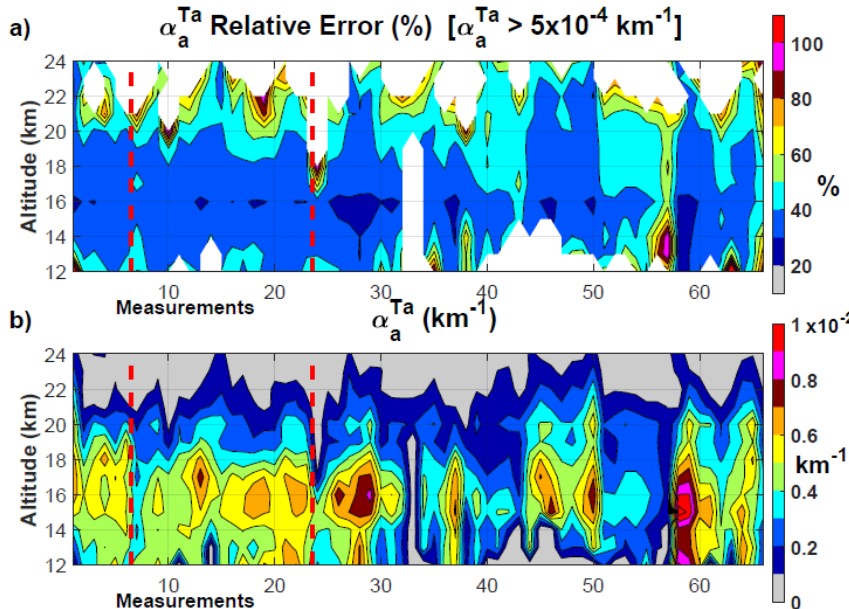

**Figure 10: Cross-section of Relative Error estimates for Lexington, panel a). Panel b) Cross-section of the consecutive measurements. Note the two data gap periods greater than 1 month: March, and July to September both in 1964. They are identified with vertical dotted red lines at the 7 and 23 measurements. In top panel the areas in white in the Relative Error cross-section represent relative errors for $\alpha_a^{Ta} \leq 5 \times 10^{-4}$ km⁻¹. They were not included in the statistics in Table 5.**



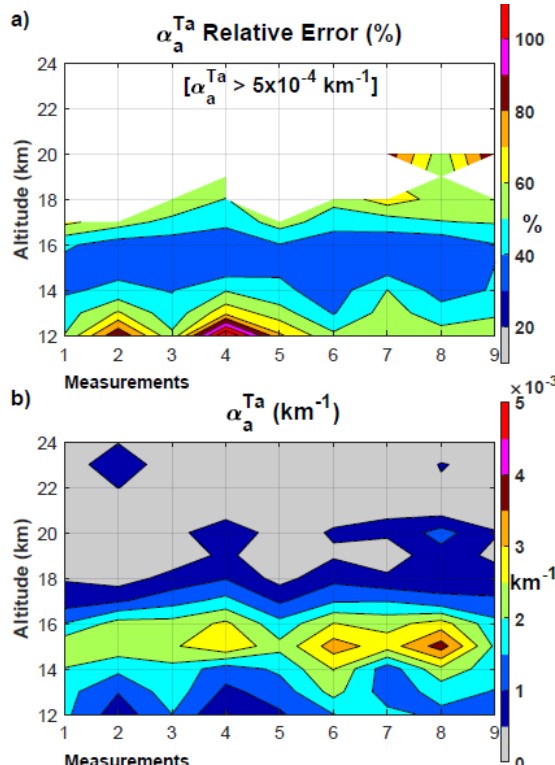

**Figure 11: Idem figure 10 but for Fairbanks.**

*3.4 Attribution of the 1963 Agung aerosol cloud within the Lexington lidar dataset:*


In this section, we seek to understand whether some of the sAOD variations observed by the Lexington lidar may originate from sources other than the March 1963 Agung eruption (such as the two stratosphere-injecting 1963 VEI3 discussed in section 3.2: Trident, Alaska and Vestmannaeyjar, Iceland). Specifically, we compare the Lexington extinction dataset to four different model-based volcanic forcing datasets for the Agung aerosol cloud. Three of the four Agung forcing datasets

are from two different interactive stratospheric aerosol models: two different $SO_2$ emissions scenarios from the UM-UKCA model (Dhomse et al., 2020) and a third simulation from the 2D-AER model (Arfeuille et al. 2014), as applied within the CMIP6 volcanic aerosol dataset (Luo et al., 2016). The fourth simulations is from an idealised model representation of the Agung cloud, based on a simple parameterization for the progression of the tropical reservoir of volcanic aerosol, and its dispersion to mid-latitudes (Ammann et al., 2003), used to represent historical volcanic forcings in some CMIP5 climate

model historical integrations (see Driscoll et al., 2012).

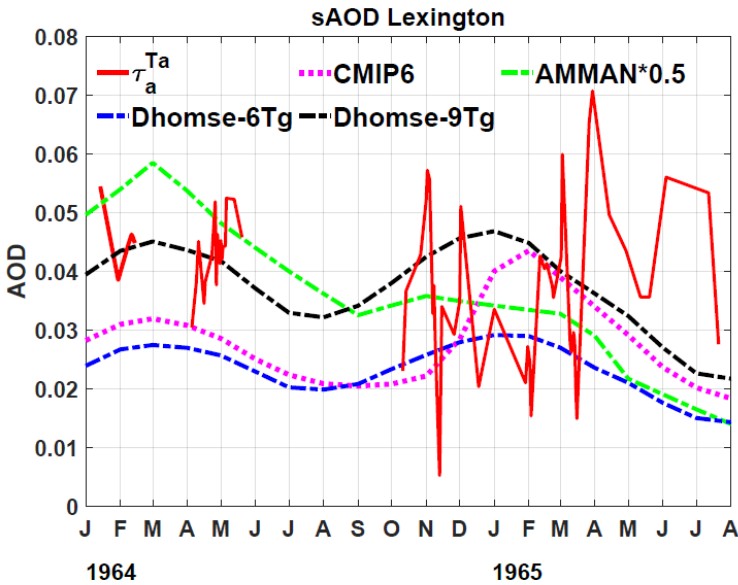

**Figure 12: Model representations of the Agung aerosol cloud sAOD compared to the Lexington dataset.**

The progression of volcanic aerosol clouds from major tropical eruptions reaching the stratosphere was established by Dyer
et al. (1970; 1974) from analysing the extensive synthesis of observations on the Agung aerosol cloud (Dyer and Hicks,
1968), and from knowledge derived from analyses of the global dispersion of radionuclides from Pacific thermonuclear
tests in the 1950s (e.g. Machta and List, 1959). The continual slow upwelling circulation in the tropics, and the sub-tropical
barrier at the edge of the tropical pipe, combine to cause the long-lived tropical stratospheric reservoir (Dyer, 1974; Grant
et al., 1996) which is the reason why tropical eruptions have such prolonged radiative cooling compared to mid-latitude
eruptions. The Brewer -Dobson circulation (Brewer, 1949; Dobson, 1956) has a strong seasonal cycle, transporting air
preferentially towards the winter pole, causing an increasing mid-latitude sAOD trend during autumn and a decreasing mid-
latitude sAOD trend during spring (in both hemispheres). Each of the model lines in Figure 12 show this circulation-driven
seasonal variation in sAOD, with the transport of the Agung aerosol remaining in the tropical reservoir predicted to
increase during October and November, reaching a peak in January to March in both 1964 and 1965. The model predicted
variations are consistent with the initial observed sAOD values of 0.04-0.05 in January and February 1964 being higher
than most of the 0.01-0.04 sAOD values observed in October and November 1964, and the expected variations from the
models suggest the suddenly higher sAOD values ~0.05 may be from a different source than Agung. However, whereas
sAOD values would be expected to increase going into winter, the December January and February sAOD at Lexington are
mostly lower than during the autumn, which indicates an additional source of stratospheric aerosol may have continued to
add to the Agung cloud sAOD throughout the autumn of 1964. Furthermore, the 1965 Lexington observations show a
continuing increase in sAOD into the springtime, whereas the models predict the sAOD from Agung would have reduced
by a factor of 2 during the first 6 months of 1965. The analysis suggests another source of sAOD influential during this
period (either the two VEI3 volcanic eruptions in 1963/4 or some other source of material into the stratosphere) must have
caused the observed increase in stratospheric AOD during 1965, with a potentially substantial influence also during autumn
665  1964.

Figure 13 compares the vertical structure of the 9Tg representation of the Agung aerosol cloud from Dhomse et al. (2020)
at 42N, compared to the Lexington observations, confirming that these model simulations capture the altitude of the cloud
during the early period of the measurements (January to May 1964). However, although the magnitude of the simulated
aerosol extinction compared well to the original Lexington dataset (Dhomse et al., 2020), with the two-way transmittance



corrections applied here, the 9Tg simulation is low-biased compared to the lidar measurements, even in this earlier period, suggesting a with the 12Tg UM-UKCA simulation likely to compare better (not shown). None of the 4 model-generated Arung forcing datasets can explain the observed increase in extinction during Jan to July 1965, with the sudden peaks in April and June 1965 having a quite different vertical structure to the early 1964 measurements, the sAOD in 1965 having a substantial component from the altitude range 18-20km. This vertical profile analysis again suggests the episodic sAOD

enhancements in spring 1965 were from a different source than the 1964 measurements, the altitude of the peak extinction in that first year of the dataset broadly consistent with the UM-UKCA simulations of the Agung cloud.

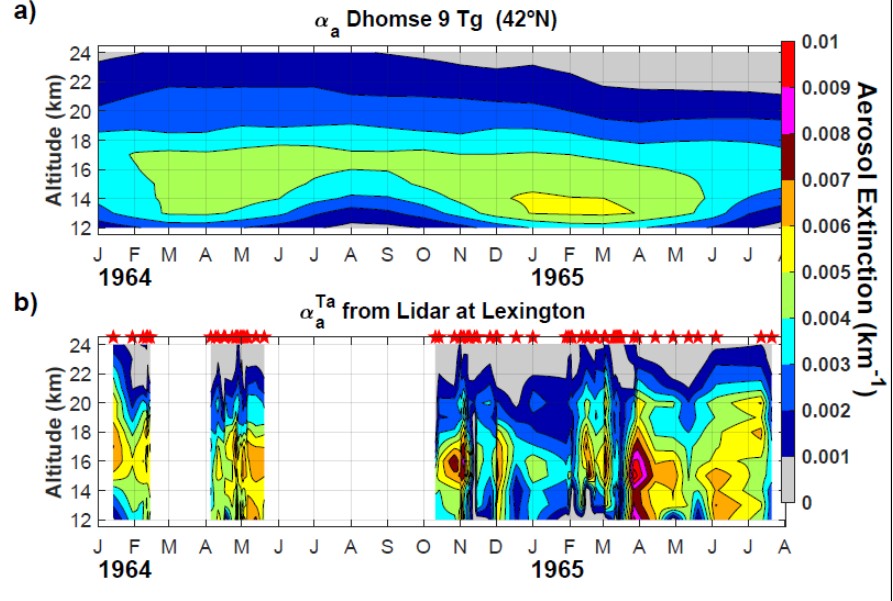

**Figure 13: Cross sections of $\alpha_a(532, z)_*$ from Dhomse et al., (2020) at 42 °N and corrected the two-way transmittance**

$\alpha_a^{Ta}(532, z)$ **from lidar for Lexington.**



**Summary:**

We report completing the processing of the first set of volcanic stratospheric aerosol lidar profiles, from the 1963 Mt. Agung eruption. The results show the high level of variability of the stratospheric aerosol extinction for Lexington between January 1964 and July 1965 mainly attributed to the 1963 Mt. Agung eruption. At Lexington the highest aerosol extinction values and aerosol optical depths are $1.1 \times 10^{-2}$ $km^{-1}$ and 0.076 respectively and were registered by the end of March 1965, almost at the end of the year and a half long record. Based on contemporary and updated reports of additional volcanic
eruptions in the northern hemisphere between1963 and 1965 we found a probable explanation to the apparent contradictory temporal trend of the sAOD magnitudes. Further research, combining observational data and modelling should be conducted to elucidate the individual contribution from each of those eruptions to the stratospheric aerosol layer at this location of the northern hemisphere.

The level of the relative errors are unusually high considering that under high loads of volcanic aerosols in the stratosphere,
the signal to noise ratio is high in the returned lidar signal. The analysis of the contributions of the variables along the different steps of the processing algorithm, allowed identifying the two main sources of error. The main one, accounting for a little more than 30 % of the relative error is associated with the division of the molecular backscatter by the aerosol backscatter, directly linked to low magnitudes of the backscattering ratio. Those low magnitudes are produced by two factors: the first is the lack of two-way transmittance corrections in the backscattering ratio calculation from the raw
squared distance-corrected signal. The second is the normalization method conducted in the region considered to be empty of aerosols, when in many profiles the signal plots reveal its presence. We suggested alternatives to search for the original signal profile records or to reconstruct the original signal profiles from the plotted backscattering ratio records, including the normalization region from 25 to 30 km. The search for original records should include looking for the at least 25 missing profiles from the total of at least 100 Fiocco mentions.

In general the results reported should be considered as the first estimates. We report the comparison of the aerosol extinction values and aerosol optical depths we calculated with information available up to the present, showing reasonable results. Improvements in the two factors cited above lead to an increase in magnitude of the aerosol extinction and optical depth in several of the profiles.

We have also compared the Lexington sAOD timeseries to 4 different model representations of the 1963 Agung aerosol
cloud, and illustrate how the model predictions suggest the sAOD above Lexington from Agung must have decreased from January to July 1965, whereas the 1965 lidar observations show a clear increase in sAOD through the spring into summer. Comparison of the vertical structure of the 1965 measurements to the UM-UKCA Agung aerosol simulations show the Agung cloud descending to lower altitude in 1965 than in 1964, whereas the lidar measurements show more sudden aerosol extinction enhancements, reaching up to 20km in altitude during 1965. Contemporary records of two VEI-3 high latitude
eruptions (in Alaska and Iceland) suggest their volcanic clouds reached the stratosphere in both cases, and model comparisons strengthen the attribution of the January to July 1965 $sAOD_{550}$ increase to a source other than the 1963 Agung eruption.

The datasets of the original rescued backscattering ratios and the calculated aerosol backscatter (both at 694 and 532 nm) and the aerosol extinction at 532 nm (both corrected and uncorrected for two-way aerosols transmittances) at Lexington are
available at https://doi.pangaea.de/10.1594/PANGAEA.922105 (*Dataset in Review*) (Antuña-Marrero et al., 2020a).

**Acknowledgements:**

To the memory of Dr. Giorgio Fiocco, Dr. Gerald W. Grams, for their pioneering research on lidar measurements in
particular the stratospheric aerosols from the Mt. Agung 1963 eruption. We have included them as co-authors as an homage

to their pioneering work. We acknowledge funding from the U.K. National Centre for Atmospheric Science (NCAS) for Dr. Graham Mann via the NERC multi-centre Long-Term Science programme on the North Atlantic climate system (ACSIS). We also acknowledge support from the Copernicus Atmospheric Monitoring Service (CAMS), one of 6 services that together form Copernicus, the EU's Earth observation programme. The global aerosol development tender within

CAMS (CAMS43) funded Juan-Carlos Antuna's 1-month visit to Leeds in March 2019, and 50% co-funded the PhD studentship of Sarah Shallcross, with matched funding from the Institute for Climate and Atmospheric Science, School of Earth and Environment, Univ. Leeds. Dr. Sandip Dhomse and Dr. Graham Mann received funding via the NERC highlight topic consortium project SMURPHS ("Securing Multidisciplinary UndeRstanding and Prediction of Hiatus and Surge periods", NERC grant NE/N006038/1. The UM-UKCA simulations were performed on the UK ARCHER national

supercomputing service with data analysis and storage within the UK collaborative JASMIN data facility. We acknowledge the contribution from Brent Holben for providing the information about the contemporary turbidity measurements. Also the PI's of the AERONET sites for their valuable datasets and Norman T. O'Neill from CARTEL for his contribution with relevant articles. Thanks to Terry Deshler and Horst Jäger for contributing the magnitudes of relative errors for the backscatter to extinction conversion coefficients and the wavelengths exponents. Their comments and suggestions were

very valuable too. JCAM recognizes the support from the Optics Atmospheric Group, Department of Theoretical, Atomic and Optical Physics, University of Valladolid Spain.

**Data availability:**

The data reported in this article is available at https://doi.pangaea.de/10.1594/PANGAEA.922105 (*Dataset in Review*)
(Antuña-Marrero et al., 2020a).

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
