# Peer review of "Recovery of the first ever multi-year lidar dataset of the stratospheric aerosol layer, from Lexington, MA, and Fairbanks, AK, January 1964 to July 1965."

_Earth System Science Data, 2020_

## Referee Comment (RC1) · Anonymous Referee #1 · 14 Jan 2021

**1   Major Comments**

The authors report on the reanalysis of historic first lidar measurements of stratospheric aerosol in the 1960s. Rescueing such old datasets, and re-evaluating them is a very worthwile undertaking, well suited for ESSD. Unfortunately, I find this is a very wordy and lengthy manuscript, which confuses me with lots of unnecessary or even irrelevant information. Generally the manuscript would benefit much from focusing, substantial shortening (by 30% or more), and also from English editing. I provide a few

example lines / sentences below, but nearly all sections could take much more short-ening, e.g. the lengthy introduction about model simulations of stratospheric aerosol, which are motivation, but in detail not relevant here.

The complicated derivation of $\alpha_a(532, z)$ in section 2.4 does not make sense to me. G-66 list $SR_o(694, z)$, but without correction for the 2-way transmission $T_{2w}^2$ due to Rayleigh-extinction, ozone absorption, and aerosol extinction. The way I read it, the authors then derive $\beta_a(694, z)$ from $SR_o(694, z)$ (their Eq. 6), again without correction for the 2-way transmissions. This step requires atmospheric density profiles, and Rayleigh backscattering cross-sections (their Eqs. 3 to 5). They then correct $\beta_a(694, z)$ for the 2-way transmissions due to molecular Rayleigh-extinction and ozone absorption, but not for the 2 way transmission due to aerosol scattering, and go on to derive $\beta_a(532, z)$ (their Eq. 7, using wavelength dependence from Jäger and Desh-ler, 2002, 2003, both missing in the references, see also Jäger et al., GRL, 2005, https://doi.org/10.1029/95GL01521). Then $\beta_a(532, z)$ is converted to $\alpha_a(532, z)$ (us-ing Jäger and Deshler 2002, 2003), and $\alpha_a(532, z)$ is corrected for two-way aerosol transmission (Eqs. 9 and 10). At the end of this process, $\alpha_a(532, z)$ is not consistent with $\beta_a(532, z)$, because the derived $\alpha_a(532, z)$ was not fed back into the derivation of $\beta_a(532, z)$). Results at 532 nm are also very far from the original measurement, $SR_o(694, z)$, or $\beta_a(694, z)$ at 694 nm.

In my opinion, it would be much more logical (and more accurate) to iteratively de-rive a consistent pair $\beta_a(694, z)$ and $\alpha_a(694, z)$ from the tabulated $SR_o(694, z)$ of G-66. The initial step would neglect aerosol extinction $\alpha_a(694, z) = 0$, and derive $\beta_a(694, z)$ (similar to Eqs. 2 to 6), but with appropriate corrections for the 2-way transmissions due to Rayleigh-extinction and ozone absorption (similar to Eq. 7). It is important, and should be mentioned, that at 694 nm and over the altitude range from 12 to 25 km considered here, these 2-way transmissions corrections are small, each less than 2%, because ozone absorption and Rayleigh extinction are very small at 694 nm in the stratosphere. The combined 2-way Rayleigh and ozone transmission at 694

nm through the stratosphere is about 0.97, very close to 1. In the next iterations, $\alpha_a(694, z) = EB(z) \cdot \beta_a(694, z)$ is assumed, and the calculation steps are repeated, including the estimated 2-way aerosol transmission (from $\alpha_a(694, z)$), and provide a new estimate for $\beta_a(694, z)$. Compared to the combined 2-way Rayleigh and ozone transmission ($\approx$0.97), the 2-way aerosol transmission is smaller ($\exp(-2 \cdot 0.04) \approx 0.92$ according to Fig. 9), so accounting for it is more important. The iterations are repeated, and usually converge after 3 to 5 steps. At the end they provide a consistent pair $\beta_a(694, z)$ and $\alpha_a(694, z)$. Note that this is not the case in the current approach of the authors, becaused the derived $\alpha_a(532, z)$ cannot be fed back into the calculation of $\beta_a(694, z)$. If $\alpha_a(532, z)$ is desired, it can be estimated from $\alpha_a(694, z)$ in a final step (using Jäger and Deshler 2002, 2003).

I think the authors need to explain / justify why their approach is valid, and why the approach suggested above was not taken.

Given this problem, as well as the very lengthy and cumbersome text, I feel that the paper needs major revisions. In doing these, the authors should remember that conciseness is very important for any scientific paper. Most readers will be familiar with the basics, and only need to be informed about important and new results. There is no need to start from Adam and Eve, which tends to be the case here.

**2 More Specific Comments**

Abstract: Somewhere, it should be mentioned that the primary quantity measured by a stratospheric lidar (and also produced in the dataset) is the backscatter ratio or the aerosol backscattering coefficient, not the extinction coefficient. Extinction is a derived / secondary quantity. It relies much more on assumptions (about the extinction to backscatter rato, also called lidar ratio) than backscatter. Extinction is usually small, but is necessary to derive the best possible backscatter profile.

line 56: something missing / incomplete sentence?

line 77/78: something missing / incomplete sentence?

line 106/107: replace "very high frequency" by "short pulse". The text says "nano-second", but Table 1 says $< 1\mu$sec. What is correct? I would assume micro-second, because in the 1960s nano-seconds were very hard to achieve, and even harder to measure.

line 107 to 110: Both sentences could easily be deleted.

line 111: after "detected", add "between 10 and 30 km altitude"

line 150: add "in the stratosphere" after $T_{2w}^2$

line 155: Add "This is a good assumption for times of low stratospheric aerosol loading. For enhanced stratospheric aerosol, e.g. after volcanic eruptions, however, aerosol extinction becomes important, reduces the stratospheric transmission, and makes it range dependent."

lines 159 to 161: This could be said much shorter and better. Just say "In a final step, each profile was normalized to one between 25 and 30 km."

lines 162 to 166: Again, very wordy and lengthy. Should be shortened.

lines 180 to 230: lengthy and very confusing!! People unfamiliar with stratospheric aerosol lidar will be totally confused. People familiar don't need this part, but will be puzzled now. What was done? From G-66 and Eq. 1 you can get the lidar return signal $\frac{dn(z)}{dt}$. From that you can go through the process.

lines 212 to 234: I don't understand this discussion, and I don't think it is necessary. Total aerosol optical depth is not needed, only stratospheric aerosol depth is needed between 12 and 25 km, the altitudes tabulated in G-66. The authors do not need two-way transmission from the ground to $z$, they only need stratospheric two-way transmissions from the normalization altitude (or 25 km) down to $z$. Since total aerosol optical

depth is very variable, and usually dominated by the troposphere, any use of total opti-
cal depth data here is frought with large uncertainties. Lines 290 to 340 and Fig. 1 are
also not needed for the same reasons. I strongly suggest to remove all this confusing
and unnecessary material.

Sections 2.5, 2.5.1, 2.5.2, very long and wordy. should be shortened substantially. The
key points should have already been said / explained in the description of the method
to get from $SR_o(694, z)$ tabulated in G-66 to $\beta_a(z)$ and $\alpha_a(z)$.

In section 2.5.1: I think the authors need to explain, here or when they decribe their
algorithm, that the US-Standard Atmosphere density profile needs to be backed out
from $SR_o(694, z)$ tabulated in G-66 to get the original lidar return signal $\frac{dn(z)}{dt} \cdot \frac{z^2}{K}$ in Eq.
1, and that then better / newer profiles are used to derive $\beta_a(z)$.

Also in 2.5.1 and 2.5.2, it needs to be stated that only the stratospheric parts of the two
way transmissions due to Rayleigh-extinction and ozone absorption are needed, and
that both of these are almost constant at $\approx$ 0.98, and very close to 1. This means that
it is essentially irrelevant, which atmospheric profiles are used to account for the two
transmissions.

Section 2.5.3: Should be removed entirely, as mentioned above.

Section 2.7.2: I think it would be very helpful to see a typical altitude profile of the
overall error, and the different contributions. I would expect, that the contributions
from Rayleigh and ozone two-way transmissions are quite small, and that other terms
dominate.

Section 3.1: should $\beta_m(594, z)$ not be $\beta_m(694, z)$ throughout this section (and in a few
other places)? To me, it would be clearer in most places to say "density profile from
the US Standard Atmosphere" and "density profiles from local radiosondes" instead of
"$\beta_m(694, z)$ from the US Standard Atmosphere" and "$\beta_m(694, z)$ from the soundings"

Figure 4: it would be good to have two more panels showing the difference (or the ratio)

of $\beta_{a,US}(694, z)$ and $\beta_{a,Radiosondes}(694, z)$ (or $\alpha_{a,US}(694, z)$ and $\alpha_{a,Radiosondes}(694, z)$). Fig. 2 and 3 could even be dropped then.

Figures 5, 8, 9: the indicator symbols at the top axes are not needed and confusing. They should be removed.

Figure 9 contains the same information as Fig. 5, and is more comprehenisve. Fig. 5 should be dropped. The entire discussion of results without correction for aerosol 2-way transmission should be shortened substantially. We know these results are poorer, and Fig. 9 shows it very clearly. There is not point in lengthy discussions of things that are obsolete or have been superseded.

After Figs. 6 and 7: It might be good to have a Figure similar to what I suggest above for Fig. 4, but showing the differences between profiles with and without correction for aerosol 2-way transmission.

Figure 8: these results should be included in Fig. 9, and Fig. 8 dropped.

Lines 581 to 586: This is a very complicated way of saying that the errors in the aerosol parameters blow up, when there is little aerosol, and $SR$ is close to 1.

Lines 587 to 592: A very complicated way of saying that uncertainty due to aerosol extinction correction becomes large, when aerosol extinction becomes large.

Lines 597 to 618: Difficult to read, and difficult to find take-home messages.

Section 3.4: Given the overall length of this data paper, and to maintain a better focus, I would suggest to drop this entire section, including Figs. 12 and 13. This is supposed to by an ESSD data paper, not an ACP paper.

---

## Referee Comment (RC2) · Anonymous Referee #2 · 25 Jan 2021

This paper presents in my view a valuable contribution to the rescue of old geophysical data - in this case from early lidar measurements of stratospheric aerosols - for the sake of their use in the reconstruction of past volcanic events. The work is a contribution to the Data Rescue activity of the Stratospheric Sulfur and its Role in Climate within the SPARC project.

The crux of the work is the extraction of aerosol extinction coefficients at 532 nm between 12 km and 24 km from backscattering-ratio results at 694 nm retrieved in that range, under simplifying hypotheses, from lidar measurements carried out in different periods of 1964 at two different locations (Lexington, Massachusetts, and College, Alaska).

While the "translation" from the original results (the backscattering ratios at a given wavelength under the mentioned simplifying hypotheses) to the extracted ones (the aerosol extinction coefficients at another wavelength and correcting for the simplifications) is carefully explained, I found apparent inconsistences and ambiguities in the developed formulation, as well as in the notation, that the authors should explain or, if my concerns are proven right, correct.

**I. The notation ambiguities are the following:**

1. The symbol $N_A$ is used with two different meanings. In Eq. (1) $N_A(z)$ is the molecular number density at altitude z. In Eq. (4) it is used for Avogadro's number.

2. Two different symbols, $SR_o(\lambda,z)$ and $SR(\lambda,z)$, are used for the backscattering ratio without a clear reason for establishing a difference. This is apparent in Eq. (6), where the $\beta_a^A(z)$ expression is said to be derived from Eq. (2), but the symbol $SR_o(\lambda,z)$ appears instead of the symbol $SR(\lambda,z)$ used in Eq. (2).

3. The notation $T_{2w}^2$ for the two-way atmospheric transmittance is in my view redundant: the two-way is implicit in the squared superscript. The subscript should be left to denote the origin of the transmittance, as is generically done in Eq. (8). Note, related to this, that in line 180 $T_T^2$ seems to be used with the same meaning as $T_{2w}^2$.

4. Letting aside possible corrections arising from the reasoning in point #2 sin section III of this review below, on possible inconsistences in the formulation, I think that there are possible ambiguities in the notation that should be clarified. For example, $\beta_a^A$ is used to denote the aerosol attenuated backscatter coefficient, which is subsequently corrected by the molecular and ozone transmittances in Eq. (7). The result is called $\beta_a$, which seems to imply that this is the final aerosol backscatter coefficient, from which using the factor BEc(z, t) (by the way, what does the variable t mean?) the aerosol extinction coefficient $\alpha_a$ is derived. (Eq. (9)). But then one discovers that $\alpha_a$ has still to be corrected for the aerosol transmittance, the final aerosol extinction coefficient being called $\alpha_a^{Ta}$. Although

this is a minor remark, I think it would be clearer to reserve the symbols $\beta_a^{Ta}$, $\alpha_a^{Ta}$ for the coefficients yet to be corrected for the aerosol transmittance and to use $\beta_a$, $\alpha_a$ for the final, fully corrected extinction coefficient.

**II. With respect to the inconsistences in the formul developments:**

1. While Eq. (1) is actually found in ref. **G-66** (Eq. (4.2) of this reference), it should be noted that this equation refers to the "expected signal from a molecular atmosphere" (page 50 of **G-66**), as it is made clear by the sentence (also in page 50 of **G-66**): "Thus, to derive the dust profiles, it is necessary to evaluate the intensity of the echoes for a dust-free atmosphere by using equation 3.8 for the case of Rayleigh scattering by air molecules." Therefore the statement on line 144 of the paper under review referring to Eq, (1): "The average photoelectron flux registered by the photomultiplier was represented by the expression:" is misleading: the average photoelectron flux was actually represented by

$\dfrac{dn(z)}{dt} = K\dfrac{\beta}{z^2}$ (Eq. (3.8) of **G-66**, with $K$ condensing all the multiplicative constants in that equation), Eq. (1) corresponding to the expected photoelectron flux from a molecular (reference) atmosphere.

2. The authors seem to imply that the scattering ratio $SR_0(\lambda,z)$ obtained in ref. **G-66** corresponds to the expression in Eq. (2). However, following the data processing steps described in the paper I don't arrive at that expression. I explain in detail my understanding of the steps described in the manuscript to sustain my statement. For clarity, I keep the authors' notation notwithstanding my remark #3 on the notation in section I above:

a) To begin with, the photoelectron flux will actually be, taking into account the atmospheric transmittance,

$$\frac{dn(z)}{dt} = K\frac{\beta(\lambda,z)T_{2w}^2(z)}{z^2} = K\frac{\left[\beta_m(\lambda,z) + \beta_a(\lambda,z)\right]T_{2w}^2(z)}{z^2},$$

with $\beta(\lambda.z) = \beta_m(\lambda,z) + \beta_a(\lambda,z)$ the total backscatter coefficient, being $\beta_m(\lambda,z)$ and $\beta_a(\lambda,z)$ the molecular and aerosol backscatter coefficients respectively. Note that I'm using $\beta_a(\lambda,z)$ for the "true" (i.e. not affected by any attenuation) aerosol backscatter coefficient.

b) According to the authors: "Next, the ratios between the averaged signal at each level and the values at the same level of the right side of the equation (1) were calculated for each profile between 12 and 30 km". Calling that ratio $P_n$, I understand that

$$P_n(z) = \frac{\dfrac{dn(z)}{dt}}{\dfrac{N_A(z)}{z^2}} = K\frac{\left[\beta_m(\lambda,z) + \beta_a(\lambda,z)\right]T_{2w}^2(z)}{\beta_m(\lambda,z)}$$

c) Following the authors: "A final step consisted in normalizing the ratios calculated in each profile between 12 and 24 km. To that end, for each profile the average value between 25 and 30 km of the ratios calculated in the former step were determined". I will call the average value between 25 and 30 km $\overline{P}_n(z_1, z_2)$, with $z_1 = 25$ km and $z_2 = 30$ km. Then, assuming, as **G-66** seem to do, that $\beta_a(\lambda, z)$ is negligible in that range, we would have

$$\overline{P}_n(z_1, z_2) = \frac{1}{z_2 - z_1}\int_{z_1}^{z_2} P_n(z)\,dz = \frac{K}{z_2 - z_1}\int_{z_1}^{z_2} T_{2w}^2(z)\,dz = KT_{2w}^2(z_0),$$

with, according to the finite-increment theorem, $z_0$ a range between $z_1$ and $z_2$ whose value will depend on the form of $T_{2w}^2(z)$.

d) Always following the authors: "Then for each profile the ratios in the altitude range 12 and 24 km were divided by the average value of the ratios between 25 and 30 km from the same profile". Calling, as the authors seem to do, SR($\lambda$,z) (or SR$_0$($\lambda$,z), see my remark #2 on the notation ambiguities) the result of dividing the ratio $P_n$ in the range 12 km – 24 km by $\overline{P}_n(z_1, z_2)$. Then

$$SR(\lambda, z) = \frac{P_n(z)}{\overline{P}_n(z_1, z_2)} = \frac{\left[\beta_m(\lambda,z) + \beta_a(\lambda,z)\right]T_{2w}^2(z)}{\beta_m(\lambda,z)T_{2w}^2(z_0)},$$

which do not coincide with the result given in Eq. (2).

The authors should, either point out possible mistakes I may have committed in the above development, or else correct theirs.

4. I couldn't understand the (iterative?) process described in lines $211 - 217$ to obtain $\alpha_a^{Ta}$. How is the $T_a^2(532, z)$ first guess of obtained? How is it refined? I suggest illustrating the procedure with a graph.

**III. Other remarks**

1. Line 77: "were been produce" should be just "were produced".

2. Line 80: "Giogio Fiocco" should be "Giorgio Fiocco".

3. Line 86: What do CMIP5 and CMIP6 refer to? Further explanations and possibly a reference are needed.

4. Line 137: "It is well known that solving the equation for the single wavelength elastic lidar is an ill-posed problem". What is the equation the sentence refers to? I suppose it is the lidar equation, so the sentence should read "It is well known that solving the lidar equation…" As implied in remark # 1 of section II of this review, this should be Eq. (1), but written in the form of Eq. (3.8) of **G-66.**

5. Line 105: the description "very high frequency nano-second laser" is misleading, as it seems to mix the laser pulse duration with the pulse repetition rate, the latter being very low, according to table 1, by today's standards.

6. I think it would be beneficial for the reader to point out that $\dfrac{dn(z)}{dt}$ is proportional to the collected backscattered power from that range $z$.

7. Line 146: $n_z$ should probably be $n(z)$.

8. The references Jäger and Deshler, 2002 and 2003 are missing in the reference list. Please check that all cited references are included in the list.

9. Line 181: "Hosteler" should be "Hostetler"

10. In lines 186-187 it is said: "after neglecting the dispersion of the refractive index and the King factor of the air represented by $k_{bw}$". If the King factor is not used, it is unnecessary to assign it a symbol.

11. Unnecessary bracketing is used sometimes. For example, but not only, Eq. (2) could simply be written as

$$SR(\lambda,z) = \frac{\beta_m(\lambda,z) + \beta_a^A(\lambda,z)}{\beta_m(\lambda,z)},$$

without the curly brackets. Other instances of unnecessary brackets are found throughout the paper.

12. Lines 321-322: "After the conversion to 532 nm they were respectively 0.087 and 0.242…" What does "they" in this sentence refers to?

13. Line 326: "The high TAOD values for the twos series…" Which ones are the two series?

14. Lines 367-368: "the data we rescue are a reasonable approximation of what we today know as the backscattering ratio described in equation (2)". But the definition in Eq. (2), letting aside the concerns expressed in point #2 of section II of this review as to its derivation, uses the attenuated backscatter, while, in my understanding, the present-day backscattering ratio definition is $SR = \dfrac{\beta_m + \beta_a}{\beta_m}$, with $\beta_m$ the molecular backscatter coefficient and $\beta_a$ the "true", not affected by any attenuation, aerosol backscatter coefficient.

---

## Author Comment (AC1) · 15 Mar 2021

**Replies to Reviewers Comments:**
We thank the reviewers for the comments and suggestions they made, contributing to the improvement of the manuscript.

**Introductory statement by the authors:**
The SSIRC data rescue activity has a philosophy to involve modeling scientists as well as observational scientists, both to improve communication between the often separated communities, and also to help identify priority measurement datasets and aerosol metrics that can be of most benefit to the modelling community. Reviewer 1 has requested to remove the comparisons to model predictions dataset in section 3.4 of the manuscript, but we feel strongly this is an important element of the manuscript, highlighting why the observations dataset is of such importance both to current international climate modelling activities such as CMIP6 (Eyring et al., 2016; Zanchettin et al., 2016) and to stratospheric aerosol modelling activities such as ISA-MIP (Timmreck et al. 2018). We feel that applying strict rules to separate the publication of observational datasets and modelling datasets would in this case be in confct also with the spirit of the ESSD journal to promote international interdisciplinary research.

**Anonymous Referee #1**

**1 Major Comments**

The authors report on the reanalysis of historic first lidar measurements of stratospheric aerosol in the 1960s. Rescuing such old datasets, and re-evaluating them is a very worthwile undertaking, well suited for ESSD. Unfortunately, I find this is a very wordy and lengthy manuscript, which confuses me with lots of unnecessary or even irrelevant information. Generally the manuscript would benefit much from focusing, substantial shortening (by 30% or more), and also from English editing. I provide a few example lines / sentences below, but nearly all sections could take much more shortening, e.g. the lengthy introduction about model simulations of stratospheric aerosol, which are motivation, but in detail not relevant here.

**Reply:**
Although it would have been possible for us to have written this manuscript solely to describe the recovery and processing of the lidar observational datasets, we chose to design the paper also to seek to understand the variations seen in the measurements, via comparison to interactive stratospheric aerosol model simulations of the Agung aerosol cloud (Dhomse et al., 2020). These model simulations were carried out following exactly the protocol for the co-ordinated "HErSEA Agung" experiment within the international modelling initiative ISA-MIP (Timmreck et al., 2018). The inclusion also of comparisons with the volcanic forcing datasets for the CMIP5 and CMIP6 climate modeling initatives means this section 3.4 provides valuable additional context for the recovery methodology, aligning with both of these two international community modelling activities.

Although we agree certainly a description of the model would be outside the scope of this paper, since we see section 3.4 as an important part of the paper, we choose to retain the rationale we have included in the Introduction, to explain to readers why these new measurements are of wider significance to the modelling community. For example to alert the reader to be aware of the large change in the stratospheric aerosol optical properties (and hence surface cooling) that climate model enact for the Agung cloud within CMIP6 historical integrations, compared to that enacted for CMIP5 (see Niemeier et al., 2019).

We have improved the wording in the sentences in the Abstract to make clearer the relevance of the re-calibration to account for the two-way transmittance effect, and the initial model comparison that suggests the 1965 variations in the recovered measurements are not from the Agung aerosol cloud.

We have revised lines 25 to 33 in the Abstract, to improve the communication of these aspects, the wording in the revised manuscript reading:

**Lines 25-28 re-worded to:**
*"We show that accounting for these two-way transmittance effects substantially increases the magnitude of the 1964/5 stratospheric aerosol layer's optical thickness in the Northern Hemisphere mid-latitudes, then ~50% larger than represented in the CMIP6 volcanic forcing dataset. Compared to the uncorrected dataset, the combined transmittance correction increases sAOD$_{532}$ by up to 66% for Lexington, and up to 26% for Fairbanks, individual sAEP adjustments of similar magnitude."*

**Lines 30-31 re-worded to:**
*"Within the January 1964 to August 1965 measurement timespan, the corrected Lexington sAOD$_{532}$ timeseries is substantially above 0.05 in three distinct periods: October 1964, March 1965 and May-June 1965, whereas the 6 nights the lidar measured in December 1964 and January 1965 had sAOD$_{532}$ at most ~0.03."*

**Lines 31-33 re-worded to:**
*"Comparing to interactive stratospheric aerosol model simulations of the Agung aerosol cloud shows that, although substantial variation in mid-latitude sAOD$_{532}$ are expected from the seasonal cycle in the stratospheric circulation, the Agung cloud's dispersion from the tropics would have been at its strongest in winter, and weakest in summer. The increasing trend in sAOD from January to July 1965, also considering the large variability, suggests that the observed variations are from a different source than Agung, possibly from one or both of the two VEI3 eruptions…."*

The complicated derivation of $\alpha_a(532, z)$ in section 2.4 does not make sense to me. **G-66** list SRo(694, z), but without correction for the 2-way transmission $T_{2w}^2$ due to Rayleigh-extinction, ozone absorption, and aerosol extinction. The way I read it, the authors then derive $\beta_a(694, z)$ from SRo(694, z) (their Eq. 6), again without correction for the 2-way transmissions. This step requires atmospheric density profiles, and Rayleigh backscattering cross-sections (their Eqs. 3 to 5). They then correct $\beta_a(694, z)$ for the 2-way transmissions due to molecular Rayleigh-extinction and ozone absorption, but not for the 2 way transmission due to aerosol scattering, and go on to derive $\beta_a(532, z)$ (their Eq. 7, using wavelength dependence from *Jäger and Deshler, 2002, 2003, both missing* in the references, see also Jäger et al., GRL, 2005, https://doi.org/10.1029/95GL01521). Then $\beta_a(532, z)$ is converted to $\alpha_a(532, z)$ (using Jäger and Deshler 2002, 2003), and $\alpha_a(532, z)$ is corrected for two-way aerosol transmission (Eqs. 9 and 10). At the end of this process, $\alpha_a(532, z)$ is not consistent with $\beta_a(532, z)$, because the derived $\alpha_a(532, z)$ was not fed back into the derivation of $\beta_a(532, z)$. Results at 532 nm are also very far from the original measurement, SRo(694, z), or $\beta_a(532, z)$ at 694 nm.

**Reply**: The references of Jäger and Deshler, 2002, 2003, have been added.

In my opinion, it would be much more logical (and more accurate) to iteratively derive a consistent pair $\beta_a(532, z)$ and $\alpha_a(532, z)$ from the tabulated SRo(694, z) of **G-66**. The initial step would neglect aerosol extinction $\alpha_a(694, z) = 0$, and derive $\beta_a(694, z)$ (similar to Eqs. 2 to 6), but with appropriate corrections for the 2-way transmissions due to Rayleigh-extinction and ozone absorption (similar to Eq. 7). It is important, and should be mentioned, that at 694 nm and over the altitude range from 12 to 25 km considered here, these 2-way transmissions corrections are small, each less than 2%, because ozone absorption and Rayleigh extinction are very small at 694 nm in the stratosphere. The combined 2-way Rayleigh and ozone transmission at 694 nm through the stratosphere is about 0.97, very close to 1. In the next iterations, $\alpha_a(694, z) = $ EB(z) $\beta_a(694, z)$ is assumed, and the

calculation steps are repeated, including the estimated 2-way aerosol transmission (from $\alpha_a(694, z)$), and provide a new estimate for $\beta_a(694, z)$. Compared to the combined 2-way Rayleigh and ozone transmission ($\approx 0.97$), the 2-way aerosol transmission is smaller ($\exp(-2.04) \approx 0.92$ according to Fig. 9), so accounting for it is more important. The iterations are repeated, and usually converge after 3 to 5 steps. At the end they provide a consistent pair $\beta_a(694, z)$ and $\alpha_a(694, z)$. Note that this is not the case in the current approach of the authors, because the derived $\alpha_a(532, z)$ cannot be fed back into the calculation of $\beta_a(694, z)$. If $\alpha_a(532, z)$ is desired, it can be estimated from $\alpha_a(694, z)$ in a final step (using Jäger and Deshler 2002, 2003).

I think the authors need to explain / justify why their approach is valid, and why the approach suggested above was not taken.

**Reply**: There are several approaches to process stratospheric aerosol lidar signals. The reviewer suggest one of them.

The principal effect produced by stratospheric aerosols from volcanic eruptions is the scattering of solar radiation, causing the radiative forcing of the atmosphere-earth system. Global aerosol models tend to diagnose mid-visible aerosol optical properties at either 550nm or 532nm, and we choose the latter as the standard wavelength to report the derived aerosol extinction, to align with that from modern Nd:YAG lidars.

We calculate the two-way aerosols transmittance using the total AOD from surface observations (which includes tropospheric and stratospheric AOD) but produce no profile of the two way transmittance and apply the two-way transmittance aerosol correction to derive a first guess of $\alpha_a(532, z)$. In the next step we calculate the tropospheric AOD by subtracting the stratospheric AOD from the Total AOD and producing a profile of AOD between 12 and 24 km, which is used to derive a two way transmittance profile between 12 and 24 km that will be applied again to the uncorrected $\alpha_a(532, z)$ to produce the two way transmittance corrected stratospheric aerosols profile $\alpha_a^{Ta}(532, z)$. Still and iteration of those final steps is possible, but the high magnitude of the estimated $\alpha_a^{Ta}(532, z)$ mean error, around 60%, compared to a estimated 15-20% maximum improvement achieved by the iteration procedure, makes it nonsense.

Because all the equations involved in the processing Eq. (6) to (10) consist of products and divisions, for the goal to calculate $\alpha_a^{Ta}(532, z)$ the correction by the two way transmittance may be applied directly in the calculation of $\alpha_a^{Ta}(532, z)$, not been necessary to fed back $\alpha_a(532, z)$ to calculate $\beta_a(532, z)$.

For our purposes $\alpha_a(694, z)$ has no interest at all. Moreover, because our only interest is $\alpha_a(532, z)$. We do not have any interest also to fed back $\alpha_a(532, z)$ into the calculation of $\beta_a(694, z)$ or $\beta_a(532, z)$. However any researcher interested in that variable may calculate himself using the rescued SR dataset and the available variables, all stored in PANGAEA.

Given this problem, as well as the very lengthy and cumbersome text, I feel that the paper needs major revisions. In doing these, the authors should remember that conciseness is very important for any scientific paper. Most readers will be familiar with the basics, and only need to be informed about important and new results. There is no need to start from Adam and Eve, which tends to be the case here.

**2 More Specific Comments**

Abstract: Somewhere, it should be mentioned that the primary quantity measured by a stratospheric lidar (and also produced in the dataset) is the backscatter ratio or the aerosol backscattering coefficient, not the extinction coefficient. Extinction is a derived / secondary quantity. It relies much more on assumptions (about the extinction to backscatter ratio, also called lidar ratio) than backscatter. Extinction is usually small, but is necessary to derive the best possible backscatter profile.

**Reply:** We do not agree with the reviewer. The information he suggest to include in the abstract is very basic lidar information. The sentence "Extinction is usually small, but is necessary to derive the best possible backscatter profile." Does not apply in cases of the major volcanic eruptions, which is the case we present.

line 56: something missing / incomplete sentence?

**Reply:** The reviewer is referring to this sentence:

*"The main motivation for this HErSEA multi-model experiment (Historical Eruption SO2 Emission Assessment) is to gather stratospheric aerosol observations to evaluate the model simulations, and understand the current diversity in the sulphur emission amount and altitude distribution interactive stratospheric aerosol models use when simulating the Pinatubo aerosol cloud (see section 3.3.2 of Timmreck et al., 2008)."*

We do not consider that sentence to be missing something or to be incomplete. There was a typo re: the citation of the Timmreck et al. paper being published in 2018 not in 2008, and we have corrected that in the revised manuscript. The word "evaluate" could perhaps have been improved to change "observations to evaluate the" instead to "observations to provide new constraints to evaluate the", and we have made that change also in the revised manuscript. On reflection, we see that also adding "in the periods after major tropical eruptions" is then clearer which periods are the priority within the historical record.

We have therefore re-worded that sentence into two sentences, also being clear that both aspects mentioned are motivations within the activity. The revised text reads as follows:

*"One of the main motivations within this HErSEA multi-model experiment (Historical Eruption SO$_2$ Emission Assessment) is to gather stratospheric aerosol observations in the periods after major tropical eruptions to provide new contraints to evaluate the model simulations. Another is to seek to understand whether the current diversity in the sulfur emission amount and altitude distribution that stratospheric aerosol models use when simulating the Pinatubo aerosol cloud is also seen for Agung (see section 3.3.2 of Timmreck et al., 2018)."*

line 77/78: something missing / incomplete sentence?

**Reply:** The sentence:
*"Although the descent in the peak of the backscatter ratio profile from Lexington is analysed within GF-67, only limited estimates of the cloud's aerosol extinction exist (2 x 10$^{-3}$ km$^{-1}$ at 16 km and the aerosol optical depth of 0.015 (Deirmejian, 1971) were produced."*
Has been changed:
*"Only a few quantitative estimates of the cloud's optical properties from the lidar dataset have been found; aerosol extinction of 2 x 10$^{-3}$ km$^{-1}$ at 16 km and the aerosol optical depth of 0.015 (Deirmejian, 1971)."*

line 106/107: replace "very high frequency" by "short pulse". The text says "nanosecond", but Table 1 says < 1µsec. What is correct? I would assume micro-second, because in the 1960s nano-seconds were very hard to achieve, and even harder to measure.

**Reply:** Corrected to "short pulse". It is microseconds also corrected.

line 107 to 110: Both sentences could easily be deleted.

**Reply:** We prefer to keep both sentences, to honor and publicize the broad and extensive pioneering work conducted by Dr. Giorgio Fiocco and his colleagues.

line 111: after "detected", add "between 10 and 30 km altitude"

**Reply:** Corrected.

line 150: add "in the stratosphere" after $T_{2w}^2$

**Reply:** Corrected.

line 155: Add "This is a good assumption for times of low stratospheric aerosol loading. For enhanced stratospheric aerosol, e.g. after volcanic eruptions, however, aerosol extinction becomes important, reduces the stratospheric transmission, and makes it range dependent."

**Reply:** Added

lines 159 to 161: This could be said much shorter and better. Just say "In a final step, each profile was normalized to one between 25 and 30 km."

lines 162 to 166: Again, very wordy and lengthy. Should be shortened.

**Reply:** Regarding the above 2 comments, we rewrote the sentences from line 159 to 166:

*"A final step consisted in normalizing the ratios calculated in each profile between 12 and 24 km.To that end, for each profile the average value between 25 and 30 km of the ratios calculated in the former step were determined. Then for each profile the ratios in the altitude range 12 and 24 km were divided by the average value of the ratios between 25 and 30 km from the same profile. The resulting values were considered to be the backscattering ratio (SRo(λ, z)): the ratio between the total (aerosols + molecules) backscattering divided by the molecular backscattering. The normalization procedure assigned the backscattering ratio to be equal to one above 25 km, after assuming the contribution from aerosols was negligible compared to the molecular at those levels."*

The corrected sentences are:

*A final step for each profile consisted in normalizing the ratios calculated in each profile between 12 and 24 km. The, with the average ratios between 25 and 30 km for each profile were calculated. Following the ratios between 12 and 24 km in each profile were divided by the respective average value ratio between 25 and 30 km, producing the derived SR(694, z) under the assumptions already cited. The normalization procedure assumed the contribution from aerosols was negligible compared to the molecular at the normalization levels. This assumption would lead above 24 km, leading to an under-estimate of stratospheric aerosol since there would have been aerosol at these altitudes (Russell, et al., 1979)..*

lines 180 to 230: lengthy and very confusing!! People unfamiliar with stratospheric aerosol lidar will be totally confused. People familiar don't need this part, but will be puzzled now. What was done? From **G-66** and Eq. 1 you can get the lidar return signal $\frac{dn(z)}{dt}$ . From that you can go through the process.

**Reply:** We do not agree with the reviewer. In the first paragraph of section 2.4 we explain the motivation of this section. It is a step by step explanation. The reviewer is confused: what is available are the tabulated values of SRo(λ, z), not $\frac{dn(z)}{dt}$ . It is not possible to get $\frac{dn(z)}{dt}$ back because we not know the individual profiles average ratios from 25 to 30 km.

lines 212 to 234: I don't understand this discussion, and I don't think it is necessary. Total aerosol optical depth is not needed, only stratospheric aerosol depth is needed between 12 and 25 km, the altitudes tabulated in **G-66**. The authors do not need two-way transmission from the ground to z, they only need stratospheric two-way transmissions from the normalization altitude (or 25 km) down to z. Since total aerosol optical depth is very variable, and usually dominated by the troposphere, any use of total optical depth data here is frought with large uncertainties.

**Reply:** We disagree with the reviewer. It is necessary to remember that the attenuation processes of the emitted and backscattered laser signal **takes place along the complete path between the emitter, the atmospheric backscatterers and the sensor.** In this case the emitter and the sensor are located at the surface. Then tropospheric atmospheric components (including tropospheric aerosols) contribute to the laser signal attenuation. Assuming negligible tropospheric AOD has been a common practice introducing large uncertainties. In the case of high tropospheric AOD those uncertainties are higher that the uncertainties associated to the tropospheric AOD variability. Applying the 2-way atmospheric transmittance correction to stratospheric aerosols (SA) lidar signals from major volcanic eruptions (including the tropospheric AOD) is very important to produce realistic quantitative estimates of the SA optical properties.

Lines 290 to 340 and Fig. 1 are also not needed for the same reasons. I strongly suggest to remove all this confusing and unnecessary material.

**Reply:** We disagree with the reviewer. As we stated above, it has been a common practice in the processing of stratospheric aerosols lidar signals to discard the two way attenuation produced by tropospheric aerosols. That is one of the key points of the paper: it should be done whatever

is necessary to find tropospheric AOD (if it is available still it may be used total AOD and produce tropospheric AOD after you get the first stratospheric AOD by subtraction). That will allow to conduct the 2-way aerosols transmittance correction. The figure is necessary to support why the correction by two way aerosols transmittance should be conducted.

Sections 2.5, 2.5.1, 2.5.2, very long and wordy. should be shortened substantially. The key points should have already been said / explained in the description of the method to get from SRo(694, z) tabulated in G-66 to $\beta_a(z)$ and $\alpha_a(z)$.

In section 2.5.1: I think the authors need to explain, here or when they describe their algorithm, that the US-Standard Atmosphere density profile needs to be backed out from SRo(694, z) tabulated in G-66 to get the original lidar return signal $\frac{dn(z)}{dt}\frac{z^2}{K}$ in Eq. 1, and that then better / newer profiles are used to derive $\alpha_a(z)$.

**Reply:** We have already explained above it is not possible to retrieve the original lidar returned signal.

Also in 2.5.1 and 2.5.2, it needs to be stated that only the stratospheric parts of the two way transmissions due to Rayleigh-extinction and ozone absorption are needed, and that both of these are almost constant at _ 0.98, and very close to 1. This means that it is essentially irrelevant, which atmospheric profiles are used to account for the two transmissions.

Section 2.5.3: Should be removed entirely, as mentioned above.

**Reply:** Regarding the above 4 comments with respect to section 2.5 and its subsections. We disagree with the reviewer on his requests to remove subsections of section 5. Instead, we have moved the necessary detailed explanations originally included in the manuscript to the Supplement 1. Section 2.5 is now a synthetized description of the complementary datasets used

Section 2.7.2: I think it would be very helpful to see a typical altitude profile of the overall error, and the different contributions. I would expect, that the contributions from Rayleigh and ozone two-way transmissions are quite small, and that other terms dominate.

**Reply:** Figures 10a and 11a provide time vs. altitude relative error cross section of the aerosol extinction corrected by two-way transmittance, together with the respective time vs. altitude cross section of the aerosol extinction corrected by two-way transmittance in the respective figures 10b and 10c. Together with the tabulated results in Table 5, there is enough information to evaluate the error levels on time and altitude in the main variable of interest that we did. The contributions from Rayleigh and ozone two-way transmittance have been already discussed in the literature, for example Russell (1979).

Section 3.1: should $\beta_m(\lambda, 594)$ not be $\beta_m(\lambda, 694)$ throughout this section (and in a few other places)? To me, it would be clearer in most places to say "density profile from the US Standard Atmosphere" and "density profiles from local radiosondes" instead of " $\beta_m(\lambda, 694)$ from the US Standard Atmosphere" and " $\beta_m(\lambda, 694)$ from the soundings"

**Reply:** The reviewer is right it should be $\beta_m(\lambda, 694)$. Corrected.

Figure 4: it would be good to have two more panels showing the difference (or the ratio) of $\beta_{aUS}(\lambda, 694)$ and $\beta_{aRadisondes}(\lambda, 694)$ (or $\alpha_{aUS}(\lambda, 694)$ and $\alpha_{aRadisondes}(\lambda, 694)$). Fig. 2 and 3 could even be dropped then.

**Reply:** We do not agree with the reviewer. The goal of the work reported was to recover an reprocess the lidar dataset to derive stratospheric aerosols extinction profiles. Including two more panels to show the differences between $\beta_{aUS}(\lambda, 694)$ and $\beta_{aRadisondes}(\lambda, 694)$ is less relevant for the purpose of the paper than showing the differences in the final result of the processing, $\alpha_{US}(\lambda, 694)$ and $\alpha_{Radisondes}(\lambda, 694)$, described in the section. On top of showing the little differences between $\alpha_{US}(\lambda, 694)$ and $\alpha_{Radisondes}(\lambda, 694)$ on both sites, they provide information not available until the present. Figure 2 provide a more detailed cross-sections of $\alpha_a(532, z)_*$ from the stratospheric aerosols after Agung than the $SR(\lambda, 694)$ bi-monthly isopleths in figure 10 of Grams and Fiocco (1967). For Fairbanks figure 3 provide the first corresponding stratospheric aerosols cross-sections of $\alpha_a(532, z)$ .

Figures 5, 8, 9: the indicator symbols at the top axes are not needed and confusing. They should be removed.

**Reply:** We do not agree with the reviewer. The indicator symbols in the cross-sections figures 5, 8 and 9 provide the time population of the measurements in the cross-sections, allowing to differentiate the time sections represented by measurements and the interpolated values.

Figure 9 contains the same information as Fig. 5, and is more comprehensive. Fig. 5 should be dropped. The entire discussion of results without correction for aerosol 2- way transmission should be shortened substantially. We know these results are poorer, and Fig. 9 shows it very clearly. There is not point in lengthy discussions of things that are obsolete or have been superseded.

**Reply:** We do not agree with the reviewer. Figures 5 and 9 are complementary. Figure 5 shows the variability in Lexington sAOD series and at the same time gives inside in the Lexington differences in magnitude and temporal scales respect to Fairbanks. Figure 9 put in context one of the key points we make in the paper: the lidar signal correction by two-way transmittance should not be dismissed, ignored or considered obsolete. In particular, there should be conducted the necessary effort to account for the tropospheric aerosols two-way transmittance correction.

After Figs. 6 and 7: It might be good to have a Figure similar to what I suggest above for Fig. 4, but showing the differences between profiles with and without correction for aerosol 2-way transmission.

**Reply:** We do not agree with the reviewer. As we stated in our reply to the comment on figure 4, the goal of the work reported was to recover and reprocess the lidar dataset to derive stratospheric aerosols extinction profiles. It is much less relevant to depict the differences in the profiles uncorrected and corrected by 2-way transmittance than depicting the cross-sections. That is more relevant yet in this case, where notable differences appears between them.

Figure 8: these results should be included in Fig. 9, and Fig. 8 dropped.

**Reply:** We do not agree with the reviewer. The set of measurements available for Lexington and Fairbanks have different extensions and in consequence represent different temporal scales of the sAOD variability. Including figure 8 in figure 9 will eliminate the short term AOD variability for Fairbanks, as happened in Figure 5.

Lines 581 to 586: This is a very complicated way of saying that the errors in the aerosol parameters blow up, when there is little aerosol, and SR is close to 1.

Lines 587 to 592: A very complicated way of saying that uncertainty due to aerosol extinction correction becomes large, when aerosol extinction becomes large.

**Reply:** We do not agree with the reviewer regarding the above 2 comments. We are discussing the error magnitudes in this particular case for both lidar series and pointing to the factors responsible of the high error levels.

Lines 597 to 618: Difficult to read, and difficult to find take-home messages.

**Reply:** The text in lines 597 to 618 was modified to make easy to read. It is now:

> *"The time vs. altitude contours of the $\left(\frac{\delta\alpha_a^{Ta}}{\alpha_a^{Ta}}\right)$ relative errors and of the $\alpha_a^{Ta}(z,n)$ are shown in figures 9 and 10 for Lexington and Fairbanks respectively. The regions with maximum magnitudes of $\alpha_a^{Ta}$ at both sites are associated with the lower relative errors as expected. At Lexington, for $\alpha_a^{Ta} > 8 \times 10^{-3}$ km$^{-1}$ the relative errors are <= 30%. It is also evident that relative errors equal or lower than 50% dominate both in time and altitude. In the case of Fairbanks, for $\alpha_a^{Ta} > 2 \times 10^{-3}$ km$^{-1}$ the relative errors are <= 40%. The relative errors of $\alpha_a^{Ta}$, in table 4, produce $\tau_a^{Ta}$ relative errors above 100%. Those estimated values of the relative errors for $\tau_a^{Ta}$ together with the ones in table 4 are substantially larger than other sets of volcanically perturbed stratospheric aerosols lidar measurements.*
>
> *The high error magnitudes in the $\left(\frac{\delta\beta_a}{\beta_a}\right)$ at 694 nm estimation could be reduced in case the $SR_o$ values increase. In several of the 75 $SR_o$ profiles a renormalization processing could*

*increase $SR_0$ magnitude. This is reasonable since the normalization altitude range (no aerosol present) was 25 to 30 km, where there certainly would be some aerosol present. Inspection of the plots of $SR_0$ vs altitude in figures 14, 15 and 16 in **G-66** shows the presence of aerosols between 25 and 30 km. And in some of the profiles $SR_0$ is above 1 at all levels (1.0 indicates no aerosol). In addition, the introdution of the two-way transmittance correction in the processing of $SR_0$, will increase $SR_0$ from the raw returned lidar signal.*
*Options are available to find the raw lidar data to conduct the reproccesing described above. These include searching for the filmed images of the oscilloscopes used as registers and/or the original punched cards (probably transferred to tapes) both reported in **G-66**. A last resort would be the digitalization of the $SR_0$ from the figures cited above. The original signal profiles could then be reconstructed inverting the normalization procedure applied to produce the $SR_0$ profiles."*

Section 3.4: Given the overall length of this data paper, and to maintain a better focus, I would suggest to drop this entire section, including Figs. 12 and 13. This is supposed to by an ESSD data paper, not an ACP paper.

**Reply:** As we explained at the start of this reply to the reviewer's comments, we strongly disagree with this comment from the reviewer. We designed this paper to include this important initial comparison to assess the magnitude of the stratospheric AOD (sAOD), including with the substantial increase when the two-way transmittance effect is resolved. We feel strongly the context from comparing to the Agung sAOD within the CMIP6 volcanic forcing dataset used in CMIP6 historical integrations, and to the ISA-MIP Agung interactive stratospheric aerosol simulations from Dhomse et al. (2020) is an important and valuable element of the manuscript.

The reviewer seems to be suggesting that ESSD papers should only focus on observational datasets, with a narrow focus only to present the dataset recovered, without comparing to other similar or complimentary datasets (whether they be observation-based or model-based data).

We strongly disagree with that suggestion, and feel the brief model-observation comparison in section 3.4 is entirely consistent with the inter-disciplinary and international remit for the ESSD journal. Whilst we agree that fully interpreting the data is out of scope for ESSD, it would seem perverse to exclude relevance of the dataset for the international ISA-MIP and CMIP6 activities.

**Other corrections:**

1. The Equation on line 444 was incorrectly numbered (18) while there was jump on the numbering jump equation (17) to (19). The numbers were reassigned after the number (17) in the order they were listed in the manuscript. A new re-assignation of equations numbers was made from equation 14 on, after the transfer of section 2.5 to the Supplement 1 eliminated equations (12) and (13) from the manuscript.

2. The decision to move the subsections in section 2.5 to Supplement 1 included eliminating figure 1 and table 2 from the manuscript. Consequently all the figures and tables were re-numbered. Also several references not cited in the new version of the manuscript were erased.

3. The term "*cross section*" was replaced by "*contours*" in the manuscript.

4. Several sentences have been rewritten:

**Lines 134-135:** The sentences

*"The lidar signal returns at both sites were registered photographically from oscilloscopes covering up to 40 km and then digitized. Then the digitized lidar return signals from a set of daily laser shots were averaged in 1 km bins (G-66; GF-67)."*

Were replaced by:

*"A single laser shot was registered by photographing the contribution of daylight return signal on an oscilloscope covering up to the 40 km, and then digitized by hand. The digitized return signals registered by the photomultipliers from a set of laser shots were then averaged in 1 km bins (G-66; GF-67)."*

**Line 231-233:** The sentence:

*"There have been abundant accounts about the changes of the physical-chemical properties of aerosols in the eastern US from the sixties until the present (Went, 1960; Husar et al., 1991)."*

Was replaced by:

*"Changes of the aerosols physical and chemical properties from the sixties until the present in the eastern US has been documented (Went, 1960; Husar et al., 1991)."*

**Line 380-381:** The sentence:

*"It is a plausible assumption because the profile βm used the US 1962 Standard Atmosphere for the vertical resolution of the lidar."*

I rewrote it:

*"The use at the lidar levels of interpolated $\beta\_m$ values from the lower resolution ones calculated using the US 1962 Standard Atmosphere, support the former assumption."*

**Lines 421-424:** The sentences

*"On top of the figures we plotted the dates the measurements were conducted (red starts at 24.5 km level). In the case of Lexington the two data gaps higher than 1 month, March and July to September both in 1964 have been left blank in the cross-sections plots. The temporal/vertical cross -section of the aerosols extinctions were generated using a linear time interpolation."*

Were replaced by:

*"The two data gaps longer than 1 month, March, and July to September both in 1964, have been left blank. The temporal/vertical contours of the aerosol extinctions were generated using a linear time interpolation."*

**Lines 435-441:** The sentences

*"Regarding the magnitudes of $\alpha_a(532, z)_{US}$ in figure 1, they are slight ly higher than the ones from $\alpha_a(532, z)_{US}$ . That is also the case in figure 3 showing the cross-sections for Fairbanks, with panels similar to figure 2. This is quantified in table 3. At both sites the mean and maximum values for $\Delta\tau_{a*}$ and $\Delta\alpha_{a*}$ are positive showing that the magnitudes of $\alpha_{aUS}$ and $\tau_{aUS}$ are in general higher than $\alpha_{a*}$ and $\tau_{a*}$. Also in the table we appreciate that the magnitudes of the mean percent difference increase of both variables is around 1%. The fact described above disagrees with the possibility* **G-66** *mentions about lower aerosol backscatter from the retrieval they conducted, using the 1962 US Standard Atmosphere, and the more realistic ones using soundings."*

Were replaced by:

*"The magnitudes of $\alpha_a(532, z)_{US}$ in are slightly higher than the ones from $\alpha_a(532, z)_*$ for both sites, and it is also true for $\tau_{a_{US}}$ and $\tau_{a*}$. This is quantified in table 2. The magnitudes of the mean percent difference increase of both variables is around 1%. This difference disagrees with G-66 where he found retrievals using the 1962 US Standard Atmosphere slightly lower than the more realistic ones using soundings, but the differences are within calculated errors."*

**Lines 451-453:** The sentences

*"The values in the denominators $M_{dUS}$ and $M_d$ are the mean values of $N_{dUS}(z)$ and $Nd(z)$ between 25 and 30 km respectively, replicating the procedure used by* **G-66***. In figure 4 the differences $\Delta N_d(z)$ for all the 66 soundings at Nantucket used to calculate $N_d(z)$ and the 9 for Fairbanks are plotted. For Lexington, on panel a), $N_{dUS}(z)$ values are both negative and positive, but higher values of $N_{dUS}(z)$ dominate."*

Were replaced by:

*"$M_{dUS}$ and $M_d$ are the mean values of $N_{dUS}(z)$ and $N_d(z)$ between 25 and 30 km, replicating the procedure used by* **G-66***. In figure 3 the differences $\Delta N_d(z)$ for 66 soundings at Nantucket and the 9 for Fairbanks are plotted. For Lexington, $\Delta N_d(z)$ values are both negative and positive, but higher values of $N_{dUS}(z)$ dominate."*

**Lines 463-469:** The sentences

*"Also figure 5 shows the monthly mean $\tau_a$ for the northern hemisphere (Sato et al., 1993). The means for the entire period of measurements available at each site are 0.0215 and 0.0099 respectively. The magnitude of the mean $\tau_{a*}$ at Fairbanks are half that of Lexington, providing evidence of the decreasing aerosol amount with increasing latitude. At the same time, some of the daily $\tau_{a*}$ values at Lexington are around the magnitude of the mean $\tau_{a*}$ at Fairbanks, because of the variability of $\alpha_a(532, z)_*$. Few $\tau_{a*}$ values from Lexington have magnitudes near the values of Sato $\tau_a$, the current reference for this period. However, as we will see in the next section a better agreement is found when the measurements are corrected by two way transmittance attenuation."*

Were replaced by:

*"The means for the entire period of measurements available at each site are 0.0215 and 0.0099 respectively. Also shown is a monthly mean $\tau_a$ for the northern hemisphere (Sato et al., 1993). The mean $\tau_{a*}$ at Fairbanks is half that of Lexington, providing evidence of the decreasing aerosol amount with increasing latitude. Because of the variability of $\alpha_a(532, z)_*$ , $\tau_{a*}$ values from Lexignton vary widely from the Fairbanks mean to the Sato magnitude, the current reference for this period. However, as we will see in the next section better agreement is found when the measurements are corrected with two-way transmittance attenuation."*

**Line 456 (Former Figure 2 caption, currently Figure 1):** The sentences

*"The red stars indicate the dates the measurements were conducted. The measurement gaps longer than 1 month, March, and July to September both in 1964, have been left blank."*

Were added at the end of the Figure 1 caption.

**Lines 522-528:** The sentences

*During the course of more than two decades after the pioneering stratospheric aerosols measurements with lidar work by Fiocco and Grams (1964) multiple researchers contributed to the development of the processing algorithms to retrieve aerosols optical properties and its errors (Russell et al, 1979, Klett, 1981; Klett, 1985, Kovalev, 2015). Those facts explain the limitations that do not allow the retrieval of the full set of optical variables characterizing the stratospheric aerosols from the Fiocco and Grams dataset. However using a Junge size-distribution model, and assuming Mie scattering with refractive index 1.5, they produced estimates of the aerosol content of the stratosphere at 16 km: number concentration, surface area and the aerosol density per unit volume of air.*

Were replaced by:

*"Since the pioneering lidar work by Fiocco and Grams (1964) multiple researchers have contributed to the development of the processing algorithms to retrieve aerosol optical properties and errors (Russell et al, 1979, Klett, 1981; Klett, 1985, Kovalev, 2015). These works explain the limitations on retrieving the full set of optical variables characterizing the stratospheric aerosols from the Fiocco and Grams dataset. However assuming a Junge size-distribution model and Mie scattering with refractive index 1.5, Fiocco and Grams did produce estimates of the aerosol content of the stratosphere at 16 km: number concentration, surface area, and the aerosol density per unit volume of air."*

**Lines 539-543:** The sentence

*"An additional validation of those results, in particular for $\tau_a^{Ta}(532, z)$ at Lexington appears in figure 9, where the stratospheric $\tau_a(532, z)$ for the northern hemisphere from January 1964 to July 1965 has been plotted (Sato et al., 1993). The magnitude of $\tau_a^{Ta}(532, z)$ is the same at Lexington (and also at Fairbanks, figure 8) as the $\tau_a(532, z)$ from Sato et al., (1993)."*

Was erased.

**Line 703-704:** The sentence

*"The search for original records should include looking for the at least 25 missing profiles from the total of at least 100 Fiocco mentions".*

Was replaced by:

*"Future search for original records should take into account also the 25 missing files from the more of 100 referred by Fiocco."*

6. Multiple words were replaced to improve and make easy to understand the manuscript. They could be seen in the manuscript with the changes not accepted.

**References**

Eyring, V., Bony, S., Meehl, G. A., Senior, C. A., Stevens, B., Stouffer, R. J., and Taylor, K. E.: Overview of the Coupled Model Intercomparison Project Phase 6 (CMIP6) experimental design and organization, *Geosci. Model Dev.*, **9**, 1937-1958, https://doi.org/10.5194/gmd-9-1937-2016, 2016.

Timmreck, C., Mann G. W., Aquila V., Hommel R., Lee L. A., Schmidt A., Brühl C., Carn S., Chin M., Dhomse S. D., Diehl T., English J. M., Mills M. J., Neely III R. R., Sheng J., Toohey M. & Weisenstein D.: The Interactive Stratospheric Aerosol Model Intercomparison Project (ISA-MIP): motivation and experimental design, Geosci. Mod. Dev., 11, 2581-2608, https://doi.org/10.5194/gmd-11-2581-2018, 2018.

Zanchettin, D., Khodri M, Timmreck C, Toohey M, Schmidt A, Gerber EP, Hegerl G, Robock A, Pausata FSR, Ball WT, Bauer SE, Bekki S, Dhomse SS, LeGrande AN, Mann, GW, Marshall L, Mills M, Marchand M, Niemeier U, Poulain V, Rozanov E, Rubino A, Stenke A, Tsigaridis K, Tummon F., The Model Intercomparison Project on the climatic response to volcanic forcing (VolMIP): experimental design and forcing input data for CMIP6. *Geosci. Mod. Dev.*, **9**(8), 2701-2719, https://doi.org/10.5194/gmd-9-2701-2016, 2016.

---

## Author Comment (AC2) · 15 Mar 2021

**Replies to Reviewers Comments:**
We thank the reviewers for the comments and suggestions they made, contributing to the improvement of the manuscript.

**Introductory statement by the authors:**
The SSIRC data rescue activity has a philosophy to involve modeling scientists as well as observational scientists, both to improve communication between the often separated communities, and also to help identify priority measurement datasets and aerosol metrics that can be of most benefit to the modelling community. Reviewer 1 has requested to remove the comparisons to model predictions dataset in section 3.4 of the manuscript, but we feel strongly this is an important element of the manuscript, highlighting why the observations dataset is of such importance both to current international climate modelling activities such as CMIP6 (Eyring et al., 2016; Zanchettin et al., 2016) and to stratospheric aerosol modelling activities such as ISA-MIP (Timmreck et al. 2018). We feel that applying strict rules to separate the publication of observational datasets and modelling datasets would in this case be in confct also with the spirit of the ESSD journal to promote international interdisciplinary research.

**Anonymous Referee #2**

This paper presents in my view a valuable contribution to the rescue of old geophysical data - in this case from early lidar measurements of stratospheric aerosols - for the sake of their use in the reconstruction of past volcanic events. The work is a contribution to the Data Rescue activity of the Stratospheric Sulfur and its Role in Climate within the SPARC project.

The crux of the work is the extraction of aerosol extinction coefficients at 532 nm between 12 km and 24 km from backscattering-ratio results at 694 nm retrieved in that range, under simplifying hypotheses, from lidar measurements carried out in different periods of 1964 at two different locations (Lexington, Massachusetts, and College, Alaska).

While the "translation" from the original results (the backscattering ratios at a given wavelength under the mentioned simplifying hypotheses) to the extracted ones (the aerosol extinction coefficients at another wavelength and correcting for the simplifications) is carefully explained, I found apparent inconsistences and ambiguities in the developed formulation, as well as in the notation, that the authors should explain or, if my concerns are proven right, correct.

A revision of the English writing and a more direct style, with less involved sentences, would probably be beneficial as well.

See attached pdf for review details.
* * *
**Review of ESSD-2020-246**

This paper presents in my view a valuable contribution to the rescue of old geophysical data - in this case from early lidar measurements of stratospheric aerosols - for the sake of their use in the reconstruction of past volcanic events. The work is a contribution to the Data Rescue activity of the Stratospheric Sulfur and its Role in Climate within the SPARC project.

The crux of the work is the extraction of aerosol extinction coefficients at 532 nm between 12 km and 24 km from backscattering-ratio results at 694 nm retrieved in that range, under simplifying hypotheses, from lidar measurements carried out in different periods of 1964 at two different locations (Lexington, Massachusetts, and College, Alaska).

While the "translation" from the original results (the backscattering ratios at a given wavelength under the mentioned simplifying hypotheses) to the extracted ones (the aerosol extinction coefficients at another wavelength and correcting for the simplifications) is carefully explained, I found apparent inconsistences and ambiguities in the developed formulation, as well as in the notation, that the authors should explain or, if my concerns are proven right, correct.

**I. The notation ambiguities are the following:**

1. The symbol $N_A$ is used with two different meanings. In Eq. (1) $N_A(z)$ is the molecular number density at altitude z. In Eq. (4) it is used for Avogadro's number.

**Reply:** We agree with the reviewer. We made the pertinent corrections: $N_A$ is only used for Avogadro's number and $N_d(z)$ for the molecular number density at altitude z.

2. Two different symbols, $SR_o(\lambda,z)$ and $SR(\lambda,z)$, are used for the backscattering ratio without a clear reason for establishing a difference. This is apparent in Eq. (6), where the $\beta_a^A(z)$ expression is said to be derived from Eq. (2), but the symbol $SR_o(\lambda,z)$ appears instead of the symbol $SR(\lambda,z)$ used in Eq. (2).

**Reply:** We agree with the reviewer. The two different symbols $SR_o(\lambda,z)$ and $SR(\lambda,z)$ could be really confusing. They were used with the purpose to differentiate the derived SR at 694 nm to the general SR at any wavelength used in the algorithm description. We replaced $SR_o(\lambda,z)$ by SR(694, z) in the manuscript.

In addition the first sentence in line 144 was modified. It reads now:

*"We first describe the procedure applied in G-66 to derive the backscattering ratio (SR(694, z))."*.

We also corrected Eq. (6) accordingly. It is now: $\beta_a^A(\lambda, z) = (SR(694, z) - 1) \beta_m(\lambda, z)$

3. The notation $T_{2w}^2$ for the two-way atmospheric transmittance is in my view redundant: the two-way is implicit in the squared superscript. The subscript should be left to denote the origin of the transmittance, as is generically done in Eq. (8). Note, related to this, that in line 180 $T_T^2$ seems to be used with the same meaning as $T_{2w}^2$.

**Reply:** We agree with the reviewer. We went further and simplified the symbol used for the two way transmittance eliminating the squared superscript but retaining the subscript to denote the origin of the transmittance.

In addition in section *"2.7.1 Backscattering ratio relative error"* we replaced $T_{2w}$ by $T_T$ to be consistent with the definition of two-way total transmittance.

4. Letting aside possible corrections arising from the reasoning in point #2 sin section III of this review below, on possible inconsistences in the formulation, I think that there are possible ambiguities in the notation that should be clarified. For example, $\beta_a^A(z)$ is used to denote the aerosol attenuated backscatter coefficient, which is subsequently corrected by the molecular and ozone transmittances in Eq. (7). The result is called $\beta_a$ which seems to imply that this is the final aerosol backscatter coefficient, from which using the factor BEc(z, t) (by the way, what does the variable t mean?) the aerosol extinction coefficient $\alpha_a$ is derived. (Eq. (9)). But then one discovers that $\alpha_a$ has still to be corrected for the aerosol transmittance, the final aerosol extinction coefficient being called $\alpha_a^{Ta}$. Although this is a minor remark, I think it would be clearer to reserve the symbols $\beta_a^{Ta}$, $\alpha_a^{Ta}$ for the coefficients yet to be corrected for the aerosol transmittance and to use $\beta_a$, $\alpha_a$ for the final, fully corrected extinction coefficient.

**Reply:** We do not agree with the reviewer. In the early times after the first lidars were operative the application of the two-way transmittance correction for the processing of stratospheric aerosol lidar returns was commonly neglected. The use of the symbols $\beta_a^{T_a}$ , $\alpha_a^{T_a}$ for the two-way transmittance corrected aerosol backscatter and aerosol extinction has the purpose to highlight the application of this correction.

Taking into account the reviewer comment on the meaning of variable t in the backscattering to extinction conversion coefficients from λ = 694 nm to λ = 532 nm (BEc(z, t)) we added *"altitude and time dependent"* on line 207 it, then that sentence is now:

*"…where BEc(z, t) are the altitude and time dependent backscattering to extinction conversion coefficients from λ = 694 nm to λ = 532 nm also derived for the Mt Pinatubo (Jäger and Deshler, 2003).*

**II. With respect to the inconsistences in the formul developments:**

1. While Eq. (1) is actually found in ref. **G-66** (Eq. (4.2) of this reference), it should be noted that this equation refers to the "expected signal from a molecular atmosphere" (page 50 of **G-66**), as it is made clear by the sentence (also in page 50 of **G-66**): "Thus, to derive the dust profiles, it is necessary to evaluate the intensity of the echoes for a dust-free atmosphere by using equation 3.8 for the case of Rayleigh scattering by air molecules."

Therefore the statement on line 144 of the paper under review referring to Eq, (1): "The average photoelectron flux registered by the photomultiplier was represented by the expression:" is misleading: the average photoelectron flux was actually represented by $\frac{dn(z)}{dt} = K\frac{\beta}{z^2}$ (Eq. (3.8) of **G-66**, with K condensing all the multiplicative constants in that equation), Eq. (1) corresponding to the expected photoelectron flux from a molecular (reference) atmosphere.

**Reply:** The reviewer is right, we made a mistake. Corrected and added an explicit reference to the Eq. in **G-66**.

2. The authors seem to imply that the scattering ratio SRo(λ,z) obtained in ref. **G-66** corresponds to the expression in Eq. (2). However, following the data processing steps described in the paper I don't arrive at that expression. I explain in detail my understanding of the steps described in the manuscript to sustain my statement. For clarity, I keep the authors' notation notwithstanding my remark #3 on the notation in section I above:

a) To begin with, the photoelectron flux will actually be, taking into account the atmospheric transmittance,

$$\frac{dn(z)}{dt} = K\frac{\beta(\lambda,z)T_{2w}^2(z)}{z^2} = K\frac{\left[\beta_m(\lambda,z)+\beta_a(\lambda,z)\right]T_{2w}^2(z)}{z^2},$$

with $\beta(\lambda,z) = \beta_m(\lambda,z) + \beta_a(\lambda,z)$ the total backscatter coefficient, being $\beta_m(\lambda,z)$ and $\beta_a(\lambda,z)$ the molecular and aerosol backscatter coefficients respectively. Note that I'm using $\beta_a(\lambda,z)$ for the "true" (i.e. not affected by any attenuation) aerosol backscatter coefficient.

b) According to the authors: "Next, the ratios between the averaged signal at each level and the values at the same level of the right side of the equation (1) were calculated for each profile between 12 and 30 km". Calling that ratio $P_n$, I understand that

$$P_n(z) = \frac{\dfrac{dn(z)}{dt}}{\dfrac{N_A(z)}{z^2}} = K\frac{\left[\beta_m(\lambda,z)+\beta_a(\lambda,z)\right]T_{2w}^2(z)}{\beta_m(\lambda,z)}$$

c) Following the authors: "A final step consisted in normalizing the ratios calculated in each profile between 12 and 24 km. To that end, for each profile the average value between 25 and 30 km of the ratios calculated in the former step were determined". I will call the average value between 25 and

30 km $\bar{P}_n(z_1, z_2)$, with $z_1 = 25$ km and $z_2 = 30$ km. Then, assuming, as **G-66** seem to do, that $\beta_a(\lambda, z)$ is negligible in that range, we would have

$$\bar{P}_n(z_1, z_2) = \frac{1}{z_2 - z_1} \int_{z_1}^{z_2} P_n(z)\, dz = \frac{K}{z_2 - z_1} \int_{z_1}^{z_2} T_{2w}^2(z)\, dz = K T_{2w}^2(z_0),$$

with, according to the finite-increment theorem, $z_0$ a range between $z_1$ and $z_2$ whose value will depend on the form of $T_{2w}^2(z)$

d) Always following the authors: "Then for each profile the ratios in the altitude range 12 and 24 km were divided by the average value of the ratios between 25 and 30 km from the same profile". Calling, as the authors seem to do, SR($\lambda$,z) (or SR$_0$($\lambda$,z), see my remark #2 on the notation ambiguities) the result of dividing the ratio $P_n$. in the range 12 km – 24 km by $\bar{P}_n(z_1, z_2)$. Then

$$SR(\lambda, z) = \frac{P_n(z)}{\bar{P}_n(z_1, z_2)} = \frac{\left[\beta_m(\lambda, z) + \beta_a(\lambda, z)\right] T_{2w}^2(z)}{\beta_m(\lambda, z) T_{2w}^2(z_0)},$$

which do not coincide with the result given in Eq. (2).
The authors should, either point out possible mistakes I may have committed in the above development, or else correct theirs.
**Reply**: The derivation conducted by the reviewed is right, no mistakes. The 2 two-way transmission terms are part or the exact definition of SR($\lambda$,z) with no assumptions about the two-way transmission. However, the sentence before Eq. (2) states:

> *"That definition is associated to the fact that in the retrieval of SRo (z) the two-way total transmittance $T_T^2(z)$ correction was neglected (Hostetler et al., 2006)".*

The neglected two-way transmittance implies $T_{2w}^2(z) = 1$, ending in the formulation in Eq. (2). We took into account the fact that in **G-66** SRo(694,z) was derived with similar assumption.

4. I couldn't understand the (iterative?) process described in lines 211 – 217 to obtain $\alpha_a^{Ta}$. How is the $T_a^2(532, z)$ first guess of obtained? How is it refined? I suggest illustrating the procedure with a graph.
**Reply**: We agree with the reviewer the way it has been described is complicated to understand. We rewrote it. It is now:

> *"Because the information available to calculate $T_a(532, z)$ should be determined using the total aerosol optical depth (TAOD) measurements from sun photometers we applied a two-step procedure. The first step consists of using the TAOD to calculate a first guess $T_a(532)$, which is a unique value for all the altitudes. It is follow by calculation of a first guess $\alpha_a^{Ta}(532, z)_*$ profile. Then the stratospheric AOD (sAOD) is calculated integrating $\alpha_a^{Ta}(532, z)_*$ between 12 and 24 km. The second step calculates (see Suplement-1 for details on the calculations of TAOD):*
>
> $$tAOD = TAOD - sAOD \qquad (11)$$
>
> *producing a profile of $T_a(532, z)$ with the particularity of having a constant value of $T_a(532)$ from the surface to 11 km and then a profile of $T_a(532, z)$ between 12 and 24 km. This profile of $T_a(532, z)$ is applied in equation (11) getting the definitive values of $\alpha_a^{Ta}(532, z)$."*

**III. Other remarks**

1. Line 77: "were been produce" should be just "were produced".
**Reply**: Corrected
2. Line 80: "Giogio Fiocco" should be "Giorgio Fiocco".
**Reply**: Corrected
3. Line 86: What do CMIP5 and CMIP6 refer to? Further explanations and possibly a reference are needed.
**Reply**: Defined CMIP5 and CMIP6 (Coupled Model Intercomparison Projects 5 and 6) and the respective references added:

Taylor, K. E., Stouffer, R. J., and Meehl, G. A.: An overview of CMIP5 and the experiment design, B. Am. Meteorol. Soc., 93, 485–498, 2012.

Eyring, V., Bony, S., Meehl, G. A., Senior, C. A., Stevens, B., Stouffer, R. J., and Taylor, K. E.: Overview of the Coupled Model Intercomparison Project Phase 6 (CMIP6) experimental design and organization, Geosci. Model Dev., 9, 1937-1958, https://doi.org/10.5194/gmd-9-1937-2016, 2016.

Zanchettin, D., Khodri M, Timmreck C, Toohey M, Schmidt A, Gerber EP, Hegerl G, Robock A, Pausata FSR, Ball WT, Bauer SE, Bekki S, Dhomse SS, LeGrande AN, Mann, GW, Marshall L, Mills M, Marchand M, Niemeier U, Poulain V, Rozanov E, Rubino A, Stenke A, Tsigaridis K, Tummon F., The Model Intercomparison Project on the climatic response to volcanic forcing (VolMIP): experimental design and forcing input data for CMIP6. *Geosci. Mod. Dev.*, **9**(8), 2701-2719, https://doi.org/10.5194/gmd-9-2701-2016, 2016.

4. Line 137: "It is well known that solving the equation for the single wavelength elastic lidar is an ill-posed problem". What is the equation the sentence refers to? I suppose it is the lidar equation, so the sentence should read "It is well known that solving the lidar equation…" As implied in remark # 1 of section II of this review, this should be Eq. (1), but written in the form of Eq. (3.8) of G-66.

**Reply**: The reviewer is right. Corrected.

5. Line 105: the description "very high frequency nano-second laser" is misleading, as it seems to mix the laser pulse duration with the pulse repetition rate, the latter being very low, according to table 1, by today's standards.

**Reply**: Corrected. Replaced "very high frequency nano-second laser" by "a short pulse (micro-second)".

6. I think it would be beneficial for the reader to point out that $\frac{dn(z)}{dt}$ is proportional to the collected backscattered power from that range z.

**Reply**: We agree with the reviewer suggestion. Added.

7. Line 146: $n_z$ should probably be n(z).

**Reply**: The reviewer is correct. Corrected.

8. The references Jäger and Deshler, 2002 and 2003 are missing in the reference list. Please check that all cited references are included in the list.

**Reply**: Corrected.

9. Line 181: "Hosteler" should be "Hostetler"

**Reply**: Corrected.

10. In lines 186-187 it is said: "after neglecting the dispersion of the refractive index and the King factor of the air represented by kbw". If the King factor is not used, it is unnecessary to assign it a symbol.

**Reply**: WE are not assigning the King Factor a symbol. The expression Sm = (8π/3)kbw defining the molecular extinction to backscatter ratio on line 186, contains it, what we do is defining a symbol we used.

11. Unnecessary bracketing is used sometimes. For example, but not only, Eq. (2) could simply be written as

$$SR(\lambda, z) = \frac{\beta_m(\lambda, z) + \beta_a^A(\lambda, z)}{\beta_m(\lambda, z)},$$

without the curly brackets. Other instances of unnecessary brackets are found throughout the paper.

**Reply**: Corrected. Eliminated unnecessary bracketing in equations 2, 15 and 17

12. Lines 321-322: "After the conversion to 532 nm they were respectively 0.087 and 0.242…" What does "they" in this sentence refers to?

**Reply**: The section containing the text has been moved to the Supplement 1 (attached). There the sentence has been modified. It is now:

> *"After the conversion to 532 nm they were respectively 0.087 from Fairbanks, AK, in July 1978 (Shaw, 1982) and 0.242 from the July 1997- 2019 climatology at Bonanza Creek, AK. We used the contemporary July 1978 value for both July and August 1964 lidar measurements at Fairbanks."*

13. Line 326: "The high TAOD values for the twos series…" Which ones are the two series?

**Reply**: The section containing the text has been moved to the Supplement 1 (attached). There the sentence has been modified. It is now:

> *"The high TAOD values of the two series, from the Blue Hill Observatory (1961-1966) and the Eastern US (1972-1975), are representative of what have been reported for the Eastern US (Husar et al., 1991)."*

14. Lines 367-368: "the data we rescue are a reasonable approximation of what we today know as the backscattering ratio described in equation (2)". But the definition in Eq. (2), letting aside the concerns expressed in point #2 of section II of this review as to its derivation, uses the attenuated backscatter, while, in my understanding, the present-day backscattering ratio definition is $SR = \dfrac{\beta_m + \beta_a}{\beta_m}$,

with $\beta_m$ the molecular backscatter coefficient and a $\beta_a$ the "true", not affected by any attenuation, aerosol backscatter coefficient.

**Reply**: We agree with the reviewer. We eliminated that the sentence.

**Other corrections:**

1. The Equation on line 444 was incorrectly numbered (18) while there was jump on the numbering jump equation (17) to (19). The numbers were reassigned after the number (17) in the order they were listed in the manuscript. A new re-assignation of equations numbers was made from equation 14 on, after the transfer of section 2.5 to the Supplement 1 eliminated equations (12) and (13) from the manuscript.

2. The decision to move the subsections in section 2.5 to Supplement 1 included eliminating figure 1 and table 2 from the manuscript. Consequently all the figures and tables were re-numbered. Also several references not cited in the new version of the manuscript were erased.

3. The term "*cross section*" was replaced by "*contours*" in the manuscript.

4. Several sentences have been rewritten:

**Lines 134-135:** The sentences

> *"The lidar signal returns at both sites were registered photographically from oscilloscopes covering up to 40 km and then digitized. Then the digitized lidar return signals from a set of daily laser shots were averaged in 1 km bins (G-66; GF-67)."*

Were replaced by:

> *"A single laser shot was registered by photographing the contribution of daylight return signal on an oscilloscope covering up to the40 km, and then digitized by hand. The digitized return signals registered by the photomultipliers from a set of laser shots were then averaged in 1 km bins (G-66; GF-67)."*

**Line 231-233:** The sentence:

> *"There have been abundant accounts about the changes of the physical-chemical properties of aerosols in the eastern US from the sixties until the present (Went, 1960; Husar et al., 1991)."*

Was replaced by:

> *"Changes of the aerosols physical and chemical properties from the sixties until the present in the eastern US has been documented (Went, 1960; Husar et al., 1991)."*

**Line 380-381:** The sentence:

*"It is a plausible assumption because the profile βm used the US 1962 Standard Atmosphere for the vertical resolution of the lidar."*

I rewrote it:

*"The use at the lidar levels of interpolated β_m values from the lower resolution ones calculated using the US 1962 Standard Atmosphere, support the former assumption."*

**Lines 421-424:** The sentences

*"On top of the figures we plotted the dates the measurements were conducted (red starts at 24.5 km level). In the case of Lexington the two data gaps higher than 1 month, March and July to September both in 1964 have been left blank in the cross-sections plots. The temporal/vertical cross -section of the aerosols extinctions were generated using a linear time interpolation."*

Were replaced by:

*"The two data gaps longer than 1 month, March, and July to September both in 1964, have been left blank. The temporal/vertical contours of the aerosol extinctions were generated using a linear time interpolation."*

**Lines 435-441:** The sentences

*"Regarding the magnitudes of $\alpha_a(532, z)_{US}$ in figure 1, they are slight ly higher than the ones from $\alpha_a(532, z)_{US}$. That is also the case in figure 3 showing the cross-sections for Fairbanks, with panels similar to figure 2. This is quantified in table 3. At both sites the mean and maximum values for $\Delta\tau_{a*}$ and $\Delta\alpha_{a*}$ are positive showing that the magnitudes of $\alpha_{aUS}$ and $\tau_{aUS}$ are in general higher than $\alpha_{a*}$ and $\tau_{a*}$. Also in the table we appreciate that the magnitudes of the mean percent difference increase of both variables is around 1%. The fact described above disagrees with the possibility **G-66** mentions about lower aerosol backscatter from the retrieval they conducted, using the 1962 US Standard Atmosphere, and the more realistic ones using soundings."*

Were replaced by:

*"The magnitudes of $\alpha_a(532, z)_{US}$ in are slightly higher than the ones from $\alpha_a(532, z)_*$ for both sites, and it is also true for $\tau_{a_{US}}$ and $\tau_{a*}$. This is quantified in table 2. The magnitudes of the mean percent difference increase of both variables is around 1%. This difference disagrees with G-66 where he found retrievals using the 1962 US Standard Atmosphere slightly lower than the more realistic ones using soundings, but the differences are within calculated errors."*

**Lines 451-453:** The sentences

*"The values in the denominators $M_{dUS}$ and $M_d$ are the mean values of $N_{dUS}(z)$ and $Nd(z)$ between 25 and 30 km respectively, replicating the procedure used by **G-66**. In figure 4 the differences $\Delta N_d(z)$ for all the 66 soundings at Nantucket used to calculate $N_d(z)$ and the 9 for Fairbanks are plotted. For Lexington, on panel a), $N_{dUS}(z)$ values are both negative and positive, but higher values of $N_{dUS}(z)$ dominate."*

Were replaced by:

*"$M_{dUS}$ and $M_d$ are the mean values of $N_{dUS}(z)$ and $N_d(z)$ between 25 and 30 km, replicating the procedure used by **G-66**. In figure 3 the differences $\Delta N_d(z)$ for 66 soundings at Nantucket and the 9 for Fairbanks are plotted. For Lexington, $\Delta N_d(z)$ values are both negative and positive, but higher values of $N_{dUS}(z)$ dominate."*

**Lines 463-469:** The sentences

*"Also figure 5 shows the monthly mean $\tau_a$ for the northern hemisphere (Sato et al., 1993). The means for the entire period of measurements available at each site are 0.0215 and 0.0099 respectively. The magnitude of the mean $\tau_{a*}$ at Fairbanks are half that of Lexington, providing evidence of the decreasing aerosol amount with increasing latitude. At the same time, some of the daily $\tau_{a*}$ values at Lexington are around the magnitude of the mean $\tau_{a*}$ at Fairbanks, because of the variability of $\alpha_a(532, z)_*$. Few $\tau_{a*}$ values from Lexington have*

*magnitudes near the values of Sato $\tau_a$, the current reference for this period. However, as we will see in the next section a better agreement is found when the measurements are corrected by two way transmittance attenuation."*

Were replaced by:

*"The means for the entire period of measurements available at each site are 0.0215 and 0.0099 respectively. Also shown is a monthly mean $\tau_a$ for the northern hemisphere (Sato et al., 1993). The mean $\tau_{a*}$ at Fairbanks is half that of Lexington, providing evidence of the decreasing aerosol amount with increasing latitude. Because of the variability of $\alpha_a(532, z)_*$ , $\tau_{a*}$ values from Lexignton vary widely from the Fairbanks mean to the Sato magnitude, the current reference for this period. However, as we will see in the next section better agreement is found when the measurements are corrected with two-way transmittance attenuation."*

**Line 456 (Former Figure 2 caption, currently Figure 1):** The sentences

*"The red stars indicate the dates the measurements were conducted. The measurement gaps longer than 1 month, March, and July to September both in 1964, have been left blank."*

Were added at the end of the Figure 1 caption.

**Lines 522-528:** The sentences

*During the course of more than two decades after the pioneering stratospheric aerosols measurements with lidar work by Fiocco and Grams (1964) multiple researchers contributed to the development of the processing algorithms to retrieve aerosols optical properties and its errors (Russell et al, 1979, Klett, 1981; Klett, 1985, Kovalev, 2015). Those facts explain the limitations that do not allow the retrieval of the full set of optical variables characterizing the stratospheric aerosols from the Fiocco and Grams dataset. However using a Junge size -distribution model, and assuming Mie scattering with refractive index 1.5, they produced estimates of the aerosol content of the stratosphere at 16 km: number concentration, surface area and the aerosol density per unit volume of air.*

Were replaced by:

*"Since the pioneering lidar work by Fiocco and Grams (1964) multiple researchers have contributed to the development of the processing algorithms to retrieve aerosol optical properties and errors (Russell et al, 1979, Klett, 1981; Klett, 1985, Kovalev, 2015). These works explain the limitations on retrieving the full set of optical variables characterizing the stratospheric aerosols from the Fiocco and Grams dataset. However assuming a Junge size-distribution model and Mie scattering with refractive index 1.5, Fiocco and Grams did produce estimates of the aerosol content of the stratosphere at 16 km: number concentration, surface area, and the aerosol density per unit volume of air."*

**Lines 539-543:** The sentence

*"An additional validation of those results, in particular for $\tau_a^{Ta}(532, z)$ at Lexington appears in figure 9, where the stratospheric $\tau_a(532, z)$ for the northern hemisphere from January 1964 to July 1965 has been plotted (Sato et al., 1993). The magnitude of $\tau_a^{Ta}(532, z)$ is the same at Lexington (and also at Fairbanks, figure 8) as the $\tau_a(532, z)$ from Sato et al., (1993)."*

Was erased.

**Line 703-704:** The sentence

*"The search for original records should include looking for the at least 25 missing profiles from the total of at least 100 Fiocco mentions".*

Was replaced by:

*"Future search for original records should take into account also the 25 missing files from the more of 100 referred by Fiocco."*

6. Multiple words were replaced to improve and make easy to understand the manuscript. They could be seen in the manuscript with the changes not accepted.

**References**

Eyring, V., Bony, S., Meehl, G. A., Senior, C. A., Stevens, B., Stouffer, R. J., and Taylor, K. E.: Overview of the Coupled Model Intercomparison Project Phase 6 (CMIP6) experimental design and organization, *Geosci. Model Dev.*, **9**, 1937-1958, https://doi.org/10.5194/gmd-9-1937-2016, 2016.

Timmreck, C., Mann G. W., Aquila V., Hommel R., Lee L. A., Schmidt A., Brühl C., Carn S., Chin M., Dhomse S. D., Diehl T., English J. M., Mills M. J., Neely III R. R., Sheng J., Toohey M. & Weisenstein D.: The Interactive Stratospheric Aerosol Model Intercomparison Project (ISA-MIP): motivation and experimental design, Geosci. Mod. Dev., 11, 2581-2608, https://doi.org/10.5194/gmd-11-2581-2018, 2018.

Zanchettin, D., Khodri M, Timmreck C, Toohey M, Schmidt A, Gerber EP, Hegerl G, Robock A, Pausata FSR, Ball WT, Bauer SE, Bekki S, Dhomse SS, LeGrande AN, Mann, GW, Marshall L, Mills M, Marchand M, Niemeier U, Poulain V, Rozanov E, Rubino A, Stenke A, Tsigaridis K, Tummon F., The Model Intercomparison Project on the climatic response to volcanic forcing (VolMIP): experimental design and forcing input data for CMIP6. *Geosci. Mod. Dev.*, **9**(8), 2701-2719, https://doi.org/10.5194/gmd-9-2701-2016, 2016.

---

## Referee Report (RR1)

**Review of manuscript ESSD-2020-246 (1[st] revised version)**

The paper has been streamlined, taking some parts to a supplement, some errors have been corrected and, overall, it is more readable now. However, I still have some strong concerns, especially in the description of how the correction for the stratospheric aerosol transmittance (neglected in the dataset under rescue) is carried out (section 2.4), which, in my view, has not been clarified in the revision.

**Major concerns**

1. Lines 175-176: the authors state that "The lidar backscattering ratio ($SR(\lambda, z)$) is commonly defined as the ratio between the total backscatter ($\beta_T(\lambda, z)$) and the molecular backscatter $\beta_m(\lambda, z)$, at the altitude z and wavelength $\lambda$", with which I agree. However I find again the next sentences, namely

"$\beta_T(\lambda, z)$ is the sum of $\beta_m(\lambda, z)$ and the aerosol attenuated backscatter ($\beta_a^A(\lambda,z)$). That definition is related to the fact that in the retrieval of $SR_0(z)$ the two-way total transmittance($T_T^2$) correction was neglected (Hostetler et al., 2006)"

most confusing. That cannot be the definition of the backscattering ratio, where the aerosol backscatter coefficient should be the "true" one, not the aerosol attenuated backscatter coefficient. I could accept that Eq. (2) is what the author of **G-66**, obtained because of his neglecting of stratospheric transmittance, but not as the common definition. I think a difference should be clearly made between this scattering ratio with $\beta_a$ non-corrected for atmospheric transmission and the corrected one (and transposed to another wavelength in addition), which, in my understanding, is what the authors aim to produce. The authors seem implicitly recognize that in their reply to my first review when they say "The 2 two-way transmission terms are part or the exact definition of $SR(\lambda,z)$ with no assumptions about the two-way transmission". Furthermore I also find the reference to Hostetler el al., 2016 (dealing with CALIPSO inversion algorithm) misleading, as it does not support the neglect (at least I don't see how) of the two-way total transmittance in **G-66**.

2. In addition, I still cannot understand how the sunphotometer-derived AOD is used to correct for the stratospheric AOD. It would seem that an AOD-constrained inversion is attempted but it is not clear how it is achieved:

      2.1. What does "sup" (the lower limit of the integral in the exponential in Eq. (8)) denote?

      2.2. I have several problems with the first guess $T_a(532)$ mentioned in line 210:
          a) How is it chosen?
          b) It is stated it is "a unique value for all altitudes" (line 210). What does "unique" mean in this context?
          c) The z-dependence seems to have disappeared. That would imply that $\alpha_a^{Ta}(532,z) = 0$.

      2.3. Likewise, how is the first-guess $\alpha_a^{Ta}(532,z)_*$ chosen?

2.4. What does mean that the profile of $T_a(532,z)$ has a constant value of $T_a(532)$ from the surface to 11 km? Again, if $T_a$ is constant it means that the extinction coefficient is 0 (Eq. (8)). I suspect that the authors mean that they don't care about the profile of $T_a$ between the surface and the 11 km height and that they consider only the *value* of $T_a$ between the surface and that height; but in any case, the way they express it is, in my understanding, formally wrong.

2.5. For the same reason, I suspect the $T_a(532,z)$ profile between 12 and 24 km mentioned in line 215 might not correspond to the two-way transmittance between the surface and height z, but between 12 km and the height z. The authors might have a problem with their notation, failing to indicate the limits between which the atmospheric transmittance is calculated. Note that the definition of the transmittance (Eq. (8)) involves a definite integral, so the limits should be indicated, unless one of them is always conventionally the same, which I think might not the case throughout the paper.

I would tend to agree with the other reviewer's remark in his/her first review that an iterative procedure would be in order. From the authors reply to that remark (not very clear anyway to me) it seems they justify that only one (or two?) iterations would suffice, because there are other sources of uncertainty overshadowing that one. But this should be reflected in the paper text and the way of performing the aerosol correction should be clearly stated, in a way that could be reproduced by a reader. In the present form of the explanation, I would be unable to do it.

3. Lines 203-204: the authors say, referring to Eq. (9) that "$EB_c(z, t)$ are the altitude and time dependent backscattering to extinction conversion coefficients from $\lambda = 694$ nm to $\lambda = 532$ nm" but only "532" appears on both sides of Eq. (2), and the reference given (Jäger and Deshler, 2003) shows that they actually convert from backscatter to extinction at 532 nm. The conversion from 694 to 532 nm has been made in Eq. (7) through the $kb(z,t)$ exponent.

4. If $EB_c(z,t)$ (and $kb(z,t)$ by the way), are time dependent, how and where is the time dependence taken into account? I suspect it is used to calculate the $\beta_a$ and $\alpha_a$ uncertainties (Eqs. (20) and (21)), but this should be stated the first time the coefficients appear. Otherwise it is misleading, because the left-hand sides of Eqs. (7) and (9) should have a time dependence. Which are the "nominal" values used for $kb(z,t)$ and $EB_c(z,t)$, around which their uncertainties have been used to calculate the $\beta_a$ and $\alpha_a$ uncertainties in Eqs. (20) and (21)?

**Other concerns**

1. Line 143: it would be helpful to explain that $SR(694,z)$ means the backscattering ratio at 694 nm and range z.

2. Line after Eq. (1): while $\dfrac{dn(z)}{dt}$ is the photoelectron flux (electrons/s) resulting from scattering at range z, defining n(z) as "the number of photons at the altitude z" does not make

much sense. In the explanation of Eq. (1), leave $\dfrac{dn(z)}{dt}$ as the photoelectron flux (electrons/s) resulting from the photons scattered at range z.

3. The authors say that they have eliminated the squared superscript to indicate the two-way transmittance, but this has not been done consistently. There are examples of the superscript still remaining (for example, but not only, in lines 149,150 and 177).

4. There is still one instance (line 177) where SRo(z) is used.

5. Line 215: "This profile of Ta(532,z) is applied in equation (11)". Equation (10) is probably meant.

---

## Author Response (AR2)

**Reply to Anonymous Reviewer 2's review of revised manuscript**

Anonymous Reviewer 2 has reviewed our 1[st] revised version of the manuscript, finding "Good" for all of the 6 sections of the review, except for "Presentation quality", which they have graded "Fair".

The reviewer has then made recommendation of "reconsider still requiring major revisions" with:

 "The authors should correct what in my view are inconsistencies and explain better how the aerosol transmittance correction is performed to retrieve the "true" scattering ratio."

In the remainder of this response (see below), we have replied to each of the Reviewer 2's comments, showing the reviewer's comments in black (Times font), and our replies to these in dark-red colour (Arial font). Where excerpts of text are shown from the original or revised manuscripts, these are shown in italics.

**Review of manuscript ESSD-2020-246 (1[st] revised version)**

The paper has been streamlined, taking some parts to a supplement, some errors have been corrected and, overall, it is more readable now. However, I still have some strong concerns, especially in the description of how the correction for the stratospheric aerosol transmittance (neglected in the dataset under rescue) is carried out (section 2.4), which, in my view, has not been clarified in the revision.

**Major concerns**

Lines 175-176: the authors state that *"The lidar backscattering ratio (SR($\lambda$, z)) is commonly defined as the ratio between the total backscatter ($\beta_T(\lambda$, z)) and the molecular backscatter $\beta_m(\lambda$, z), at the altitude z and wavelength $\lambda$"*, with which I agree.

However I find again the next sentences, namely *"$\beta_T(\lambda$, z) is the sum of $\beta_m(\lambda$, z) and the aerosol attenuated backscatter ($\beta_a^A(\lambda,z)$). That definition is related to the fact that in the retrieval of SRo(z) the two-way total transmittance($T^2$) correction was neglected (Hostetler et al., 2006)"* most confusing.

That cannot be the definition of the backscattering ratio, where the aerosol backscatter coefficient should be the "true" one, not the aerosol attenuated backscatter coefficient. I could accept that Eq. (2) is what the author of **G-66**, obtained because of his neglecting of stratospheric transmittance, but not as the common definition.

The reviewer is right, and that was a typo in that 2[nd] sentence. Sorry for that.. We have corrected the typo (line 176 of the revised manuscript), replacing:

*"…aerosol attenuated backscatter ($\beta_a^A(\lambda,z)$)."*     with      *"…the aerosol backscatter ($\beta_a(\lambda,z)$)."*

I think a difference should be clearly made between this scattering ratio with $\beta_a$ non-corrected for atmospheric transmission and the corrected one (and transposed to another wavelength in addition), which, in my understanding, is what the authors aim to produce.

Yes, the reviewer is understanding correctly – our aim is to explain each step of the data processing methodology.

We have clarified the text to make it clearer where we are describing our specific recovery methodology, and where we are making a more general statement about a particular metric or measured quantity.

The authors seem implicitly recognize that in their reply to my first review when they say "The 2 two-way transmission terms are part or the exact definition of SR($\lambda$,z) with no assumptions about the two-way transmission".

Yes, that's correct – we were trying to explain the method, but that was a typo, now corrected.

Furthermore I also find the reference to Hostetler el al., 2016 (dealing with CALIPSO inversion algorithm) misleading, as it does not support the neglect (at least I don't see how) of the two-way total transmittance in **G-66**.

Yes, we agree that follow-on sentence was confusing. Sorry. We have deleted it in the revised manuscript:

In summary, we agree with the reviewer's main point that the wording of this part of the revised manuscript required still some improvement.  We have improved the text to now introduce the total and aerosol attenuated backscatter after

equation 5 (lines 190 and 210 of the revised manuscript). As the reviewer hints, our aim is to be fully transparent in the method to keep the aerosol backscatter and aerosol extinction uncorrected by two-way aerosol transmittance in the initial derivation (at the measured wavelength of 694nm), correcting for aerosol transmittance only after conversion to the target wavelength of 532nm. The reason for postponing the aerosol transmittance correction is that the Blue Hill and Fairbanks sun photometer TAOD measurements were at 500nm wavelength (see supplementary material), much closer to the 532nm wavelength than 694nm. Note also that the order of the equations 7 and 8 has been changed in the revised manuscript in this improvement to the explanation of the methodology.

In summary, we have improved this part of the revised manuscript to address the reviewer's valid concerns, and have copied and pasted this revised text for lines 190-210 below for maximum clarity:

"*The single-wavelength elastic lidar systems provide profiles of attenuated total backscatter. The "true" total backscatter is calculated from $\beta_T(\lambda, z) = \beta_T^A(\lambda, z)\ T_j(\lambda, z)$, where $\beta_T^A(\lambda, z) = [\beta_m^A(\lambda, z) + \beta_a^A(\lambda, z)]$. Substituting into Eq.(2), taking into account that $\beta_m(\lambda, z) = \beta_m^A(\lambda, z)\ T_j(\lambda, z)$ and re-arranging specifically for this case of the 694nm backscattering ratio, gives:*

$$\beta_a^A(\lambda, z) = \frac{[\ SR(694, z) - 1]\ \beta_m(\lambda, z)}{T_j(\lambda, z)} = \frac{[\ SR(694, z) - 1]\ \beta_m(\lambda, z)}{T_m(\lambda, z)\ T_{O3}(\lambda, z)} \qquad (6)$$

*Whereas the term $T_j(\lambda, z)$ in equation 6 usually specifies the full two-way transmittance correction, at this stage of the processing methodology, we correct only for the attenuation due to molecular backscatter and ozone absorption:*

$$T_j(\lambda, z) = e^{-2\int_{sup}^{z} \alpha_j(\lambda, z)dz} \qquad (7).$$

*The $\alpha_j(\lambda, z)$ term is thus the vertical profile of 694nm extinction due only to molecular scattering ($\alpha_m(\lambda, z)$) and ozone absorption ($\alpha_{O3}(\lambda, z)$), with $z_0$ being the altitude of the lidar. This postponement of the aerosol attenuation correction is due to the 500nm wavelength of the contemporaneous AOD measurements being much closer to the target wavelength of 532nm (see Supplementary Material), the method preserving the two-way molecular and ozone transmittance corrections to be applied at the measurement wavelength of 694 nm.*

*The conversion to aerosol extinction was carried out after first translating the aerosol backscatter from 694nm to 532nm, applying the corresponding wavelength exponent kb(z,t), calculated from in-situ size distribution measurements of the Northern Hemisphere mid-latitude Pinatubo aerosol cloud (Jaeger and Deshler, 2002; 2003):*

$$\beta_a(532, z) = \left[\frac{532}{694}\right]^{kb(z,t)} \beta_a(694, z). \qquad (8)$$

*Note again that the derived 532nm aerosol backscatter ($\beta_a(532, z)$) has at this point only been corrected for two-way molecular scattering and ozone absorption transmittance effects.*

*The aerosol extinction, $\alpha_a(532, z)$, at this point still uncorrected for two-way aerosol transmittance, is then calculated by the expression:*

$$\alpha_a(532, z) = EB_c(z, t)\ \beta_a(532, z) \qquad (9)$$

*where EB_c(z, t) are altitude- and time-dependent coefficients to convert from aerosol backscatter to aerosol extinction (at λ = 532 nm), derived from the same Pinatubo aerosol size distribution measurements (Jäger and Deshler, 2003).*

*Both the EBc and kb factors are derived from log-normal size distribution fits to balloon-borne optical particle counter measurements of the Pinatubo aerosol cloud from Laramie, Wyoming, USA. Each of the conversion factors in Jäger and Deshler (2002; 2003) represent averages over 4 height ranges: tropopause–15, 15–20, 20–25, and 25–30 km, and are provided for the 4-month periods November–February, March–June, and July–October of each year after the eruption. We used EBc(z,t) and kb(z, t) for the equivalent 4-month periods after the March 1963 Agung eruption, based on matching the same time-offset after the June 1991 Pinatubo eruption.*

2. In addition, I still cannot understand how the sunphotometer-derived AOD is used to correct for the stratospheric AOD. It would seem that an AOD-constrained inversion is attempted but it is not clear how it is achieved:

The method we applied uses monthly means of the total AOD (TAOD) from the sunphotometer, converted from 500 to 532 nm (described above) assuming this to be tropospheric (tAOD) in the first step, to produce for each site a $T_a(532, z)_*$

profile, so as to account for the tropospheric aerosol transmittance from the lidar altitude across the troposphere up to 11 km and the stratospheric aerosol transmittance in the lower stratosphere from 12 to 24 km. However, when the stratospheric AOD ($sAOD_*$) is calculated in the next step (from the first guess $\alpha_a^{Ta}(532, z)_*$ ), the resulting first guess total AOD ($tAOD + sAOD_*$) will be higher than the observed TAOD. The second step is aimed to estimate the magnitude of a consistent value for tAOD for each measurement to constrain $tAOD + sAOD \approx TAOD$.

**2.2.1 What does "sup" (the lower limit of the integral in the exponential in Eq. (8)) denote?**

In the original and 1$^{st}$-revised manuscripts, we used the term "sup" to denote the altitude of the lidar site. On reflection, perhaps that was not so clear, and in this 2$^{nd}$-revised manuscript we have changed all instances of "sup" instead to "$z_o$". We also added in the manuscript, at the end of the sentence on line 197:

"… and $z_o$ being the altitude of the lidar."

**2.2.2 I have several problems with the first guess $T_a(532)$ mentioned in line 210:**

**a) How is it chosen?**

We produced a first guess $T_a(532, z)_*$ for each measurement day at each site, in the range from 11 to 24 km. The $T_a(532, 11km)_*$, a unique value from the lidar altitude to 11 km, was calculated using Eq.(8) and the TAOD(532) value for the month the measurement was conducted. $T_a(532, z)_*$ values from 12 to 24 km were calculated for each level using Eq.(8) and the uncorrected $\alpha_a(532, z)$.

**b) It is stated it is "a unique value for all altitudes" (line 210). What does "unique" mean in this context?**

By unique, we mean the same value is used for all the altitudes from the lidar up to 11 km.

**c) The z-dependence seems to have disappeared. That would imply that $\alpha_a^{Ta}(532, z) = 0$**

There is no z dependence below 11 km, but we do the calculation for each level at 12km and above, so there is a z-dependence in the main region of interest, from 12 to 24 km.

**2.3 Likewise, how is the first-guess $\alpha_a^{Ta}(532, z)$**

The first guess ($\alpha_a^{Ta}(532, z)_*$ ), was derived applying the correction for the first guess two-way aerosol transmittance $T_a(532, z)_*$ in Eq.(10).

**2.4. What does mean that the profile of $T_a(532,z)$ has a constant value of $T_a(532)$ from the surface to 11 km? Again, if $T_a$ is constant it means that the extinction coefficient is 0 (Eq. (8)). I suspect that the authors mean that they don't care about the profile of $T_a$ between the surface and the 11 km height and that they consider only the *value* of $T_a$ between the surface and that height; but in any case, the way they express it is, in my understanding, formally wrong.**

Apologies, we may have used the term "constant" incorrectly there, perhaps we should have said "representative value" or similar. As we explained above we used the same unique/representative value from the lidar altitude to 11 km.

**2.5. For the same reason, I suspect the Ta(532, z) profile between 12 and 24 km mentioned in line 215 might not correspond to the two-way transmittance between the surface and height z, but between 12 km and the height z. The authors might have a problem with their notation, failing to indicate the limits between which the atmospheric transmittance is calculated. Note that the definition of the transmittance (Eq. (8)) involves a definite integral, so the limits should be indicated, unless one of them is always conventionally the same, which I think might not the case throughout the paper.**

No, the method is different between 12km and 24km. For that main region of interest, Ta(532,z) was calculated at each level z, to account for the corresponding two-way transmittance between the surface and height z.

I would tend to agree with the other reviewer's remark in his/her first review that an iterative procedure would be in order. From the authors reply to that remark (not very clear anyway to me) it seems they justify that only one (or two?) iterations would suffice, because there are other sources of uncertainty overshadowing that one. But this should be reflected in the paper text and the way of performing the aerosol correction should be clearly stated, in a way that could be reproduced by a reader. In the present form of the explanation, I would be unable to do it.

We agree with the reviewer. We added the sentence at the end of section 2.4:

*"Although more iteration of those final steps would be possible, with the high magnitude of the estimated $\alpha_a^{Ta}(532,z)$ mean error, around 60%, compared to an estimated 15-20% maximum improvement achieved by the iteration procedure, we do not believe those additional calculations would be worthwhile in this case."*

We modified the text between the lines 206 and 231. We moved the 2 paragraphs from lines 217 to 230, inserting them in line 206. And several parts of the text where rewritten taking in consideration the reviewer's comments in this point 2. The revised text now reads:

*"The algorithms for the solution of the single wavelength lidar equations apply the two-way transmittance correction to the raw lidar returned signal, together with squared distance correction, before the backscattering ratio is calculated. In our case the available information we have the backscattering ratios which have been derived without conducting the two-way transmittance correction (G-66) for any species. That is the reason that this correction was included in the retrieval of $\beta_a(694,z)$ in equation (8). However only the molecular and ozone two way transmittance corrections ($T_m(694,z)$, $T_{O3}(694,z)$) were included in this step.*

*As explained above, the aerosol two-way transmittance correction, $T_a(532,z)$ , was deliberately postponed until the final step to derive $\alpha_a^{Ta}(532,z)$ , due to the available contemporaneous measurement information for AOD being at λ = 500nm (see Supplementary Material). We converted the measured AOD at 500nm to 532nm, using Ångstrom exponents covering the cited wavelength range from 1995 to 2019 from the nearest Aerosol Robotic Network (AERONET, 2020) stations. Although the tropospheric aerosol layer in the eastern US will have had different physical and chemical properties in the 1960s (e.g. Went, 1960; Husar et al., 1991), this is only a small change in wavelength, with the method then introducing much less error than had we converted from 500nm to 694nm to apply the aerosol attenuation correction. We note that monthly mean TAOD at 532 nm from Blue Hill Observatory, MA, from 1961 to 1966, were measured to be in the range from 0.1 to 0.4, consistent with the elevated background TAOD reported for the Eastern US during the sixties of the 20th century (Husar et al., 1981; Supplement 1).*

*We produced a first guess $T_a(532,z)_*$ for each measurement day at each site, in the range from 11 to 24 km. The $T_a(532,11km)_*$, a unique value from the lidar altitude to 11 km, was calculated using Eq.(7) and the TAOD value for the month the measurement was conducted. $T_a(532,z)_*$ values from 12 to 24 km were calculated using Eq.(7) and the uncorrected $\alpha_a(532,z)$.*

*The first guess aerosol scattering corrected by the total two-way transmittance ($\alpha_a^{Ta}(532,z)_*$), was derived applying the correction for the two-way aerosol transmittance $T_a(532,z)$*

$$\alpha_a^{Ta}(532,z) = \frac{\alpha_a(532,z)}{T_a(532,z)} \qquad (10)$$

*Since we are using the measured TAOD, which includes the stratospheric AOD, to calculate the first guess $T_a(532,z)_*$ we applied a second step after producing a first guess $\alpha_a^{Ta}(532,z)_*$ profile, where we then calculate the stratospheric AOD ($sAOD_*$), integrating $\alpha_a^{Ta}(532,z)_*$ between 12 and 24 km Then $sAOD_*$ is used in Eq.(11) to estimate the tropospheric AOD (tAOD) for each measurement:*

$$tAOD = TAOD - sAOD_* \qquad (11)$$

*Then the former $T_a(532,11km)_*$ values for each measurement will be replaced by new ones derived using the calculated tAOD corresponding to each measurement. Then the final profile of $T_a(532,z)$ for each measurement will consists of the the new $T_a(532,11km)$ calculated using tAOD and the already derived $T_a(532,z)$ values from 12 to 24 km that were calculated using the uncorrected $\alpha_a(532,z)$ in Eq.(7). Those profiles of $T_a(532,z)$ are applied in equation (10) producing the definitive values of $\alpha_a^{Ta}(532,z)$.*

*The method we applied uses monthly means of the total AOD (TAOD) from the sunphotometer, converted from 500 to 532 nm (described above), assuming this to be tropospheric (tAOD) in the first step, to produce for each site a $T_a(532,z)_*$ profile so as to account for the tropospheric aerosol transmittance from the lidar altitude across the troposphere up to 11 km and the stratospheric aerosol transmittance in the lower stratosphere from 12 to 24 km. However, when the stratospheric AOD ($sAOD_*$) is calculated in the next step (from the first guess $\alpha_a^{Ta}(532,z)_*$), the resulting first guess total AOD (tAOD + $sAOD_*$) will be higher than the observed TAOD. The second step is aimed to estimate the magnitude of a consistent value of tAOD for each measurement, to comply with the constraint $tAOD + sAOD \leq TAOD$.*

*Although more iteration of those final steps would be possible, with the high magnitude of the estimated $\alpha_a^{Ta}(532,z)$ mean error, around 60%, compared to an estimated 15-20% maximum improvement achieved by the iteration procedure, we do not believe those additional calculations would be worthwhile in this case."*

3. Lines 203-204: the authors say, referring to Eq. (9) that "EB$_c$(z, t) are the altitude and time dependent backscattering to extinction conversion coefficients from λ = 694 nm to λ = 532 nm" but only "532" appears on both sides of Eq. (2), and the reference given (Jäger and Deshler, 2003) shows that they actually convert

from backscatter to extinction at 532 nm. The conversion from 694 to 532 nm has been made in Eq. (7) through the kb(z,t) exponent.

The reviewer is right. We have re-worded the text in that section to more clearly explain the methodology (see the excerpts of text from the improved manuscript at the end of our reply to the reviewer's point 1).

4.  If $EB_c(z,t)$ (and kb(z,t) by the way), are time dependent, how and where is the time dependence taken into account? I suspect it is used to calculate the $\beta_a$ and $\alpha_a$ uncertainties (Eqs. (20) and (21)), but this should be stated the first time the coefficients appear. Otherwise it is misleading, because the left-hand sides of Eqs. (7) and (9) should have a time dependence. Which are the "nominal" values used for kb(z,t) and $EB_c(z,t)$, around which their uncertainties have been used to calculate the $\beta_a$ and $\alpha_a$ uncertainties in Eqs. (20) and (21)?

We have re-worded the text in that section to more clearly explain the methodology (see the excerpts of text from the improved manuscript at the end of our reply to the reviewer's point 1).

**Other concerns**

1.  Line 143: it would be helpful to explain that SR(694,z) means the backscattering ratio at 694 nm and range z.

Added

2.  Line after Eq. (1): while $\frac{dn(z)}{dt}$ is the photoelectron flux (electrons/s) resulting from scattering at range z, defining n(z) as "the number of photons at the altitude z" does not make much sense. In the explanation of Eq. (1), leave $\frac{dn(z)}{dt}$ as the photoelectron flux (electrons/s) resulting from the photons scattered at range z.

Corrected.

Erased: "*n(z) is the number of photons at the altitude z*".

Added: *"(electrons/s)" after "The average photoelectron flux…"*

3.  The authors say that they have eliminated the squared superscript to indicate the two-way transmittance, but this has not been done consistently. There are examples of the superscript still remaining (for example, but not only, in lines 149,150 and 177).

The reviewer is right. Corrected. The squared superscript was eliminated in lines 149, 150, 152, 178, 195 and 206.

In figure 5 the labels of both panels were corrected, replacing respectively:

"*Nantucket sounding (No T-2 Corr.)*" by " $\alpha_a$ "Nantucket sounding"

"*Nantucket sounding (T-2 Corr.)*"     by " $\alpha_a^{Ta}$ "Nantucket sounding"

4.  There is still one instance (line 177) where SRo(z) is used.

The reviewer is right. Corrected. In lines 177, 470, 489, 490, 492, 493, 494, 497 and 498.

5.  Line 215: "This profile of Ta(532,z) is applied in equation (11)". Equation (10) is probably

meant.

The reviewer is right. Corrected.

**Other Changes-Corrections made by the authors in the current version:**

1) The reference:

Antuña-Marrero, J. C., Mann, G. W. , Keckhut, P., S. Avdyushin, B. Nardi and and L. W. Thomason, Ship-borne lidar measurements showing the progression of the tropical reservoir of volcanic aerosol after the June 1991 Pinatubo eruption. , *Earth Syst. Sci. Data,* (*Under Discussion*), https://doi.org/10.5194/essd-2020-81, 2020b.

Was updated to the one from the published paper:

Antuña-Marrero, J.-C., Mann, G. W., Keckhut, P., Avdyushin, S., Nardi, B., and Thomason, L. W.: Shipborne lidar measurements showing the progression of the tropical reservoir of volcanic aerosol after the June 1991 Pinatubo eruption, Earth Syst. Sci. Data, 12, 2843–2851, https://doi.org/10.5194/essd-12-2843-2020, 2020a.

2) We have added Dhomse et al. (2021) as a 2nd citation for the UM-UKCA Agung aerosol dataset, in addition to the Dhomse et al. (2020) ACP paper, since the 6Tg Agung simulation is now published as a "complete" volcanic forcing dataset, in the sense that these simulations are now provided in a form for use in climate model simulations. That is, they're provided as SW & LW waveband-mapped aerosol optical properties (extinction, absorption and asymmetry parameter). The 3 lines we have added the Dhomse et al. (2021) reference are listed below:

Line 536 -- changed "(Dhomse et al., 2020)" to "(Dhomse et al., 2020; Dhomse et al., 2021)".

Line 571 -- there was a missing cite on line 590 -- added "(Dhomse et al., 2020; Dhomse et al., 2021)" after "UM-UKCA Agung aerosol simulations".

Line 579 (caption to Figure 12) -- changed "(Dhomse et al., 2020)" to "(Dhomse et al., 2020; Dhomse et al., 2021)".

The added reference is:

Dhomse, S. S., W. Feng, A. Rap, K. S. Carslaw, N. Bellouin and G. W. Mann, 2021, "SMURPHS/ACSIS Agung volcanic forcing dataset (mapped to UM wavebands) -- from HErSEA ensemble of interactive strat-aerosol GA4 UM-UKCA runs (Dhomse et al., 2020, ACP)" (Version v1) [Data set]. Zenodo. http://doi.org/10.5281/zenodo.4744687